# CellRank 2: unified fate mapping in multiview single-cell data

**Philipp Weiler** [1,2,8], **Marius Lange** [1,2,3,8], **Michal Klein**[1,4], **Dana Pe'er**[5,6] & **Fabian Theis** [1,2,7] ✉

Single-cell RNA sequencing allows us to model cellular state dynamics and fate decisions using expression similarity or RNA velocity to reconstruct state-change trajectories; however, trajectory inference does not incorporate valuable time point information or utilize additional modalities, whereas methods that address these different data views cannot be combined or do not scale. Here we present CellRank 2, a versatile and scalable framework to study cellular fate using multiview single-cell data of up to millions of cells in a unified fashion. CellRank 2 consistently recovers terminal states and fate probabilities across data modalities in human hematopoiesis and endodermal development. Our framework also allows combining transitions within and across experimental time points, a feature we use to recover genes promoting medullary thymic epithelial cell formation during pharyngeal endoderm development. Moreover, we enable estimating cell-specific transcription and degradation rates from metabolic-labeling data, which we apply to an intestinal organoid system to delineate differentiation trajectories and pinpoint regulatory strategies.

Single-cell assays uncover cellular heterogeneity at unprecedented resolution and scale, allowing complex differentiation trajectories to be reconstructed using computational approaches[1–5]. While these trajectory inference (TI) methods have uncovered numerous biological insights[6], they are typically designed for snapshot single-cell RNA sequencing data and cannot accommodate additional information relevant for understanding cell-state dynamics, including experimental time points, multi-modal measurements, RNA velocity[7,8] and metabolic labeling[9–13].

We, and others, have developed methods to analyze emerging data modalities, such as CellRank[14] for RNA velocity, Waddington optimal transport (WOT)[15] for experimental time points and dynamo[16] for metabolic-labeling data; however, each method only addresses a single modality, thereby ignoring much of the upcoming multi-modal information for trajectory analysis. This specialization renders many biological systems inaccessible; for example, adult hematopoiesis violates assumptions of current RNA velocity models[17], precluding us from applying CellRank to this well-studied system and prompting the question of whether the algorithm could be developed further to reconstruct differentiation dynamics using another aspect of these data.

To address this challenge, we decompose TI into two components, modality-specific modeling of cell transitions, followed by modality-agnostic TI, and developed CellRank 2, a robust, modular framework to analyze multiview data from millions of cells. CellRank 2 generalizes CellRank to exploit the full potential of alternative sources of information, such as pseudotime and developmental potential, and new data modalities, such as experimental time points and metabolic labels, to study complex cellular state changes and identify initial and terminal states, fate probabilities and lineage-correlated genes. Compared to our earlier work, the new framework is modular, applicable to many more data modalities and substantially faster (Methods).

[1]Institute of Computational Biology, Department of Computational Health, Helmholtz Munich, Munich, Germany. [2]School of Computation, Information and Technology, Technical University of Munich, Munich, Germany. [3]Department of Biosystems Science and Engineering, ETH Zürich, Basel, Switzerland. [4]Machine Learning Research, Apple, Paris, France. [5]Computational and Systems Biology Program, Sloan Kettering Institute, Memorial Sloan Kettering Cancer Center, New York, NY, USA. [6]Howard Hughes Medical Institute, Chevy Chase, MD, USA. [7]TUM School of Life Sciences Weihenstephan, Technical University of Munich, Munich, Germany. [8]These authors contributed equally: Philipp Weiler, Marius Lange. ✉e-mail: fabian.theis@helmholtz-munich.de

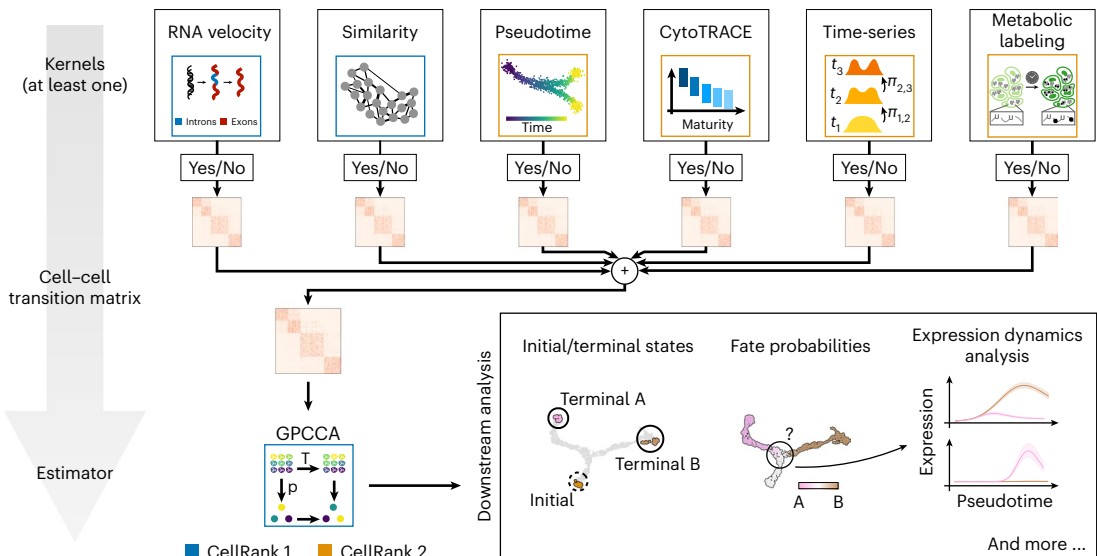

**Fig. 1 | CellRank 2 provides a unified framework for studying single-cell fate decisions using Markov chains.** CellRank 2 uses a modular design. Data and problem-specific kernels calculate a cell–cell transition matrix inducing a Markov chain (MC); of these kernels, at least one has to be used, but multiple can be combined via a kernel combination (Methods). Estimators analyze the MC to infer initial and terminal states, fate probabilities and lineage-correlated genes. Using fate probabilities and a pseudotime allows for studying gene expression changes during lineage priming. Features inherited from the original CellRank implementation are indicated in blue and new features are in orange.

We demonstrate CellRank 2's flexibility across a series of applications: using a pseudotime in a hematopoiesis context, determining developmental potentials during embryoid body formation and combining experimental time points with intra-time point information for pharyngeal endoderm development. Our approach for incorporating experimental time recovers terminal states more faithfully compared to traditional approaches mapping between time points[15], allows studying gene expression change continuously across time and predicts putative progenitors missed by alternative approaches. We also introduce a new computational approach for learning cellular dynamics from metabolic-labeling data and show that it reveals regulatory mechanisms by recovering cell-specific transcription and degradation rates in mouse intestinal organoids.

## Results

### A modular framework for studying state-change trajectories

CellRank 2 models cell-state dynamics from multiview single-cell data. It automatically determines initial and terminal states, computes fate probabilities, charts trajectory-specific gene expression trends and identifies lineage-correlated genes (Methods). A broad and extensible range of biological scenarios can be studied using its robust, scalable and modular design (Methods).

Similar to CellRank[14], we employ a probabilistic system description wherein each cell constitutes one state in a Markov chain with edges representing cell–cell transition probabilities; however, we now enable deriving these transition probabilities from various biological priors. Following previous successful TI approaches[1–5], we assume gradual, memoryless transitions of cells along the phenotypic manifold. The assumption of memoryless transitions is justified as we model average cellular behavior (Methods).

To allow broad applicability, we divide CellRank 2 into kernels for computing a cell–cell transition matrix based on multiview single-cell data and estimators for analyzing the transition matrix to identify initial and terminal states, compute fate probabilities and perform other downstream tasks. CellRank 2 provides a set of diverse kernels that derive transition probabilities based on gene expression, RNA velocity, pseudotime, developmental potentials, experimental time points and metabolically labeled data (Fig. 1a and Extended Data Fig. 1). Depending on the dataset and the biological question, we use a single kernel or combine several kernels into multiview Markov chains. For an initial, qualitative overview of recovered cellular dynamics, we introduce a random walk-based visualization scheme (Methods).

For many biological processes, the starting point can be quantified robustly and cells ordered along a pseudotime. We propose to use this fact by biasing the edges of a nearest-neighbor graph toward mature cell states to estimate cell–cell transitions (Methods); developmental potentials can be used similarly. CellRank 2 generalizes earlier concepts[5,18] to arbitrary pseudotimes and atlas-scale datasets with the PseudotimeKernel and CytoTRACEKernel. More complex systems with unknown initial states or longer developmental time scales can be captured faithfully through multiple experimental time points. To reconstruct the overall differentiation dynamics described by both across and within time points, we extend classical optimal transport (OT)[15], in particular WOT[15], with our RealTimeKernel to include within-time-point dynamics (Methods). In contrast, metabolic labels offer an experimental approach to overcome the discrete nature of distinct experimental time points[9,11,12]. Based on this information, we developed an inference approach quantifying kinetic rates that allow us to infer cell transitions (Methods). In the following, we provide details on each kernel and demonstrate the versatility of our approach through diverse applications. Finally, various kernels may be combined to yield a more complete picture of cellular dynamics through multiview modeling.

Once we have inferred a transition matrix, we use an estimator module[19,20] to uncover cellular trajectories, including initial and terminal states, fate probabilities and lineage-correlated genes. Critically, estimators are view-independent and are, thus, applicable to any transition matrix (Methods). To scale these computations to large datasets, we assume that each cell gives rise to only a small set of potential descendants. This assumption yields sparse transition matrices for every kernel and allows CellRank 2 to compute fate probabilities 30 times faster than CellRank (Extended Data Fig. 2a and Methods).

The modular and robust design makes CellRank 2 a flexible framework for the probabilistic analysis of state dynamics in multiview single-cell data; it enables the rapid adaptation of computational workflows to emerging data modalities, including lineage tracing[21–23]

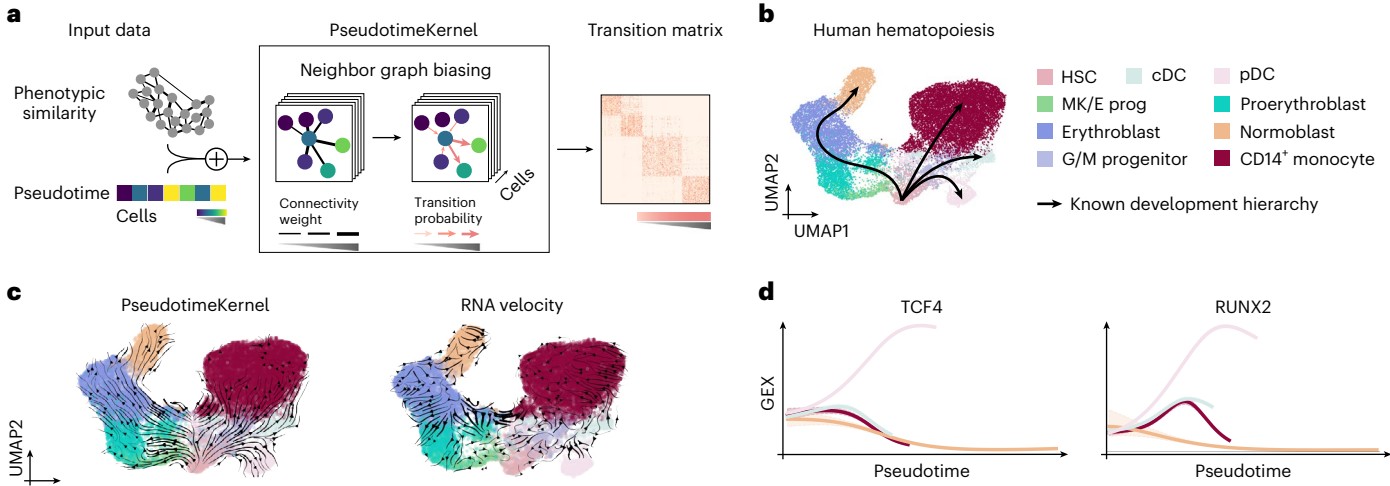

**Fig. 2 | Leveraging pseudotemporal ordering for cellular fate mapping.**
**a**, The PseudotimeKernel biases the edges of a phenotypic similarity-based nearest-neighbor graph toward increasing pseudotime, defining cell–cell transition probabilities. **b,c**, UMAP embedding of 24,440 peripheral blood mononuclear cells[26], colored by cell type (cDC; G/M progenitor, granulocyte/myeloid progenitor; HSC; MK/E prog, megakaryocyte/erythrocyte progenitors; pDC). We illustrate the well-known differentiation hierarchy in black (**b**) as well

as projected velocity fields based on the PseudotimeKernel (**c**, left) and RNA velocity (**c**, right). **d**, Correlating fate probabilities with gene expression recovers known lineage drivers for the pDC lineage[27,28]. We show lineage-specific trends as proposed in our earlier work[14] by fitting generalized additive models to gene expression (*y* axis) in pseudotime (*x* axis); the contribution of each cell to each lineage is weighted according to CellRank 2-recovered fate probabilities. Colors correspond to lineages as shown in **b**.

and spatiotemporal data[24], the support of new data modalities with kernels and the support of new analyses with estimators (Methods).

### Overcoming RNA velocity limitations

RNA velocity infers incorrect dynamics in steady-state human hematopoiesis due to violated model assumptions, even though pseudotime can be recovered faithfully (Supplementary Fig. 1 and Supplementary Note 1). Specifically, the assumption of constant rates made by conventional RNA velocity models is violated and genes important for this system exhibit high noise or low coverage. The remarkable success of traditional pseudotime approaches in systems with well-known initial conditions motivated us to circumvent RNA velocity limitations by developing the PseudotimeKernel, which computes pseudotime-informed transition probabilities and a corresponding vector field (Fig. 2a and Methods). Building upon conceptual ideas proposed for Palantir[5], our approach generalizes to any precomputed pseudotime and uses a soft weighting scheme[25].

We applied the PseudotimeKernel to human hematopoiesis[26] and computed transition probabilities based on diffusion pseudotime (DPT)[1] for the normoblast, monocyte and dendritic cell lineages (Fig. 2b and Extended Data Fig. 3a). The PseudotimeKernel correctly recovered all four terminal states and the initial state (Extended Data Fig. 3b,c and Methods). To additionally visualize the recovered dynamics, we generalized the streamline projection scheme from RNA velocity[7,8] to any neighbor-graph-based kernel (Fig. 2c and Methods). We correlated gene expression with lineage-specific fate probabilities to identify candidate genes that may be involved in lineage commitment (Methods); this approach correctly identified the transcription factors RUNX2 and TCF4 as regulators of the plasmacytoid dendritic cell (pDC) lineage[27,28] (Fig. 2d).

Compared to the PseudotimeKernel, an RNA velocity-based analysis failed to recover the classical dendritic cell (cDC) lineage (Extended Data Fig. 3b) and fate probabilities assigned by the VelocityKernel violated the known lineage commitment and hierarchy, including high transition probabilities from proerythroblast and erythroblast cells to monocytes instead of normoblasts (Extended Data Fig. 3d). This inconsistency to known ground truth transitions stems from

violated assumptions of the RNA velocity model (Supplementary Fig. 1 and Supplementary Note 1). For an additional, quantitative metric, we computed the log ratio of the kernels' cross-boundary correctness (CBC) scores[29] (Methods). This metric provides a quantitative measure of two kernel-derived cell–cell transition matrices for known transitions between cell states. As indicated by the visualization of fate probabilities, the PseudotimeKernel significantly outperforms the competing approach for most cell state transitions (6 out of 8; Extended Data Fig. 3e and Methods). As an alternative comparison, we introduce the terminal state identification (TSI) score to quantify the identification of known terminal states compared to an optimal identification strategy (TSI = 1; Methods). Our pseudotime-based approach again outperformed the RNA-velocity-based alternative (TSI = 0.9 versus TSI = 0.81; Extended Data Fig. 3f).

Our PseudotimeKernel generalizes to any pseudotime, allowing users to choose the algorithm most suitable for their dataset[30]. In systems with simpler differentiation hierarchies and known initial states, CellRank 2's PseudotimeKernel yields additional insights into terminal states and fate commitment compared to classical pseudotime approaches.

### Learning vector fields from developmental potentials

Pseudotime inference requires the initial state to be specified. If the initial state is not known, CytoTRACE[18] can be used to infer a stemness score by assuming that, on average, naive cells express more genes than mature cells. We found this assumption to be effective for many early developmental scenarios, but, critically, CytoTRACE does not scale in time and memory usage when applied to large datasets and does not resolve individual trajectories through terminal states and fate probabilities (Extended Data Fig. 2b). We thus developed the CytoTRACEKernel by revising the original CytoTRACE approach, such that edges of *k*-nearest-neighbor graphs point toward increasing maturity and quantify cell–cell transition probabilities on atlas-scale datasets (Extended Data Fig. 4a and Methods). Results from our kernel agree with the original approach across multiple datasets (Extended Data Fig. 4b,c and Methods). Further, we compared computational performance on a mouse organogenesis atlas[31] containing 1.3 million cells. While the

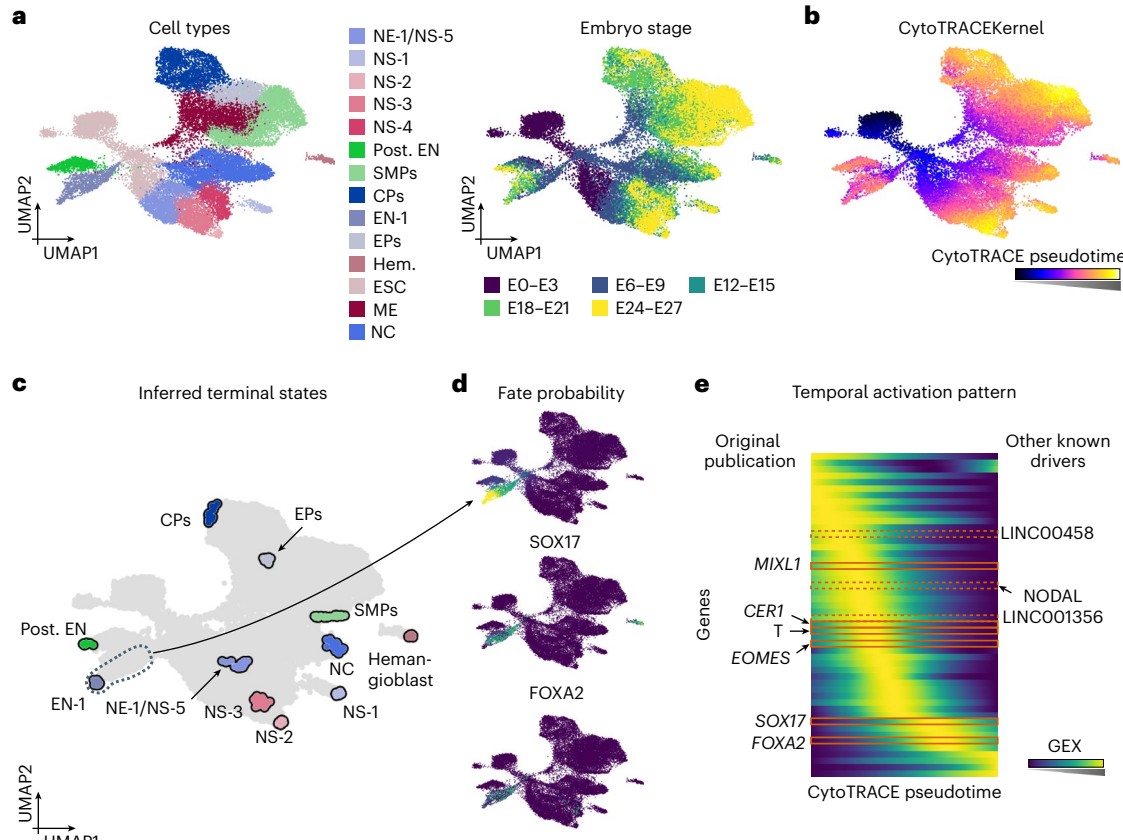

**Fig. 3 | The CytoTRACEKernel recovers temporal gene activation.**
**a,b**, UMAP embedding of 31,029 embryoid body cells[32], colored by original cluster annotation (CPs, cardiac precursors; EN, endoderm; EPs, epicardial precursors; ESC, embryonic stem cell; Hem., hemangioblast; ME, mesoderm; NC, neural crest; NE, neuroectoderm; NS, neuronal subtypes; Post. EN, posterior endoderm; SMPs, smooth muscle precursors; **a**, left), embryo stage (E) (**a**, right) and pseudotimes from the CytoTRACEKernel (**b**). **c**, Terminal states inferred using the CytoTRACEKernel. **d**, UMAP embedding colored by fate probabilities (top) and gene expression of recovered drivers (bottom) of the endoderm lineage. **e**, Smoothed gene expression along the CytoTRACE pseudotime for the automatically identified top 50 lineage-correlated genes, sorted according to their pseudotime peak. Genes identified in the original publication (left) and known drivers (right)[32–34] additionally recovered with CellRank 2 are indicated.

original implementation failed above 80,000 cells, our adaptation ran on the full dataset in under 2 min (Extended Data Fig. 2b).

We applied the CytoTRACEKernel to study endoderm development in pluripotent cell aggregates known as embryoid bodies[32] (Fig. 3a). The CytoTRACE-based pseudotime increased smoothly throughout all experimental time points, as expected (Fig. 3b and Extended Data Fig. 5a) and allowed us to identify 10 of 11 terminal cell populations and the correct initial state (Fig. 3c and Extended Data Fig. 5b). In contrast, Palantir[5] and DPT[1] identified a bimodal population distribution of early cells in the first stage, resulting in a compressed range of pseudotimes for all other populations and stages (Extended Data Fig. 5a).

The endoderm gives rise to internal organs; thus, we correlated fate probabilities with gene expression to infer lineage-correlated genes that may direct organogenesis, identifying the MIXL1, FOXA2 and SOX17 transcription factors (TFs), in agreement with the original publication[32] (Fig. 3d). To uncover potential upstream regulators of these TFs, we visualize smooth gene expression trends of top-ranked lineage-correlated genes of the endoderm trajectory in a heatmap and sorted genes according to their peak in CytoTRACE pseudotime (Fig. 3e). We found LINC00458, LINC01356, NODAL and nine TFs to peak before FOXA2 and SOX17. All are known mouse endodermal development genes[33,34] and our prediction that LINC00458 expression peaks before LINC01356 has also been observed previously[33].

CellRank 2's CytoTRACEKernel allowed us to infer cellular dynamics from a snapshot of endoderm development without having to specify an initial state for pseudotime computation. We recovered terminal states, known driver genes and their temporal activation pattern.

## Adding a temporal resolution to fate mapping

Single-cell time series are increasingly popular for studying non-steady-state differentiation programs. The computational challenge lies in matching cells sequenced at different time points to reconstruct trajectories of state change. Most previous methods have either determined population dynamics[35] or used OT[15] but ignore transitions within time points that contain valuable information for directing transitions and detecting terminal states. We developed the RealTimeKernel, which combines WOT-computed[15] inter-time-point transitions with similarity-based intra-time-point transitions to allow for multiview modeling (Fig. 4a and Methods). Notably, considering inter-time-point transitions enables unbiased identification of terminal and initial states from time-course studies through a more granular mapping of cell fate (Fig. 4b and Extended Data Fig. 6a–c). To gain a preliminary understanding of the underlying differentiation dynamics, we visualize high-dimensional RealTimeKernel-derived random walks in the embedding space (Extended Data Fig. 6b).

Many OT implementations, including WOT[15], use entropic regularization[36] to speed up computation; however, this practice introduces dense transition matrices, which slows downstream applications, hindering us from analyzing large datasets. We therefore developed an adaptive thresholding scheme to sparsify transition matrices (Methods), yielding ninefold and 56-fold faster macrostate and fate

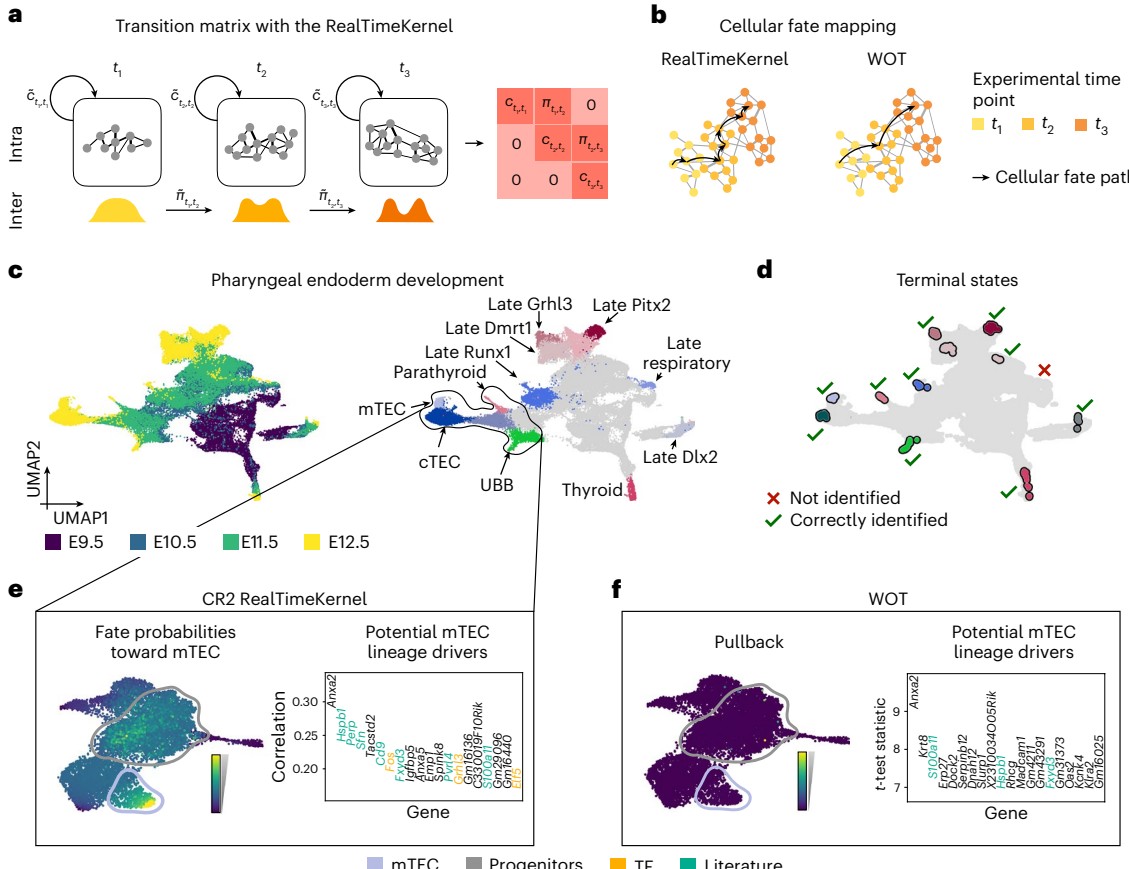

**Fig. 4 | Inferring state trajectories through time-resolved measurements. a**, The RealTimeKernel combines across time-point transitions from WOT[15] with within-time-point transitions from gene expression similarity to account for the asynchrony observed in many cellular processes. All views are combined in a single transition matrix. **b**, By including within-time-point information, the RealTimeKernel enables recovering more granular state transitions; WOT only considers transitions between consecutive time points. **c**, UMAP embedding of pharyngeal organ development[37] (*n* = 55,044 cells) colored by embryonic day (E; left) and original cell type annotation (right; cTEC, mTEC, UBB); gray color encodes early, uncommitted cells. **d**, Using the RealTimeKernel, CellRank 2 correctly identifies 10 out of 11 terminal states. The black outline highlights mTECs and potential precursor cells. **e,f**, Fate probabilities toward the mTEC terminal state (left) and top 20 lineage-correlated genes identified (right) based on the RealTimeKernel (**e**) or WOT's pullback distribution (**f**). We highlight TFs in yellow and known mTEC development genes in green. CellRank 2 identifies putative drivers by correlating fate probabilities with gene expression, WOT by comparing high- and low-probability cells.

probability computation, respectively, on a mouse embryonic fibroblast (MEF) reprogramming dataset[15] (Extended Data Fig. 2c). To validate our thresholding scheme, we correlated fate probabilities toward the four terminal states, with and without thresholding, and found very high correlations within each lineage (Pearson correlation coefficient *r* > 99; Extended Data Fig. 6c and Methods). Overall, we obtained a perfect TSI score using the RealTimeKernel (TSI = 1) but a suboptimal one for the VelocityKernel (TSI = 0.79) (Extended Data Fig. 6d).

Numerous applications, including gene trend plotting and lineage-correlated gene identification, require continuous temporal information rather than discrete time points. Thus, we comprised a new, real-time informed pseudotime approach, which uses experimental time points but embeds them in the continuous landscape of expression changes (Methods). As expected in this system, our new pseudotime indeed correlates better with experimental time than traditional pseudotime approaches on the MEF reprogramming data (Extended Data Fig. 7a–c). Compared to WOT, we enable studying gradual fate establishment along a continuous axis (Extended Data Fig. 7d–g).

The pharyngeal endoderm, an embryonic tissue, plays a crucial role in patterning the pharyngeal region and developing organs[37], such as the parathyroid, thyroid and thymus[38–40]. Multiple experimental time points can capture its development, making it an ideal candidate

system for our RealTimeKernel. We analyzed gene expression change from embryonic days (E) 9.5 to 12.5 (ref. 37) (Fig. 4c) and automatically recovered the initial state (Extended Data Fig. 8a) and 10 of the 11 terminal states manually assigned in the original publication (Fig. 4d). Additionally, using the RealTimeKernel led to a higher TSI score compared to the VelocityKernel (TSI = 0.92 versus TSI = 0.46, Extended Data Fig. 8b). Correlating fate probabilities with gene expression correctly recovered known lineage drivers of the parathyroid (*Gcm2*)[41], thyroid (*Hhex*)[42] and thymus (*Foxn1*)[37] (Supplementary Fig. 2).

To disentangle the trajectory leading to medullary thymic epithelial cells, a stromal cell type associated with thymic adhesion[43], we first took the subset of parathyroid, ultimobranchial body, medullary and cortical thymic epithelial cells (mTECs and cTECs) and their progenitors (Extended Data Fig. 8c and Methods). We successfully recovered the initial state (Extended Data Fig. 8c) and each terminal state and scored a higher TSI metric compared to relying on RNA velocity estimates (TSI = 1.0 versus TSI = 0.91; Extended Data Fig. 8d). Computing fate probabilities toward terminal states, we discovered a progenitor cell cluster with an increased probability of assuming mTEC fate (Fig. 4e and Extended Data Fig. 9a,b). It is easy to overlook this putative mTEC ancestor cluster in the two-dimensional uniform manifold approximation and projection (UMAP) embedding, highlighting the importance of our high-dimensional fate analysis (Extended Data Fig. 9c). Next, we

used our correlation-based analysis to identify possible drivers of this fate decision and found TFs (*Fos*, *Grhl3* and *Elf5*) and genes relevant for the thymus lineage among the 20 genes with highest correlation (Fig. 4e): *Sfn* and *Perp* are part of the p53 signaling pathway controlling murine mTEC differentiation[44,45]; additionally, the TF p63 targets *Perp* and is involved in murine mTEC differentiation[46,47]. Similarly, we recovered previously reported markers of murine mTECs, including *Grhl3*, *Pvrl4* and *Cd9* (ref. 37,48). In addition to these known markers of mTECs in mice, our top-ranked genes also included *S100a11* and *Fxyd3*, markers of mTEC subpopulations in different human settings[45,49–52], and *Hspb1*, a marker of later-stage murine mTECs[53]. Notably, the original study of our dataset identified the TF *Grhl3* as a putative early mTEC marker with higher specificity compared to markers traditionally used.

Unlike CellRank 2, WOT relies solely on inter-time-point information. Applied to the pharyngeal endoderm data, it failed to identify the putative mTEC ancestor cluster. Additionally, even when we leveraged the knowledge of putative mTEC progenitors identified by the RealTimeKernel at the earlier experimental time points, classical WOT identified fewer driver gene candidates with known functions in mTEC development at these time points (Fig. 4f and Extended Data Fig. 9d,e). We speculate that this decrease in performance is caused by WOT relying on pullback distributions, which assign a likelihood to each early-day cell to differentiate into any late-day cell but do not take intra-time point dynamics into account. In contrast, CellRank 2 computes continuous fate probabilities with a global transition matrix, combining transitions within and across time points (Methods). Finally, classical differential expression testing also recovered fewer known driver genes and TFs as our correlation-based analysis (Extended Data Fig. 9f).

Our RealTimeKernel incorporates gene expression changes within and across experimental time points. Notably, these complementary views allowed identifying a putative progenitor population and substantially more relevant drivers compared to approaches focusing on a single data view.

### Estimating kinetic rates and fate from metabolic labels

The destructive nature of standard single-cell protocols prohibits directly examining gene expression changes over time. Metabolic labeling of newly transcribed mRNA molecules[9,11,12,31]; however, yields time-resolved single-cell RNA measurements that should substantially improve our ability to learn system dynamics. The temporal resolution is in the order of minutes to hours and thus much finer compared to typical time-course studies. We developed an approach to learn directed state-change trajectories from metabolic-labeling data using pulse–chase experiments (Fig. 5a and Methods).

Similar to previous approaches[9], we model mRNA dynamics through a dynamical system, including mRNA molecule transcription and degradation rates[9]. We estimate these rates for each cell and gene by considering the dynamical information conveyed through metabolic labels (Fig. 5b and Methods). Based on a cell–cell similarity graph, for each cell, gene and labeling time, we identify a neighborhood in which a sufficient number of cells express the given gene. Next, we estimate transcription and degradation rates based on these cell sets by minimizing the squared Euclidean distance between observed and estimated transcripts. With these parameters, we infer a high-dimensional velocity vector field used to obtain cell–cell transition probabilities with the VelocityKernel.

We applied our devised method to data from murine intestinal organoids labeled with scEU-seq[9], focusing on the enterocyte, enteroendocrine, goblet and paneth lineages (Extended Data Fig. 10a and Methods). Following parameter estimation, we computed the underlying velocity field, inferred transition probabilities and recovered all four terminal states (Fig. 5c). Similarly, we recovered all four terminal states using classical RNA velocity with the VelocityKernel, that is, using CellRank 1 (Extended Data Fig. 10b). We assessed the quality of inferred terminal states via cell-type purity, defined as the percentage of the most abundant cell type, reasoning that a high cell-type purity results from a low inference uncertainty of the underlying transition matrix (Methods). Indeed, we observed a high cell-type purity (85% on average) for each terminal state using the velocity field derived from the metabolic-labeling information but lower cell-type purity (67% on average) when relying on classical RNA velocity estimates (Extended Data Fig. 10c). Additionally, CellRank 2 outperformed CellRank 1 in terms of the TSI score (TSI = 0.81 and TSI = 0.71, respectively; Extended Data Fig. 10d).

We compared our approach with dynamo[16], an alternative method for estimating cellular dynamics based on metabolic-labeling data. In contrast to our approach, dynamo relies on a steady-state assumption, only uses a small subset of cells for parameter inference, does not estimate cell-specific rates and infers cellular trajectories deterministically. Applied to the organoid data, dynamo only recovered the enterocyte population as a terminal state (Fig. 5c, Extended Data Fig. 10e and Methods).

Beyond identifying the most mature cell population in each lineage, we asked whether our approach ranked known lineage drivers higher than competing approaches that do incorporate labeling information (dynamo) or that do not (CellRank 1 with scVelo's dynamical model of RNA velocity and a random baseline). To assess the quality of each method's gene ranking, we curated an optimally ranked list of known regulators and markers[54] of each lineage and compared each method's ranking to it (Methods). As dynamo only identified enterocytes as a terminal cluster, it could not rank drivers of any other lineage. Using CellRank 2 based on labeling information achieved the best ranking for each of the four terminal states (Fig. 5d and Extended Data Fig. 10f) and notably outperformed competing approaches, including CellRank 1, both when correlating gene expression and the inferred transcription rates with fate probabilities to identify putative driver genes.

The estimated cell- and gene-specific kinetic rates enabled us to investigate how these lineage-correlated genes are regulated by mRNA transcription and degradation. Analyzing the regulatory strategies of known markers and regulators ranked among the top 100 lineage-correlated genes for the goblet lineage revealed two different regulatory strategies (Fig. 5e). The first strategy increases transcription rates with decreasing degradation rates (for example, *Spdef*, *Sytl2* and *Fcgbp*) and the second simultaneously increases transcription and degradation rates (for example, *Atp2a3*, *Tff3* and *Rassf6*); both align with earlier findings of cooperative (case 1) and destructive (case 2) regulation strategies[9]. Although it is so far not possible to directly measure transcription and degradation rates in single-cell sequencing protocols, the increase in transcription rate aligns with the role of these genes as regulators and markers of the goblet lineage[54]. Similarly, in the enterocyte lineage, this same set of genes predominantly exhibits either (1) a decreased transcription accompanied by an increase in degradation rate (cooperative) or (2) an increase/decrease of both rates (destructive; Fig. 5f).

## Discussion

CellRank 2 is a robust, modular and scalable framework to infer and study single-cell trajectories and fate decisions. By separating the inference and analysis of transition matrices via kernels and estimators, respectively, CellRank 2 accommodates diverse data modalities and overcomes the limitations of single data-type approaches in a consistent and unified manner. Our tool successfully performed pseudotime-based analysis of human hematopoiesis and deciphered gene dynamics during human endoderm development using stemness estimates. Notably, the modular and scalable design facilitated the rapid integration of each data modality and allowed CellRank 2 to analyze much larger datasets compared to previous approaches and implementations.

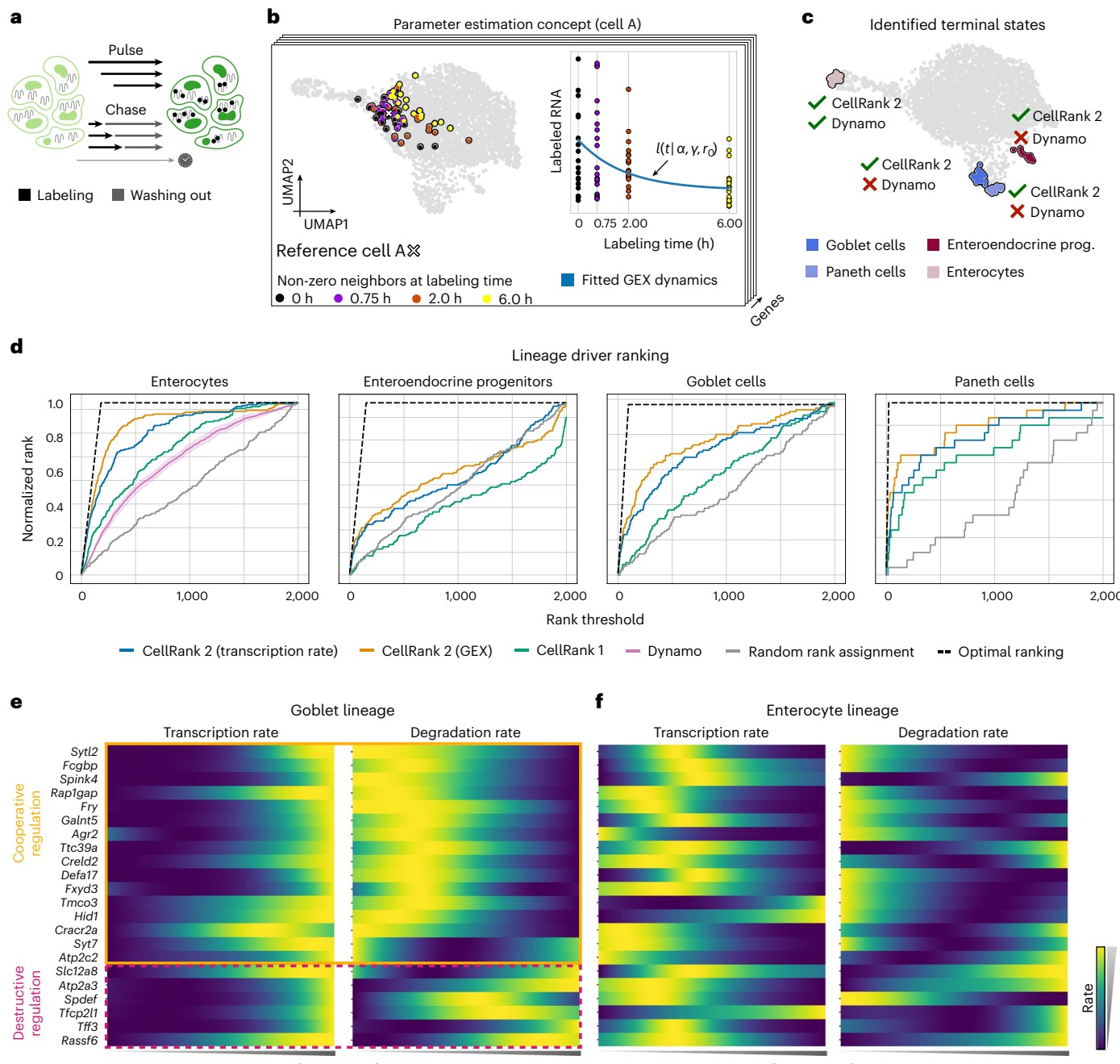

**Fig. 5 | Quantifying lineage-specific regulation strategies through metabolic labeling. a**, Cells are metabolically labeled in pulse–chase experiments[9] followed by simultaneous sequencing. Pulse experiments involve incubation with nucleoside analogs for varying durations; in chase experiments, cellular mRNA fully incorporates nucleoside analogs during a long incubation, followed by washing out of these nucleosides for varying durations. **b**, For each cell, gene and labeling duration, we identify the number of neighbors such that a predefined number of cells with non-trivial counts are included in the neighborhood, illustrated here for an exemplary cell A. These cells are then used to estimate cell and gene-specific transcription and degradation rates $\alpha$ and $\gamma$, respectively,

to model the dynamics of labeled mRNA. **c**, UMAP embedding highlighting terminal states identified using CellRank 2 and dynamo. Green ticks indicate that a method recovered the corresponding terminal state, and red crosses indicate that the terminal state was not identified. **d**, Ranking of drivers for each lineage identified by different methods. Dynamo identified only enterocytes as terminal and, thus, provides a gene ranking only for this lineage. For dynamo, the mean gene ranking and corresponding 95% confidence band are shown (Methods). **e**, Inferred transcription (left) and degradation (right) rates of top-ranked known drivers of the goblet lineage. **f**, Same as **e**, but along the enterocyte lineage.

Developing an efficient OT-based kernel allowed us to integrate time-series data, considering both inter- and intra-time point information. With this formulation, we identified a putative progenitor population of medullary thymic epithelial cells missed by methods that ignore dynamics within time points. Recently, time-course studies have been combined with genetic lineage tracing[55–58] or spatial resolution[59–61] and

emerging computational methods[21–24] use this information to map cells more faithfully across time. These enhanced inter-time point mappings can be used with our RealTimeKernel for further analysis, as demonstrated for lineage-traced *Caenorhabditis elegans* data in moslin[23] and spatiotemporal mouse embryogenesis data in moscot[24]. These applications highlight the importance of our view-agnostic framework for

analyzing increasingly large, complex and multi-modal time-course studies. Additionally, Mellon[62], a recently proposed alternative approach for continuous analysis of time-course data, could improve our mappings by incorporating their density estimates in the OT problem.

Our kernel-estimator design proved particularly valuable when integrating metabolic labeling to estimate cell-specific mRNA transcription and degradation rates. We demonstrated the ability of metabolic-labeling data to overcome the intrinsic limitations of splicing-based velocity inference by successfully identifying all lineages in gut organoid differentiation. Combining the inferred kinetic rates with CellRank 2 also makes it possible to study gene regulatory strategies underlying cellular state changes, as we showed for the goblet and enterocyte lineages. Parallel to our approach, others developed velvet[63] and storm[64] to estimate cellular dynamics from metabolic-labeling data; however, compared to our approach, velvet does not estimate transcription rates and assumes constant degradation rates across all cells. While storm relaxes this assumption, it does so only through post-processing steps. Additionally, storm relies on deterministic downstream analyses. In contrast, CellRank 2 estimates cell-specific transcription and degradation rates and offers probabilistic downstream analysis through flexible Markov-chain modeling.

Recent experimental advances combine single-cell metabolic-labeling techniques with droplet-based assays[11,65] or split-pool barcoding approaches[10,63] to label transcripts at atlas scale and demonstrate metabolic labeling for in vivo systems[66] and in the context of spatially resolved assays[13], underscoring the need for scalable analytical approaches as proposed in this study. We aim to expand our framework further by simultaneously inferring kinetic rates and ordering cells along differentiation trajectories.

We have introduced kernels that make use of different types of directional information of cellular state changes (Extended Data Fig. 1). If metabolic labels from pulse (chase) experiments for at least two (three) labeling durations are available, our proposed method to infer a metabolic-labeling informed vector field is suitable. The RealTimeKernel is applicable for time series in which time points are closely spaced with respect to the underlying dynamical process. The VelocityKernel can be used with RNA velocity for systems that meet the assumptions of RNA velocity inference methods[67,68]. Finally, the PseudotimeKernel can enhance the understanding of cellular state changes if a unique initial state is identifiable and differentiation proceeds unidirectionally, and the CytoTRACEKernel can be used when the initial state is unknown. Notably, the proposed kernels lead to different results if the underlying assumptions are violated or not sufficiently satisfied (Supplementary Note 1). For example, the VelocityKernel failed to faithfully recapitulate the known differentiation hierarchy of hematopoiesis due to unsatisfied assumptions of the RNA velocity model. Different kernels can be combined with user-defined global weights if multiple criteria are met, as we demonstrated for the RealTimeKernel; other studies used CellRank 2's kernel combinations to study the developmental processes in epicardioids[69] and to reveal the developmental history during human cortical gyrification[70], for example. In the future, we plan to introduce local kernel combinations that would involve kernel weights based on the relative position of cells within the phenotypic manifold, allowing for context-dependent integration of multiple data sources.

Identifying putative driver genes is another aspect that can be extended in future work. Currently, we rank putative driver genes by correlating fate probabilities with gene expression. Although this approach has proven powerful, as shown in various applications, it is solely based on correlation. To unravel the causal mechanisms linking molecular properties and changes to fate decisions, perturbation data and causal inference[71] can be combined with CellRank 2. This combination will ultimately enhance our understanding of underlying molecular drivers. Overall, we anticipate our framework to be crucial in understanding and conceptualizing fate choice as single-cell datasets grow in scale and diversity.

## Online content

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

## Methods

### CellRank 2: a unified framework to probabilistically model cellular state changes

To overcome the inherent limitations of RNA velocity and unify TI across different data views, we developed CellRank 2; our framework describes cellular dynamics probabilistically, as proposed in our earlier work[14]. Specifically, we introduced the first probabilistic modeling framework that automatically determines the direction of cellular state changes to extend TI beyond normal development. Generalizing this paradigm to different biological priors and guaranteeing applicability in many scenarios required us to rethink the CellRank structure entirely. To this end, we base our new version on three key principles:

1. Robustness: fate restriction is a gradual, noisy process requiring probabilistic treatment. Therefore, we use Markov chains to describe stochastic fate transitions, with each state of the Markov chain representing one cell.
2. Modularity: quantifying transition probabilities between cells is independent of analyzing them. Thus, we modularized the CellRank framework into kernels to compute transition probabilities and estimators to analyze transition probabilities. This structure guarantees flexibility in applications and is easily extensible.
3. Scalability: we assume each cell can transition into a small set of possible descendant states. Consequently, transition matrices are sparse, and computations scale to vast cell numbers (Extended Data Fig. 2).

**Innovations in CellRank 2.** Our design principles allowed us to improve our original work in three major aspects:

1. We design a modular interface that allows us to decouple the construction of a Markov chain from the process of formulating a hypothesis based on the Markov chain.
2. We introduce the PseudotimeKernel, CytoTRACEKernel and RealTimeKernel, as well as a method to infer kinetic rates from metabolic-labeling data, to render CellRank 2 applicable beyond RNA velocity; we do so by using a pseudotime, a measure of developmental potential, time-course data and metabolic-labeling information, respectively.
3. We make our framework faster by accelerating our main estimator by one order of magnitude, easier to use by refactoring our codebase and more interpretable by visualizing kernel dynamics via random walks.

**Key outputs of CellRank 2.** Although inputs to CellRank 2 are kernel-dependent (Extended Data Fig. 1), outputs are consistent across all kernels:

- Initial, intermediate and terminal states of cellular trajectories.
- Fate probabilities, quantifying how likely each cell is to reach each terminal (or intermediate) state.
- Gene expression trends specific to each identified trajectory.
- Putative driver genes of fate decisions through correlating gene expression with fate probability.
- Dedicated visualization tools for all key outputs, for example, circular embeddings for fate probabilities, heatmaps for cascades of trajectory-specific gene expression and line plots for gene trends along different trajectories.

**A conceptual overview of kernels in CellRank 2.** *Decoupling inference of transition probability from their analysis.* The typical CellRank 2 workflow consists of two steps: (1) estimating cell–cell transition probabilities and (2) deriving biological insights based on these estimates. Previously, we tied these two steps together[14] but realized that decoupling them yields a much more powerful and flexible modeling

framework. Treating each step separately is possible as analyzing transition matrices is independent of their construction. For example, estimating transition probabilities based on RNA velocity or a pseudotime does not change how initial and terminal states are inferred or fate probabilities estimated. Consequently, modularizing our problem-specific framework generalizes the corresponding analysis tools to other data modalities. The two steps of our inference workflow are conceptualized by kernels and estimators, respectively.

Kernels estimate transition matrices $T \in \mathbb{R}^{n_c \times n_c}$ at a cellular resolution with $n_c$ denoting the number of cells; row $T_{j,:}$ represents the transition probabilities of cell $j$ toward putative descendants. With CellRank 2, we provide means to quantify fate probabilities based on RNA velocity (VelocityKernel), pseudotime (PseudotimeKernel), a developmental potential (CytoTRACEKernel), experimental time points (RealTimeKernel) and metabolic labeling (metabolic-labeling-based vector field with the VelocityKernel).

*Initial state identification.* Kernel-derived transition matrices quantify probabilities of cell transitions to putative progenitor states. To estimate initial states, instead, we work with the transposed transition matrix, thereby quantifying transition probabilities from progenitor cells to their putative ancestors. Each kernel automatically row-normalizes the transposed transition matrix.

*Kernel combination.* Different data modalities may capture different aspects of biological processes. To take advantage of multiple data modalities, kernels can be combined to quantify the likely state change in a single, aggregated transition matrix. Consider two kernels $k^{(1)}$ and $k^{(2)}$ with corresponding transition matrices $T^{(1)}$ and $T^{(2)}$, respectively. CellRank 2 allows combining the two kernels into a joint kernel $k$ defined as $k = \alpha k^{(1)} + (1 - \alpha)k^{(2)}$ with a weight parameter $\alpha \in [0, 1]$. The corresponding normalized transition matrix $T$ is computed automatically and is thus, given by

$$T_{jk} = \alpha T_{jk}^{(1)} + (1 - \alpha)T_{jk}^{(2)}.$$

*The terminal state identification score for kernel comparison.* If terminal states of the studied system are known a priori, kernels can be compared by considering how well the kernels identify terminal states with an increasing number of macrostates; an optimal strategy identifies a new terminal state with every added macrostate until all terminal states have been identified. We summarize the performance of an arbitrary kernel relative to such an optimal identification with the TSI score: consider a system containing $m$ terminal states and the function $f$ that assigns each number of macrostates $n$ the corresponding number of identified terminal states. In the case of a strategy that identifies terminal states optimally, $f_{opt}$ describes the step function

$$f_{opt}(n) = \begin{cases} n, & n < m \\ m, & n \geq m. \end{cases}$$

We define the TSI score for an arbitrary kernel $\kappa$ as the area under the curve $f_\kappa$ relative to the area under the curve $f_{opt}$, that is

$$\mathrm{TSI}(\kappa) = \frac{\sum_{n=1}^{N_{max}} f_\kappa(n)}{\sum_{n=1}^{N_{max}} f_{opt}} = \frac{2}{m(1 + N_{max} - m)} \sum_{n=1}^{N_{max}} f_\kappa(n),$$

with maximum number of macrostates assessed $N_{max}$.

*Kernel comparison via the cross-boundary correctness score.* While the CellRank 2 framework aims at quantifying cell trajectories, correct transitions between coarse cell states, such as cell types, are sometimes known a priori. In such cases, the CBC score[29] can be used to compare

two kernels: Consider two cell states $C_1$ and $C_2$, where $C_2$ is a progenitor state of $C_1$, a precomputed nearest-neighbor graph with weights $w_{jk}$ between observations $j$ and $k$ and denote the neighborhood of observation $j$ by $N(j)$. The representation of observation $j$ is denoted by $x_j$; all cell representations are collected in the matrix $X$. We define the boundary of $C_1$ to $C_2$ as all cells with at least one neighbor in $C_2$ and denote it by $\partial_{1\to2}C_1$, that is

$$\partial_{1\to2}C_1 = \{j \in C_1 | \exists k \in N(j) : k \in C_2\}$$

For every boundary cell, we empirically define the velocity $v(j)$ of observation $j \in C_1$ as

$$v(j) = \sum_{k \in N(j) \cap C_2} w_{jk}(x_k - x_j).$$

Similarly, for a given kernel $\kappa$, we estimate the velocity of observation $j$ via

$$v^{(\kappa)}(j) = T_{j,:}^{(\kappa)}X - x_j,$$

where $T_{j,:}^{(\kappa)}$ denotes the $j$th row of the transition matrix computed with kernel $\kappa$. The CBC score $\beta^{(\kappa)}(j)$ of cell $j$ under kernel $\kappa$ is then given by the Pearson correlation between $v(j)$ and $v^{(\kappa)}(j)$.

To compare two kernels $\kappa_1$ and $\kappa_2$, for each observation, we compute the log ratio of the corresponding CBC scores $\beta^{(\kappa_1)}(j)$ and $\beta^{(\kappa_2)}(j)$. If the velocity estimate based on kernel $\kappa_1$ aligns more with the empirical estimate, the log ratio is positive and negative otherwise. A one-sided Welch's $t$-test can be used to test if kernel $\kappa_1$ significantly outperforms kernel $\kappa_2$.

*Visualizing kernel dynamics: random walks and projections.* Although inferred transition probabilities are predominantly used for more in-depth data analyses based on estimators, we also provide means to visualize cellular dynamics directly based on the kernel output. These visualizations are intended to provide a preliminary understanding of the underlying dynamics and serve as a starting point for further analyses. Here, we enable studying the evolution of cellular state change either based on random walks in the high-dimensional gene expression space or a projection of the high-dimensional vector field onto a low-dimensional latent space representation of the data.

Transition matrices induce random walks modeling the evolution of individual cells. Given a cell $j$, we successively sample its future state $k$ under the given transition matrix. Starting cells for random walks can be sampled either at random or from a user-defined early cell cluster. We terminate random walks when a predefined maximum number of steps has been performed or when a predefined set of terminal cells has been reached. By studying multiple random walks, the expected dynamics are revealed. Random walks, including their start and final cells, can then be visualized in a low-dimensional representation of the data. Within our framework, random walks are computed efficiently via a parallel implementation.

Previously, the most popular approach for visualizing RNA velocity has been the projection of the high-dimensional vector field onto a low-dimensional latent space representation[7]. With CellRank 2, we generalize this concept to any kernel based on a $k$-nearest-neighbor graph, that is, the PseudotimeKernel, CytoTRACEKernel and VelocityKernel. The projection for a given cell is calculated as follows: Consider a transition matrix $T$, cell $j$ with neighborhood $N(j)$ and $k_j$ neighbors and latent representation $z_j$. The projected velocity $v_j$ is then given by

$$v_j = \sum_{n \in N(j)} \left(T_{jn} - \frac{1}{k_j}\right)(z_n - z_j).$$

While we provide the option to visualize the projected velocity stream in low dimensions for specific kernels, we caution against the analysis thereof. Previous work[7,72,73] highlighted how the projected velocity stream is sensitive to many parameters, including the gene set, the embedding technique and more (Supplementary Note 1). Instead, we encourage visualizing cellular dynamics through random walks, sampled independently of the embedding, or through initial and terminal states, fate probabilities and other quantities inferred in high dimensions through our estimator modules.

**A conceptual overview of estimators in CellRank 2.** Based on transition matrices provided by kernels, we enable data-driven knowledge discovery. To this end, estimators first identify initial, intermediate and terminal states using the precomputed transition matrices. States are identified using concepts and results from the rich theory of Markov chains. Following this, we enable visualizing trajectory-specific gene expression trends and cascades of gene activation[14], clustering expression trends[14] or arranging cells in a circular embedding[14,74] to summarize fate probabilities. We provide the necessary tools for each step of the downstream analysis as part of CellRank 2.

*The generalized Perron cluster cluster analysis estimator.* As in our previous work, we compute macrostates and classify them as initial, intermediate and terminal by coarse-graining the cell–cell transition matrix. This approach is based on generalized Perron cluster cluster analysis[19,20] (GPCCA), a method initially developed to study conformational protein dynamics.

**Performance improvements of CellRank 2.** *Faster computation of fate probabilities.* After estimating cell–cell transition probabilities through a kernel and identifying terminal states through an estimator, we assess cellular fate toward these terminal states. For each cell, we quantify its fate probability, that is, how likely it is to differentiate into one of the terminal states. Given our Markov-chain-based framework, fate probabilities can be computed in closed form using absorption probabilities; however, calculating absorption probabilities directly scales cubically in the number of cells. To overcome this computational burden, in our previous work, we reformulated the underlying problem as a set of linear systems. These linear systems are then solved in parallel using a sparsity-optimized iterative algorithm[75]; this reformulation scales near-linearly[14].

Even though our previously proposed reformulation for computing absorption probabilities achieved a significant increase in performance compared to a naive implementation, we still encountered increased runtimes when analyzing larger datasets (Extended Data Fig. 2a). To reduce the runtime further, we devised an alternative but equivalent approach: Given a terminal state, we previously identified $n_f$ representative cells, computed absorption probabilities toward them, and aggregated them across the $n_f$ representative cells to assign a single, lineage-specific probability. In CellRank 2, we first combine the $n_f$ representative cells into a single pseudo-state and compute absorption probabilities toward it instead. While the corresponding results are mathematically equivalent, ignoring parallelization, this new approach is $n_f$ times faster. Therefore, with $n_f = 30$ by default, our improved implementation results in a 30-fold speed-up.

**Extensibility of CellRank 2.** While we already provide multiple kernels tailored to different data modalities, current and future technologies provide additional sources of information. Concrete examples include spatially resolved time-course studies[59–61] and genetic lineage-tracing data[55–58], previously already integrated in the CellRank 2 ecosystem[23,24]. Our modular interface makes CellRank 2 easily extensible toward (1) alternative single-cell data modalities by including new kernels and (2) alternative trajectory descriptions generating different hypotheses through new estimators.

## The PseudotimeKernel: incorporating previous knowledge on differentiation

Aligning cells along a continuous pseudotime mimicking the underlying differentiation process has been studied in many use cases. In particular, a pseudotime can be computed for systems where a single, known initial cellular state develops unidirectionally into a set of unknown terminal states. Based on the assigned pseudotime values, we quantify transition probabilities between cells using the PseudotimeKernel.

Given a similarity-based nearest-neighbor graph with a corresponding adjacency matrix $\tilde{c}$, the PseudotimeKernel biases graph edges toward increasing pseudotime: consider a reference cell $j$, one of its neighbors $k$, the corresponding edge weight $\tilde{c}_{jk}$ and the difference between their pseudotimes $\Delta t_{jk}$. To favor cellular transitions toward increasing pseudotime, the PseudotimeKernel downweighs graph edges pointing into the reference cell's pseudotemporal past while leaving the remaining edges unchanged. Edge weights are updated according to

$$c_{jk} = \tilde{c}_{jk} f(\Delta t_{jk}),$$

with a function $f$ implementing the thresholding scheme. In CellRank 2, we implement soft and hard thresholding. The soft scheme continuously downweighs edge weights according to

$$f(\Delta t) = \begin{cases} \frac{2}{\sqrt[\nu]{1+e^{b\Delta t}}}, & \Delta t < 0 \\ 1, & \Delta t \geq 0. \end{cases}$$

By default, the parameters $b$ and $\nu$ are set to 10 and 0.5, respectively. This concept is similar to the scheme proposed by the TI method VIA[25]. In contrast to soft thresholds, hard thresholding follows a stricter policy inspired by Palantir[5], discarding most edges that point into the pseudotemporal past.

## The CytoTRACEKernel: inferring directionality from developmental potential

CytoTRACE assigns each cell in a given dataset a developmental potential[18]. Score values range from 0 to 1, with 0 and 1 identifying mature and naive cells, respectively. Inverting the score, thus, defines a pseudotime for developmental datasets. In CellRank 2, the CytoTRACEKernel computes the CytoTRACE score and constructs the corresponding pseudotime to calculate a transition matrix as described for the PseudotimeKernel.

**Adaptation of the CytoTRACE score.** When calculating the CytoTRACE score on larger datasets, we found the score construction either intractable due to long runtimes (40,000 to 80,000 cells) or failed to compute the score at all (more than 80,000 cells) (Extended Data Fig. 2b). Thus, to ensure computational efficiency when reconstructing the CytoTRACE score for larger datasets, we sought an alternative, computationally efficient and numerically highly correlated approach.

Conceptually, CytoTRACE proposes that the number of expressed genes decreases with cellular maturity. This assumption is biologically motivated by less-developed cells regulating their chromatin less tightly[18]. The computation of the CytoTRACE score $c$ with CellRank 2 is composed of three main steps (Extended Data Fig. 4a). Consider the gene expression matrix $X$ and the smoothed gene expression matrix $X^{(smoothed)}$ found by nearest-neighbor smoothing as implemented in scVelo[8] or MAGIC[76]. For each cell $j$, we compute the number of genes it expresses (GEC $\in \mathbb{N}^{n_c}$), that is

$$\mathrm{GEC}_j = \sum_{k=1}^{n_g} \mathbb{1}(X_{jk} > 0),$$

with indicator function $\mathbb{1}(\cdot)$. The indicator function equates to one if its argument holds true and zero otherwise. Next, for each gene, we compute its Pearson correlation with GEC, select the top $L$ genes (default 200) and subset $X^{(smoothed)}$ to the identified $L$ genes. Finally, we mean-aggregate each cell's gene expression

$$\tilde{c}_j = \sum_{k=1}^{L} \tilde{X}_{jk},$$

with $\tilde{X} \in \mathbb{R}^{n_c \times L}$ denoting the subsetted, smoothed gene expression matrix $X^{(smoothed)}$. The CytoTRACE score $c$ is then given by scaling $\tilde{c}$ to the unit interval

$$c_j = \frac{\tilde{c}_j - \min \tilde{c}}{\max \tilde{c}},$$

and the corresponding pseudotime $p_{cyt}$ by inverting $c$, that is

$$p_{cyt} = 1 - c.$$

**Comparison of the CytoTRACE score construction.** Considering the nearest-neighbor-smoothed gene expression matrix, instead of an alternative, computationally more costly imputation scheme, is the main difference between our adaptation and the original CytoTRACE proposal. To impute gene expression, the original implementation solves a non-negative least squares regression problem and simulates a diffusion process[18].

To confirm that our adapted scheme yields numerically similar results, we compared the CytoTRACE scores of the original and our approach to ground truth time or stage labels on six datasets previously used to validate CytoTRACE[18] (Extended Data Fig. 4b,c). The considered datasets are bone marrow[77] (using 10x and SmartSeq2), *C. elegans* embryogenesis[78] (subsetted to ciliated neurons, hypodermis and seam, or muscle and mesoderm) and zebrafish embryogenesis[79]. For each dataset, the original CytoTRACE study derived ground truth time labels using either embryo time, stages (*C. elegans* and zebrafish embryogenesis) or a manual assignment (bone marrow). The concordance of each approach with ground truth was confirmed by calculating the Spearman rank correlation between the CytoTRACE score and ground truth time or stage labels.

## The RealTimeKernel: resolving non-equilibria systems through time-series data

Commonly used single-cell sequencing protocols are destructive by design and offer, thus, only a discrete temporal resolution. Recent advances allow reconstructing transcriptomic changes across experimental time points using OT[15]; however, these approaches focus only on inter-time-point information; conversely, the RealTimeKernel incorporates both inter- and intra-time-point transitions to draw a more complete picture of cellular dynamics.

To quantify inter-time-point transitions, the RealTimeKernel relies on WOT[15]. For each tuple of consecutive time points $t_j$ and $t_{j+1}$, WOT identifies a transport map $\pi_{t_j,t_{j+1}}$, assigning each cell at time $t_j$ its likely future state at time $t_{j+1}$. In addition, we rely on transcriptomic similarity to study transcriptomic change within a single time point $t_j$. We combine WOT-based inter-time point transport maps $\pi_{t_j,t_{j+1}}$ with similarity-based intra-time-point transition matrices $\bar{T}_{t_j,t_j}$ in a global transition matrix $T$ which contains cells from all time points. In the global transition matrix $T$, we place WOT-computed transport maps on the first off-diagonal, modeling transitions between subsequent time points, and similarity-based transition matrices on the diagonal, modeling transitions within each time point (Fig. 4a). We normalize each row to sum to one, giving rise to a Markov chain description of the system.

**Thresholding transport maps for scalability.** In the single-cell domain, most OT-based approaches, including WOT, rely on entropic regularization[36] to speed up the computation of transport maps; however, entropic regularization leads to dense transport maps $\pi_{t_j,t_{j+1}}$, rendering downstream computations based on the RealTimeKernel extremely expensive for larger datasets (Extended Data Fig. 6c); most of the entries found in $\pi_{t_j,t_{j+1}}$ are extremely small, though. As a result, these entries contribute only marginally to the observed dynamics (Extended Data Fig. 6d).

To ensure fast RealTimeKernel-based computations, we devised an adaptive thresholding scheme resulting in sparse transition matrices. Transition probabilities falling below a certain threshold are set to zero, all others are kept unchanged. Per default, we identify the smallest threshold $\tau$ that does not remove all transitions for any cell, that is

$$\tau = \min_{j,k \in \{1,\dots,n_c\}} T_{jk}, \quad \text{s.t.} \, \forall j \in \{1,\dots,n_c\} \sum_{k \in \{1,\dots,n_c\}} \mathbb{1}(\tau \geq T_{jk}) \geq 1,$$

with indicator function $\mathbb{1}(\cdot)$. Alternatively, the same heuristic can be applied for each time point independently or a user-defined threshold may be used. Following thresholding, we re-normalize the transition matrix such that rows sum to one again.

To verify that thresholding the transition matrix does not alter biological findings, we compared fate probabilities derived from the original and the thresholded transition matrix on a dataset of MEF reprogramming[15]. For each terminal state, we computed the Pearson correlation between fate probabilities estimated by each approach (Extended Data Fig. 6d).

## Estimating cellular fate from time-resolved single-cell RNA sequencing data

Traditional single-cell sequencing protocols include cell lysis and are, thus, destructive by nature. Consequently, the transcriptome can only be measured once, resulting in snapshot data. Recently, metabolic-labeling approaches have been extended to single-cell resolution, providing an opportunity to overcome this challenge by measuring newly synthesized mRNA in a given time window[80]. To label transcripts, current protocols rely on the nucleoside analogs 4-thiouridine (4sU; scSLAM-seq[12], sci-fate[10], NASC-seq[81], scNT-seq[11], Well-TEMP-seq[82] and others[83]) or 5-ethynyl-uridine (5EU; scEU-seq[9], spinDrop[65], TEMPOmap[13] and others[66]).

Our study considers two types of labeling experiments: pulse and chase[9]. Pulse experiments consist of labeling $n$ cell cultures, starting at times $t_j, j \in \{1,\dots,n\}$. Conversely, in chase experiments, cells are exposed to nucleoside analogs for long enough (for example, more than 24 h), resulting in only labeled transcripts. Following, these labeled transcripts are washed out, starting at times $t_j$. Similar to the pulse experiment, chase experiments include, in general, washing out at $n$ different times. Finally, in both types of experiments, all cells are sequenced at a time $t_f$, naturally defining the labeling time (or duration) by $\tau_l^{(j)} = t_f - t_j$.

Pulse and chase experiments allow measuring the production of mRNA. Here, we estimate cell-specific transcription and degradation rates, similar to a previous proposal in the scEU-seq study[9]. Specifically, for a particular gene, we assume mRNA levels $r$ to evolve according to

$$\dot{r} = \alpha - \gamma r,$$

with transcription rate $\alpha$ and degradation rate $\gamma$. The corresponding solution is given by

$$r(t) = r_0 e^{-\gamma t} + \frac{\alpha}{\gamma}\left(1 - e^{-\gamma t}\right).$$

Note that here, we assume gene-specific models, that is, gene–gene interactions are neglected. In the following, we will identify mRNA

measurements from pulse and chase experiments by the superscripts (p) and (c), respectively.

**Pulse experiments.** Pulse experiments study the production of labeled RNA. As labeling starts at $t_k$, and no labeled transcripts exist before, the abundance of labeled mRNA $r_l$ at times $t_k$ and $\tau_l^{(k)}$ is given by

$$r_l^{(p)}(t_k|\alpha,\gamma) = 0,$$

and

$$r_l^{(p)}\left(\tau_l^{(k)}|\alpha,\gamma\right) = \frac{\alpha}{\gamma}\left(1 - e^{-\gamma\tau_l^{(k)}}\right).$$

## Chase experiments

In chase experiments, mRNA degradation is studied by washing out labeled transcripts. Thus, labeled mRNA $r_l^{(c)}$ at time $\tau_l^{(k)}$ follows

$$r_l^{(c)}\left(\tau_l^{(k)}|\alpha,\gamma,r_0\right) = r_0 - \frac{\alpha}{\gamma}\left(1 - e^{-\gamma\tau_l^{(k)}}\right),$$

where $r_0$ corresponds to the mRNA level when starting to wash out labeled transcripts. Before washing out labeled mRNA, no unlabeled transcripts are present, and thus, their abundance at time $\tau_l^{(k)}$ is modeled as

$$r_u^{(c)}\left(\tau_l^{(k)}|\alpha,\gamma\right) = \frac{\alpha}{\gamma}\left(1 - e^{-\gamma\tau_l^{(k)}}\right).$$

**Parameter inference.** Considering measurements from both chase and pulse experiments, we denote the respective set of cells by $C$ and $P$. To estimate cell $j$ and gene $g$ specific model parameters $\alpha^{(j,g)}$, $\gamma^{(j,g)}$, and $r_0^{(j,g)}$, we proceed as follows:

1. Consider cell $j$ and its principal component analysis (PCA) representation $z_j^{(PCA)}$. For each labeling duration $k$, we determine the distance in PCA space between the reference cell $j$ to each cell with labeling duration $\tau_l^{(k)}$. For each gene $g$, we then identify the 20 nearest cells with non-trivial expression in $g$. These cells, as well as all closer neighbors (with zero counts), define the set $N_g^{(k)}$, which we consider for parameter inference.
2. To estimate model parameters, we minimize the quadratic loss $\ell$ defined as

$$\ell\left(r_0^{(j,g)},\alpha^{(j,g)},\gamma^{(j,g)}\right) = \sum_k \sum_{j \in N_g^{(k)}} \left[r_{l,j}\left(\tau_l^{(k)}\right) - \mathbb{1}(j \in C) r_l^{(c)}\left(\tau_l^{(k)}|\alpha,\gamma,r_0\right)\right.$$
$$\left. - \mathbb{1}(j \in P) r_l^{(p)}\left(\tau_l^{(k)}|\alpha,\gamma\right)\right]^2.$$

Here, $\mathbb{1}(x \in X)$ denotes the characteristic function equaling to 1 if $x \in X$, and 0 otherwise. We note that estimating the parameters of a pulse (chase) experiment requires at least two (three) labeling durations.

Our approach differs from the scEU-seq study[9] mainly in two ways. First, we base our analysis on total RNA, not spliced RNA. We reasoned that this approach circumvents limitations of identifying unspliced and spliced counts. Second, we infer rates for all genes and not only those changing substantially during development.

**Method comparison.** To benchmark the performance of different approaches, we identified and ranked potential drivers of every lineage using each approach. We compared this ranking to a curated list of known lineage markers and regulators. If the literature-based gene set were complete, an optimal method would rank the corresponding genes highest. Consequently, for each method, we quantified its

performance as follows. First, consider a lineage, a set of known drivers $D$ and a method $m$. Further, denote the set of genes by $G$, and for $g \in G$, identify its assigned rank by a superscript, for example, $g^{(j)}$ for the $j$th ranked gene $g$. Next, for each threshold $N \in \mathbb{N}$ and $N \leq |G|$, we computed how many known markers/regulators were ranked among the top $N$ genes with

$$\varphi^{(m)}(N) = \left|\left\{ g^{(j)} \mid j \leq N \wedge g^{(j)} \in D \right\}\right|.$$

Thus, we call an assignment optimal when

$$\varphi^{(\text{opt})}(N) = \begin{cases} \varphi^{(m)}(N) + 1, & N < |D| \\ |D|, & \text{otherwise}. \end{cases}$$

Next, for each method $m$, we computed the area under the curve, $\text{AUC}(m)$ of $\varphi^{(m)}$, that is

$$\text{AUC}(m) = \sum_{N=1}^{|G|} \varphi^{(m)}(N)$$

and its relative area under the curve, $\text{AUC}_{\text{rel}}(m)$ as

$$\text{AUC}_{\text{rel}}(m) = \frac{\text{AUC}(m)}{\text{AUC}^*},$$

with

$$\text{AUC}^* = \frac{|D|(|D|+1)}{2} + (|G| - |D|)|D|,$$

that is, the area under the curve of an optimal assignment.

### Datasets

Unless stated otherwise, all functions were run with default parameters. We ran our analyses in Python, relying on the standard single-cell biology tools Scanpy[84] and AnnData[85]; we specify other relevant packages where applicable. For Scanpy-based workflows[84], we computed PCA embeddings, neighbor graphs and UMAP embeddings[86] with the scanpy.tl.pca, scanpy.pp.neighbors and scanpy.tl.umap functions, respectively.

For analyses based on CellRank 2 kernels, the kernel method compute_transition_matrix computed transition probabilities, and the kernel method cbc the CBC score. We used the GPCCA estimator functions to compute macrostates (compute_macrostates)[20] and the TSI score (tsi), define terminal states (set_terminal_states), compute fate probabilities (compute_fate_probabilities) and identify lineage-correlated genes (compute_lineage_drivers). To order the putative regulators according to their peak expression in pseudotime, we first fitted generalized additive models to describe gene expression change over pseudotime with cellrank.models.GAM. Following this, we visualized the putative cascade of regulation with the cellrank.pl.heatmap function.

**Human hematopoiesis.** All analyses were conducted on the dataset preprocessed by the original study[26], subsetted to the normoblast, dendritic and monocyte lineages according to the provided cell type annotation ('HSC', 'MK/E progenitors', 'Proerythroblast', 'Erythroblast', 'Normoblast', 'cDC2', 'pDC', 'G/M prog' and 'CD14+ Mono').

*Pseudotime-based analysis.* After subsetting the data, we computed the nearest-neighbor graph on the precomputed MultiVI[84,87] latent space and the UMAP embedding with Scanpy. Following this, we computed 15 diffusion components[88] (scanpy.tl.diffmap) to then assign diffusion pseudotime values using Scanpy's dpt[1,88] function with n_dcs=6. We identified the root cell as the hematopoietic stem cell with the largest fifth diffusion component.

We computed the transition matrix with CellRank 2's PseudotimeKernel and thresholding_scheme='soft' and computed six macrostates[20]. We defined the terminal states 'pDC', 'CD14+ Mono', 'Normoblast' and 'cDC2' that corresponded to the four macrostates with the largest macrostate purity. After quantifying fate probabilities, we identified putative pDC lineage drivers with our correlation-based procedure, restricted to the hematopoietic stem cell (HSC) and pDC clusters (lineages=['pDC'] and clusters=['HSC', 'pDC']). We quantified the corresponding gene trends in the same way as described in our previous work[14].

*RNA velocity-based analysis.* To infer RNA velocity, we generally followed the instructions provided by scVelo's[8] tutorials. First, we filtered for genes expressed in at least 20 cells in both unspliced and spliced counts with scVelo's scvelo.pp.filter_genes function and normalized counts with scvelo.pp.normalize_per_cell. The neighbor graph was again computed on the MultiVI latent space, followed by count imputation through first-order moments with scVelo's scvelo.pp.moments function. We then inferred RNA velocity with the scvelo.tl.recover_dymanics function.

To quantify cellular fate, we computed transition matrices with the VelocityKernel and ConnectivityKernel and combined them with 0.8 and 0.2 weight, respectively, as proposed by the CellRank 1 workflow. We computed macrostates and fate probabilities using the GPCCA estimator as described for the pseudotime-based analysis. As the RNA-velocity-based analysis did not identify the cDC cluster as a macrostate (Extended Data Fig. 3b,f), we computed three macrostates corresponding to the terminal states normoblasts, monocytes and pDCs.

**Embryoid body development.** *Data preprocessing.* We followed Scanpy's workflow to process the raw count matrix. As a first step, we filtered out genes expressed in fewer than ten cells (scanpy.pp.filter_genes with min_cells=10). Following, we removed cells with more than 17,500 counts, cells for which more than 15% of counts originate from mitochondrial genes and cells expressing more than 3,500 genes. Following, we size normalized cells to 10,000 (scanpy.pp.normalize_total with total_sum=1e4), applied a log1p-transformation (scanpy.pp.log1p) and annotated highly variable genes with scanpy.pp.highly_variable_genes. We based all further analyses on these highly variable genes and the marker genes identified by the study introducing the embryoid body development dataset[32]. The neighbor graph was computed for 30 neighbors using 30 principal components (PCs).

*CytoTRACEKernel analysis.* To compute the CytoTRACE score[18], we first imputed the normalized count matrix by first-order moments with scVelo's scvelo.pp.moments function; the score itself was calculated with the compute_cytotrace method of the CytoTRACEKernel. We computed the transition matrix with the soft thresholding scheme (thresholding_scheme='soft') and nu=5. Putative drivers of the endoderm lineage were identified by focusing on the stem cell and endoderm clusters (lineages=['EN-1'] and clusters=['ESC']).

*Pseudotime construction.* To compute DPT[1], we calculated diffusion components (scanpy.tl.diffmap) and identified the putative root cell as the minimum in the first diffusion component. We then assigned DPT values using Scanpy's dpt function.

For the Palantir pseudotime[5], we used the corresponding Python package and followed the steps outlined in its documentation. As a first step, we computed the first five diffusion components with palantir.utils.run_diffusion_maps with n_components=5. Following this, we identified the multi-scale space of the data (palantir.utils.determine_multiscale_space) and imputed the data using MAGIC[76] (palantir.utils.run_magic_imputation). Finally, we computed the Palantir pseudotime via palantir.core.run_palantir using the same root cell as for our DPT analysis, and num_waypoints=500.

**Mouse embryonic fibroblast reprogramming.** *Data preprocessing.* For analyzing the dataset of MEF reprogramming toward induced pluripotent stem cells[15], we subsetted to the serum condition and added the category 'MEF/other' to the cell set annotations. Then, we computed the PCA embedding and nearest-neighbor graph.

*WOT-based analysis.* To construct transport maps, we used the wot package[15] and followed the provided tutorials. First, we instantiated an OT model (wot.ot.OTModel with day_field='day') and computed the transport maps next (compute_all_transport_maps). We defined the target cell sets based on the provided cell type annotation and quantified WOT-based fates toward the last experimental time point through the OT model's fates function with at_time=18.

*RealTimeKernel-based analysis.* For our RealTimeKernel-based analysis, we relied on the transport maps computed with wot. When constructing the transition matrix, we considered within-time-point transitions for every experimental time point and weighed them by 0.2 (self_transitions='all', conn_weight=0.2).

To construct the real-time-informed pseudotime, we symmetrized the global transition matrix and row-normalized it. The symmetrized matrix defined the _transitions_sym attribute of Scanpy's DPT class. Following, we computed diffusion components with the DPT class' compute_eigen function. The root cell for DPT was identified as an extremum of the most immature cell state within the first experimental time point in diffusion space. Here, we selected the maximum in the first diffusion component. Finally, we computed DPT itself with scanpy.tl.dpt.

We computed fate probabilities toward the four terminal states according to our canonical pipeline (identification of four macrostates followed by fate quantification).

*Pseudotime construction.* To assign each observation its DPT[1] value, irrespective of experimental time points, we computed diffusion maps (scanpy.tl.diffmap) and identified the root cell as the maximum value in the first diffusion component. Then, we computed DPT with scanpy.tl.dpt.

For constructing the Palantir pseudotime[5], irrespective of experimental time points, we followed the same steps as described for the embryoid body development data.

**Pharyngeal endoderm development.** *Data preprocessing.* The pharyngeal endoderm development dataset provided by the original study[37] had already been filtered for high-quality cells and genes. Consequently, we directly quantified highly expressed genes using Scanpy's highly_variable_genes function. Then, we computed the PCA embedding and nearest-neighbor graph based on 30 PCs and 30 neighbors (n_pcs=30, n_neighbors=30).

*RealTimeKernel-based analysis.* To study the pharyngeal endoderm development dataset with the RealTimeKernel, we followed the WOT tutorials to compute transport maps. First, we instantiated an OT model (wot.ot.OTModel with day_field='day') and computed the transport maps next (compute_all_transport_maps). For the RealTimeKernel, we considered within-time-point transitions for every experimental time point and weighed them by 0.1 (self_transitions='all', conn_weight=0.1).

We estimated terminal states using the GPCCA estimator with default settings by computing 13 macrostates and selecting the known terminal clusters. After calculating fate probabilities, for each lineage, we identified lineage-correlated genes as candidate driver genes with GPCCA.compute_lineage_drivers by restricting the analysis to progenitors of the corresponding lineage and excluding cell cycle, mitochondrial, ribosomal and hemoglobin genes[89].

To study mTEC development, we subsetted to the early thymus, ultimobranchial body (UBB), parathyroid, cTEC and mTEC clusters and

processed the data as described for the entire dataset. We computed the UMAP embedding using Scanpy's umap function. To compute the transition matrix, we proceeded in the same manner as described for the entire dataset. For TSI, fate quantification, and driver analysis, we followed the standard CellRank 2 pipeline.

*WOT-based analysis.* To identify putative drivers of the mTEC lineage with WOT, we used the same transport maps as in our RealTimeKernel-based analysis. We defined the target cell sets based on the provided cell type annotation considering only observations from the last time point and computed the pullback distribution from the mTEC cluster at embryonic day (E) 12.5 to E10.5 cells as it consists of progenitor cells (pull_back). The sequence of ancestor distributions was quantified with the transport model's trajectories method. WOT identifies putative drivers of the mTEC lineage as genes differentially expressed in cells most fated toward the mTEC cluster. We used the wot.tmap.diff_exp function to construct the corresponding gene ranking.

*Classical differential expression analysis.* As an alternative means to identify putative drivers of the mTEC lineage based on the fate probabilities assigned by our RealTimeKernel-based analysis, we defined two groups of cells within the general progenitor pool, those with mTEC fate probability greater than 0.5 and all other progenitor cells. We then identified differentially expressed genes between putative mTEC progenitors and all others with Scanpy's rank_genes_groups function.

**Intestinal organoids.** *Data preprocessing.* To preprocess the dataset of intestinal organoids, we first excluded dimethylsulfoxide control cells and cells labeled as tuft cells. Following, we removed genes with fewer than 50 counts, size normalized total and labeled counts, and identified the 2,000 most highly variable genes with scvelo.pp.filter_and_normalize. The neighbor graph was constructed based on 30 PCs and 30 neighbors. Finally, we computed first-order-smoothed labeled and total mRNA counts.

*Parameter estimation.* To estimate kinetic rate parameters, we made use of our new inference scheme for metabolic-labeling data implemented as part of the scVelo package. We first masked observations according to their labeling time with scvelo.inference.get_labeling_time_mask. Next, we computed pairwise distances between observations in PCA space and sorted observations in ascending order for each time point using scvelo.inference.get_obs_dist_argsort. This information allowed us to identify, for each cell and gene, how many neighbors to consider during parameter estimation to include 20 non-zero observations smoothed by first-order moments. This calculation was performed via scvelo.inference.get_n_neighbors. Finally, we estimated model parameters based on smoothed labeled counts with scvelo.inference.get_parameters.

*Labeling velocity-based analysis.* To quantify cell-specific fates, we first computed labeled velocities based on the estimated parameters. Then, we computed a transition matrix by combining the VelocityKernel and ConnectivityKernel with a 0.8 and 0.2 weight, respectively. We then inferred 12 macrostates and fate probabilities toward the known terminal states. Lineage-specific drivers were identified by restricting the correlation-based analysis to the corresponding terminal state and stem cell cluster. For putative driver gene ranking based on gene expression, we correlated fate probabilities with smoothed labeled counts.

*Dynamo-based analysis.* To analyze the intestinal organoid data with dynamo[16], we followed the tutorials provided in the documentation of the Python package. As a first step, this required us to compute the ratio of new to total RNA with dynamo.preprocessing.utils.calc_new_to_total_ratio followed by first-order moment imputation of total and new RNA using dynamo.tl.moments with our connectivity matrix and

group='time'. Dynamo's dynamo.tl.dynamics estimated the velocities with function arguments model='deterministic', tkey='time' and assumption_mRNA='ss'.

Following velocity estimation, we quantified fixed points by following dynamo's corresponding pipeline. First, we projected the high-dimensional velocity field of new RNA onto the UMAP embedding using dynamo.tl.cell_velocities with ekey='M_n' and vkey='velocity_N'. Fixed points were then identified by calling dynamo.tl.VectorField with basis='umap' and dynamo.vf.topography. As a final step, we identified all stable fixed points.

Given the stable fixed points of the system, we identified lineage-correlated genes regulating cell differentiation toward them using dynamo's least action path analysis. As a first step, this workflow required us to compute a UMAP embedding based on new RNA with dynamo.tl.reduceDimension and layer='X_new', followed by the projection of the velocity field onto the PCA space (dynamo.tl.cell_velocities with basis='pca') and learning a vector field function based on this projection (dynamo.tl.VectorField with basis='pca'). Next, we defined terminal states as the 30 nearest neighbors in UMAP space of each stable fixed point. For initial states, we computed the 30 nearest neighbors of unstable fixed points of the stem cell cluster. To compute the least action paths and account for uncertainty in initial and terminal state assignment, we randomly sampled ten pairs of initial and terminal cells and estimated the paths between them with dynamo.pd.least_action. Dynamo's dynamo.pd.GeneTrajectory class then identified genes associated with the emergence of a terminal state. The pairwise sampling of initial and terminal cells defined the confidence bands of dynamo's gene rankings shown in Fig. 5d.

*RNA velocity-based analysis*. Conventional RNA velocity was estimated with scVelo's dynamical model[8] by running scvelo.tl.recover_dynamics. To execute this function, we first preprocessed the raw data with scvelo.pp.filter_and_normalize to remove genes expressed in fewer than 50 cells (min_counts=50), size-normalizing spliced and unspliced counts and subsetting to the 2,000 most highly variable genes (n_top_genes=2000). Then, we computed the PCA embedding, calculated the neighbor graph with 30 PCs and 30 neighbors and smoothed unspliced and spliced counts by first-order moments (scvelo.pp.moments).

We combined the VelocityKernel and ConnectivityKernel weighted by 0.8 and 0.2, respectively, to estimate the cell–cell transition matrix. Next, we identified terminal states and corresponding fates and lineage-correlated gene rankings following the canonical CellRank 2 pipeline.

### Reporting summary

Further information on research design is available in the Nature Portfolio Reporting Summary linked to this article.

## Data availability

All data presented in this study are publicly available via the original publications; we provide additional access to each dataset, that is, the peripheral blood mononuclear cell, embryoid body development, MEF, pharyngeal endoderm development and intestinal organoid data, via a figshare collection at https://doi.org/10.6084/m9.figshare.c.6843633.v1 (ref. 90).

## Code availability

CellRank 2 is released under the BSD-3-Clause license, with code available at https://github.com/theislab/cellrank and deposited via Zenodo at https://zenodo.org/doi/10.5281/zenodo.10210196 (ref. 91). The inference of kinetic rates based on metabolic-labeling data is implemented as part of the scVelo package (https://github.com/theislab/scvelo). Code to reproduce the results in the paper can be found at https://github.com/theislab/cellrank2_reproducibility and deposited via Zenodo at https://doi.org/10.5281/zenodo.10809425 (ref. 92).

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

## Acknowledgements

We thank N. Battich for helpful discussions about scEU-seq data, R. Maehr and M. Lobo for insights into pharyngeal organ development and T. Nawy for helping to write the paper. We also thank I. Virshup and numerous researchers who provided feedback on our implementation on GitHub. Finally, we thank all members of the Theis and Treutlein laboratories for helpful discussions. This work was supported

by the BMBF-funded de.NBI Cloud within the German Network for Bioinformatics Infrastructure (de.NBI) (031A532B, 031A533A, 031A533B, 031A534A, 031A535A, 031A537A, 031A537B, 031A537C, 031A537D and 031A538A) and co-funded by the European Union (ERC, DeepCell, 101054957). M.L. acknowledges financial support from the Joachim Herz Foundation and through an EMBO Postdoctoral Fellowship. F.T. acknowledges support by Wellcome Leap as part of the ΔTissue Program. For all support via EU funding, the views and opinions expressed are those of the authors only and do not necessarily reflect those of the European Union or the European Research Council. Neither the European Union nor the granting authority can be held responsible for them.

## Author contributions

P.W. and M.L. contributed equally. P.W., M.L. and F.T. conceptualized the study. M.K. implemented CellRank 2 with contributions from P.W. and M.L. P.W. designed and implemented the inference scheme for metabolic-labeling data. P.W. performed analyses with input from M.L. P.W., M.L., F.T. and D.P. wrote the paper. All authors read and approved the final paper.

## Funding

## Competing interests

F.T. consults for Immunai, Singularity Bio, CytoReason, Cellarity and Curie Bio Operations and has ownership interest in Dermagnostix and Cellarity. D.P. is on the scientific advisory board of Insitro. The other authors declare no competing interests.

## Additional information

**Extended data** is available for this paper at https://doi.org/10.1038/s41592-024-02303-9.

**Correspondence and requests for materials** should be addressed to Fabian Theis.

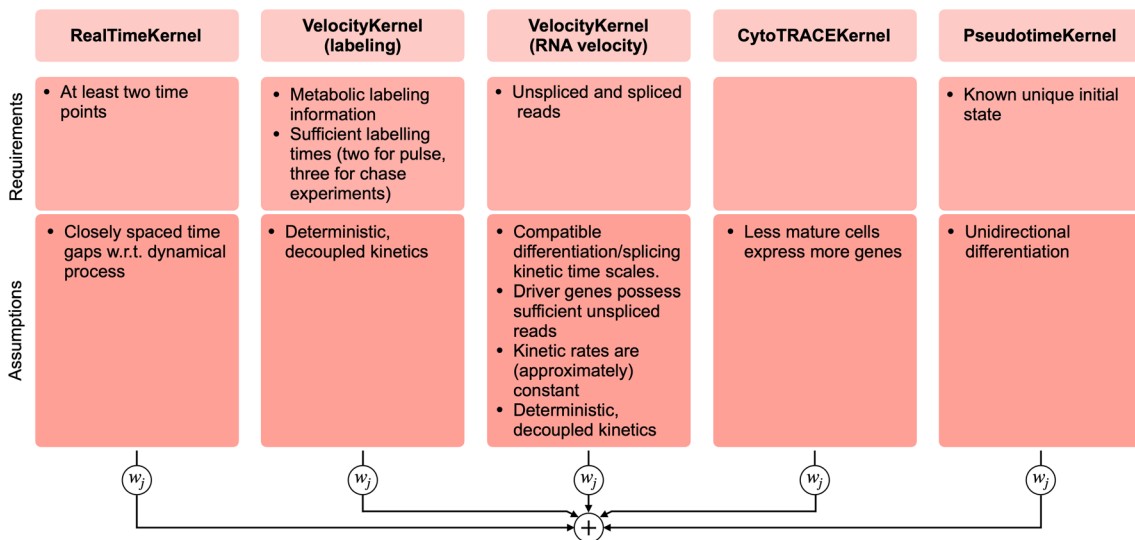

**Extended Data Fig. 1 | Guiding kernel choice in CellRank 2.** CellRank 2 implements various kernels suitable for different data modalities and experimental designs. The diagram may be used as a guide to identify the most suitable kernel. Note that assumptions change as methods evolve; for example, more recent inference schemes for RNA velocity account for non-constant kinetic rates[93].

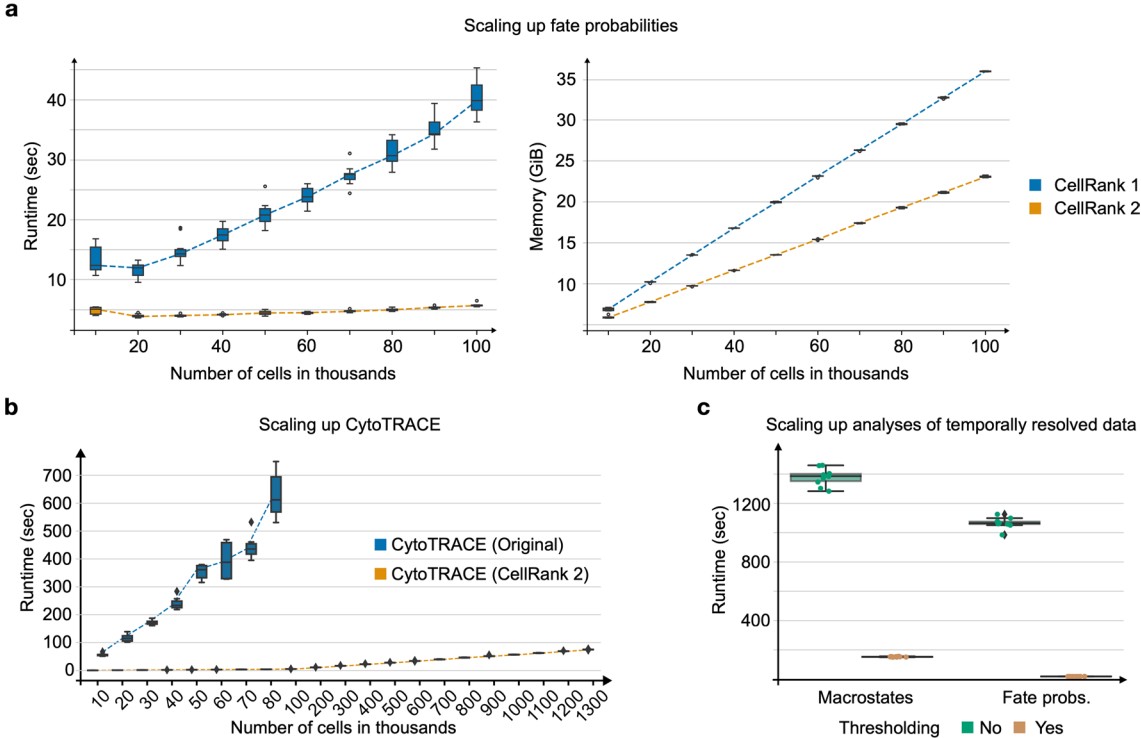

**Extended Data Fig. 2 | CellRank 2 scales to large cell numbers. a.** Runtime (left) and peak memory consumption (right) to compute fate probabilities with CellRank (orange) and CellRank 2 (blue). Both methods were run on subsets of a reprogramming dataset containing over 100, 000 cells[94]. Box plots indicate the median (center line), interquartile range (hinges), and whiskers at 1.5x interquartile range ($N$ = 10 runs each). **b.** The CellRank 2 adaptation of CytoTRACE scales to a mouse organogenesis atlas of 1.3 million cells[31], whereas CytoTRACE fails above 80, 000 cells. Box plots indicate the median (center line), interquartile range (hinges), and 1.5x interquartile range (whiskers) (50, 000 cells, original: $N$ = 6 runs; 60, 000 cells, original: $N$ = 8 runs; 80, 000 cells, original: $N$ = 9 runs; otherwise: $N$ = 10 runs). **c.** Runtime for calculating macrostates and fate probabilities using the RealTimeKernel with (brown) and without (green) thresholded transition matrix. Box plots indicate the median (center line), interquartile range (hinges), and 1.5x interquartile range (whiskers) ($N$ = 10 runs each).

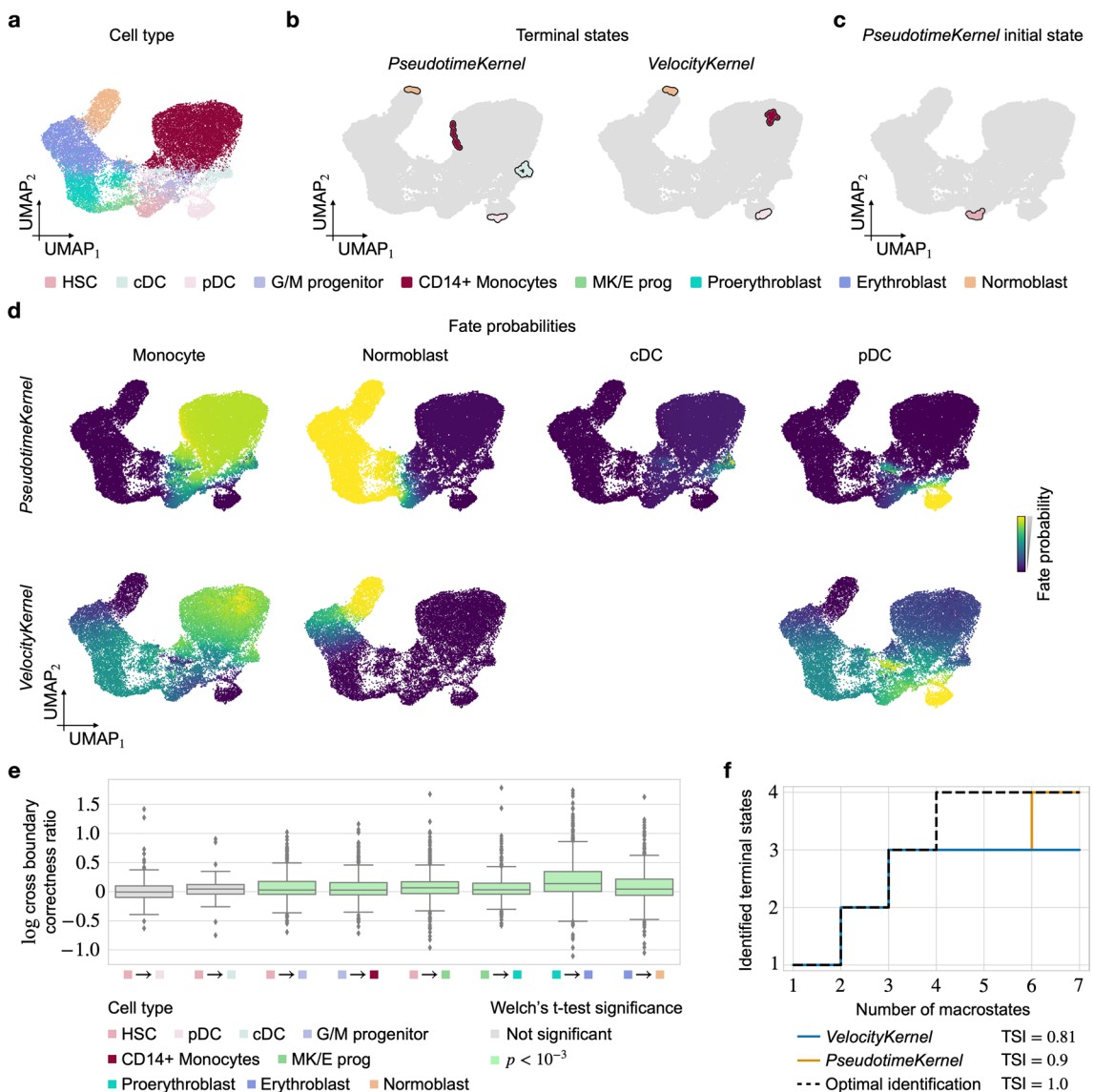

**Extended Data Fig. 3 | Performance of the VelocityKernel compared to the PseudotimeKernel. a.** UMAP embedding of entire hematopoiesis dataset[26]. Cell types are colored according to the original publication (HSC: hematopoietic stem cell, MK/E prog: megakaryocyte/erythrocyte progenitors, G/M progenitor: Granulocyte/Myeloid progenitor, pDC: plasmacytoid dendritic cell, cDC2: classical dendritic cells). **b.** Terminal states identified by the PseudotimeKernel (left) and VelocityKernel with RNA velocity (right) inferred using scVelo's *dynamical model*[8]. **c.** Initial state identified by the PseudotimeKernel. **d.** Fate probabilities towards each identified terminal state based on the PseudotimeKernel (top) and VelocityKernel (bottom). The VelocityKernel does not identify the cDC terminal state. **e.** Log-transformed ratio of cross-boundary correctness of cell type transitions of the PseudotimeKernel and VelocityKernel (HSC: hematopoietic stem cell, MK/E prog: megakaryocyte/erythrocyte progenitors, G/M progenitor: Granulocyte/Myeloid progenitor, pDC:

plasmacytoid dendritic cell, cDC2: classical dendritic cells). Values larger than zero correspond to the PseudotimeKernel outperforming the VelocityKernel; significance was tested using one-sided Welch's t-tests (Methods). Box plots indicate the median (center line), interquartile range (hinges), and 1.5x interquartile range (whiskers) (HSC to pDC: N=62 cells, $p = 0.12$; HSC to cDC: N=38 cells, $p = 0.11$; HSC to G/M progenitor: N=659 cells, $p = 1.48 \times 10^{-15}$; G/M progenitor to CD14+ monocytes: N=435 cells, $p = 2.51 \times 10^{-9}$; HSC to MK/E prog: N=489 cells, $p = 1.88 \times 10^{-13}$; MK/E prog to proerythroblast: N=513 cells, $p = 1.38 \times 10^{-9}$; proerythroblast to erythroblast: N=1052 cells, $p = 4.93 \times 10^{-75}$; erythroblast to normoblast: N=499 cells, $p = 1.19 \times 10^{-10}$). **f.** The number of identified terminal states is plotted against the number of macrostates specified. In the optimal scenario (dashed black), a new terminal state is identified for every added macrostate. The terminal state identification score (TSI) is defined by the area under a given curve relative to the optimal identification (Methods).

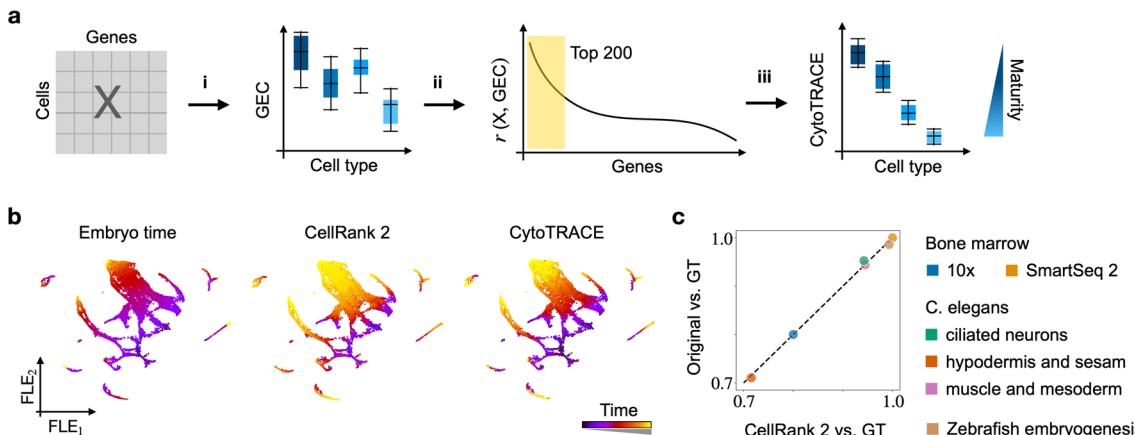

**Extended Data Fig. 4 | Developing the CytoTRACEKernel. a**. Similar to the CytoTRACE publication[18], we compute the CytoTRACE score by (**i**) calculating the number of genes expressed per cell (GEC), (**ii**) computing each gene's Pearson correlation with GEC, (**iii**) mean-aggregating imputed expression of the top 200 correlated genes (Methods). Box plots indicate the median (center line), interquartile range (hinges), and 1.5x interquartile range (whiskers); the shown box plots are schematics. **b**. Force-directed layout embedding (FLE) of 22, 370 Caenorhabditis (C.) elegans muscle and mesoderm cells undergoing embryogenesis, colored by estimated embryo time[78] (left; 130 – 830 minutes), CytoTRACE pseudotime computed using CellRank 2 (middle) and the original

implementation (right). **c**. Quantitative comparison of the two implementations of the CytoTRACE pseudotime on bone marrow[77] (using 10x and SmartSeq2), C. elegans embryogenesis[78] (subsetted to ciliated neurons, hypodermis and seam, and muscle and mesoderm), and zebrafish embryogenesis[79]. The x axis (y axis) displays Spearman's rank correlation between CellRank 2-CytoTRACE (original CytoTRACE) and ground-truth (GT) time labels. Ground-truth labels were derived from either embryo time or stages as in **b**. (C. elegans and zebrafish embryogenesis) or from manually assigned maturation labels from the original CytoTRACE study[18] (bone marrow).

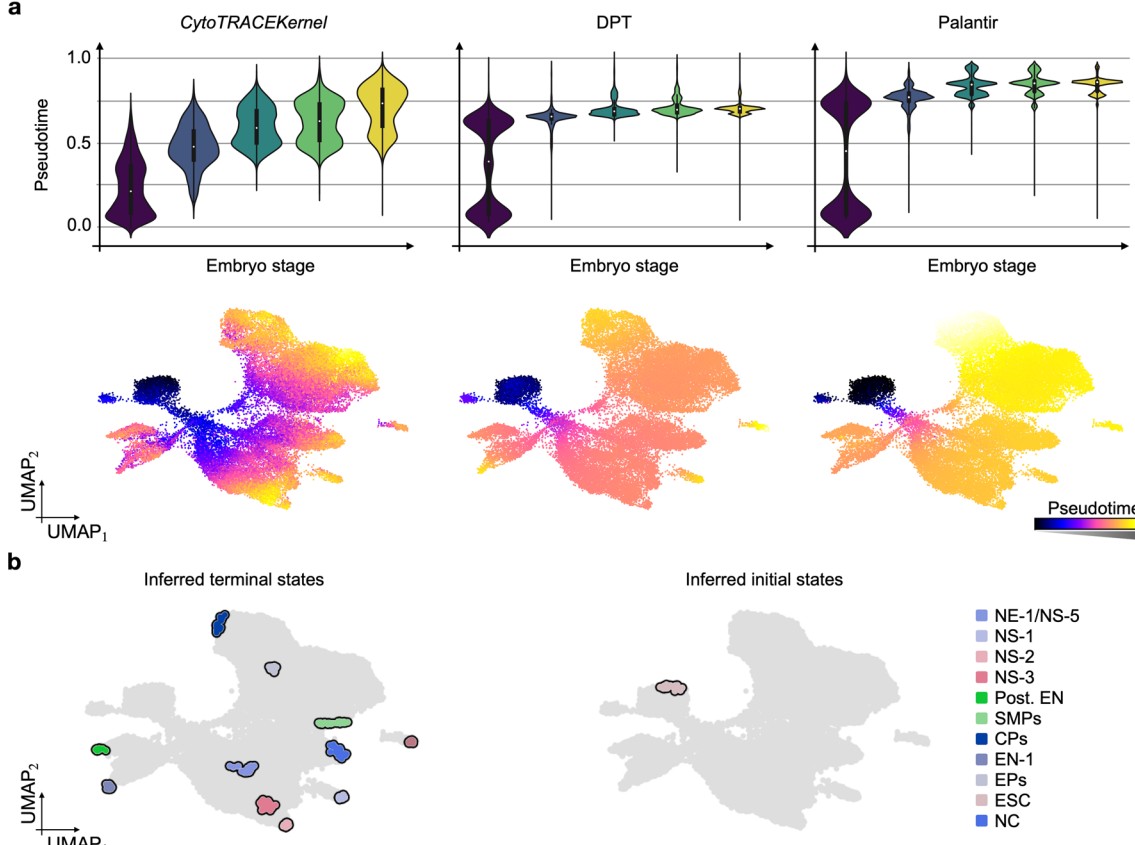

**Extended Data Fig. 5 | The CytoTRACEKernel recovers developmental progression, and terminal and initial states faithfully. a.** Distribution of pseudotimes from the CytoTRACEKernel (left), DPT (center), and Palantir (right), stratified by embryo stage (top row) and colored according to Fig. 3a. Box plots indicate the median (center line), interquartile range (hinges), and 1.5x interquartile range (whiskers) (E0-E3: $N$ = 4574 cells, E6-E9: $N$ = 7368 cells, E12-E15: $N$ = 6241 cells, E18-E21: $N$ = 6543 cells, E24-E27: $N$ = 6302 cells); UMAP embeddings (bottom) are colored by pseudotime. **b.** Terminal states (left) and initial state (right) inferred using the CytoTRACEKernel.

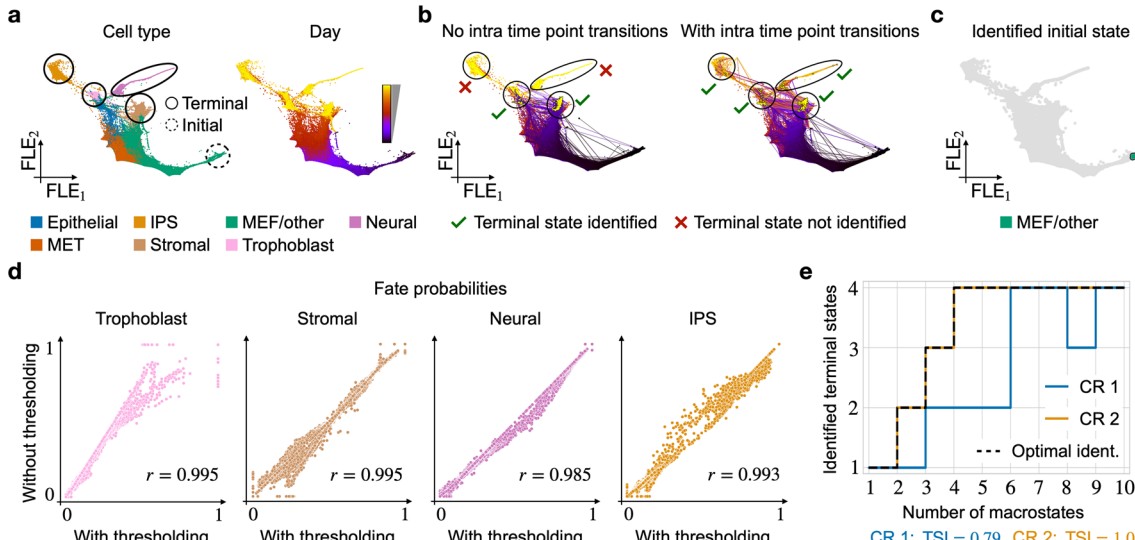

**Extended Data Fig. 6 | Developing the RealTimeKernel. a**. Force-directed layout embedding (FLE) of 165, 892 mouse embryonic fibroblasts (MEFs) reprogramming towards various endpoints during an 18-day time course[15], colored according to modified original annotations (IPS: induced pluripotent stem; left) or sequencing time points (right). Dotted (solid) circles indicate known initial (terminal) states. **b**. FLE showing simulated random walks from day 0 cells without (left; corresponds to WOT[15]) and with (right; corresponds to the RealTimeKernel) intra-time point transitions; black (yellow) dots denote a random walk's start (end); green ticks (red crosses) indicate known terminal states that are (are not) explored by random walks. **c**. Initial state identified by the RealTimeKernel. **d**. Evaluation of the effect of thresholding the transition matrix in the RealTimeKernel. Each dot corresponds to one cell's fate probability towards one of four terminal states, computed with thresholding (x axis) and without thresholding (y axis). The color coding is in agreement with **a.e**. Identification of terminal states with increasing number of macrostates using the VelocityKernel (CR 1; blue) or RealTimeKernel (CR 2; orange).

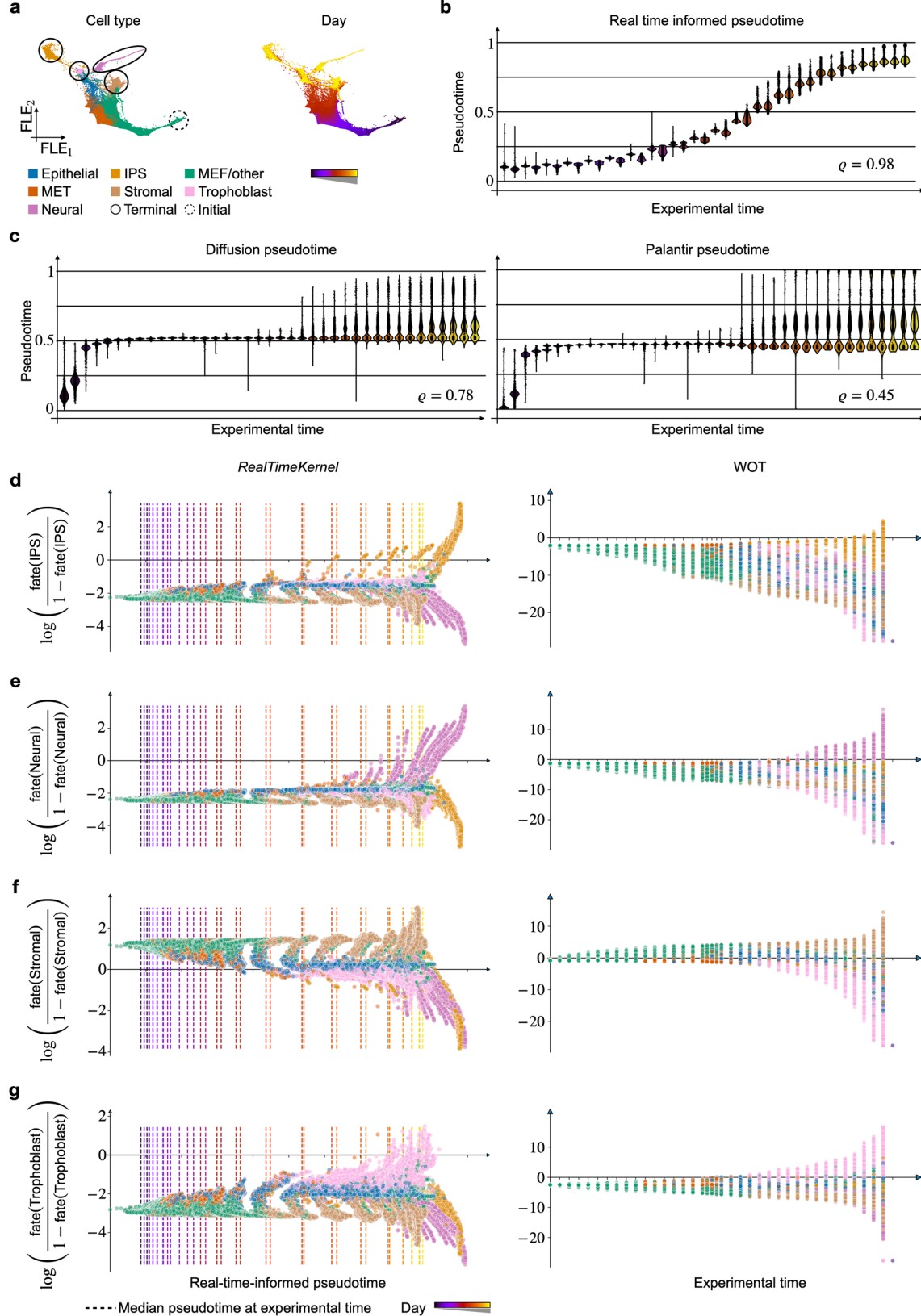

**Extended Data Fig. 7 | See next page for caption.**

**Extended Data Fig. 7 | The RealTimeKernel models fate decisions on a continuous domain. a.** Force-directed layout embedding (FLE) of 165, 892 mouse embryonic fibroblasts (MEFs) reprogramming towards various endpoints during an 18-day time course[15], colored according to modified original annotations (IPS: induced pluripotent stem; left) or sequencing time points (right). Dotted (solid) circles indicate known initial (terminal) states. **b,c.** Violin plots showing pseudotime distribution for each of 36 experimental time points, using CellRank 2's real-time-informed pseudotime (**b.**), DPT[1] (**c.**, left), or Palantir[5] (**c.**, right) (Methods). Box plots indicate the median (center line), interquartile range (hinges), and whiskers at 1.5x interquartile range. **c.** DPT[1] (center), or Palantir[5] (right) (Day 0: $N$ = 4556 cells, Day 0.5: $N$ = 3449 cells, Day 1: $N$ = 3648 cells, Day 1.5: $N$ = 1956 cells, Day 2: $N$ = 6981 cells, Day 2.5: $N$ = 6734 cells, Day 3: $N$ = 6777 cells, Day 3.5: $N$ = 7355 cells, Day 4: $N$ = 8962 cells, Day 4.5: $N$ = 7127 cells,

Day 5: $N$ = 7227 cells, Day 5.5: $N$ = 6550 cells, Day 6: $N$ = 8422 cells, Day 6.5: $N$ = 3111 cells, Day 7: $N$ = 6507 cells, Day 7.5: $N$ = 5061 cells, Day 8: $N$ = 3815 cells, Day 8.25: $N$ = 3829 cells, Day 8.5: $N$ = 3573 cells, Day 8.75: $N$ = 3088 cells, Day 9: $N$ = 2982 cells, Day 9.5: $N$ = 2266 cells, Day 10: $N$ = 2051 cells, Day 10.5: $N$ = 1941 cells, Day 11: $N$ = 2238 cells, Day 11.5: $N$ = 2164 cells, Day 12: $N$ = 2429 cells, Day 12.5: $N$ = 2253 cells, Day 13: $N$ = 2145 cells, Day 13.5: $N$ = 2034 cells, Day 14: $N$ = 3758 cells, Day 14.5: $N$ = 2723 cells, Day 15: $N$ = 3717 cells, Day 15.5: $N$ = 4851 cells, Day 16: $N$ = 3422 cells, Day 16.5: $N$ = 4645 cells, Day 17: $N$ = 3678 cells, Day 17.5: $N$ = 4068 cells, Day 18: $N$ = 3799 cells). **d-g.** Cell-specific fate change over pseudotime (RealTimeKernel, left) or experimental time (WOT, right) for the IPS (**d.**), neural (**e.**), stromal (**f.**), and trophoblast lineage (**g.**). For the RealTimeKernel, dashed vertical lines denote the mean pseudotime over all cells from a given experimental time point, recapitulating the correct ordering from **b.**

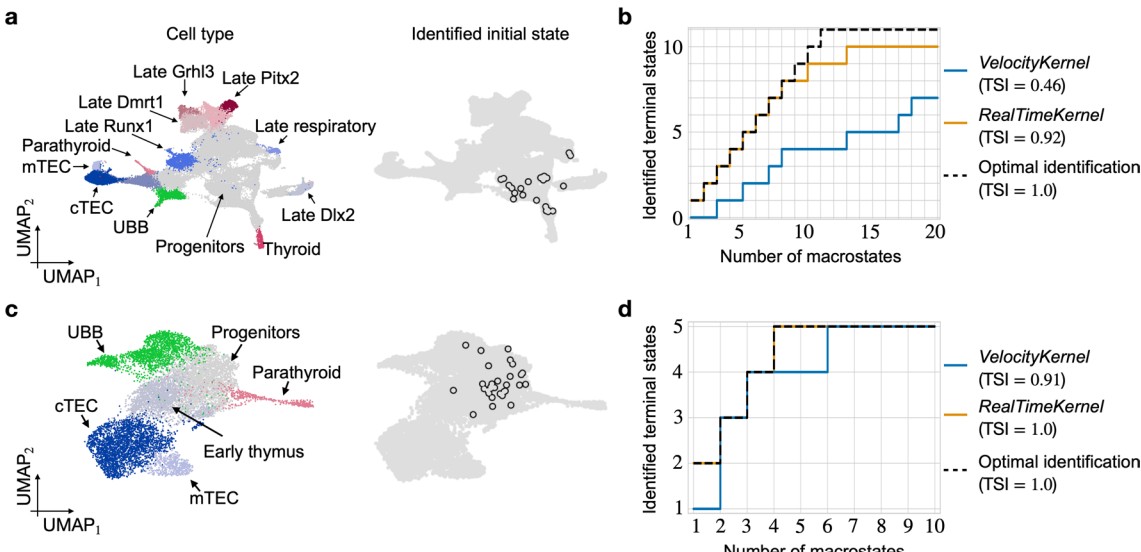

**Extended Data Fig. 8 | Initial and terminal state identification of pharyngeal endoderm development. a**. UMAP embedding of pharyngeal endoderm development dataset (left) and identified initial state population (right). Cells are colored according to cell types identified in the original study[37]. **b**. Identified number of terminal states with increasing number of macrostates using the full pharyngeal endoderm development dataset. **c**. UMAP embedding of subsetted pharyngeal endoderm dataset (right) and identified initial state (right); cells are colored according to the original study[37]. **d**. Same as **b**. but for the subsetted case.

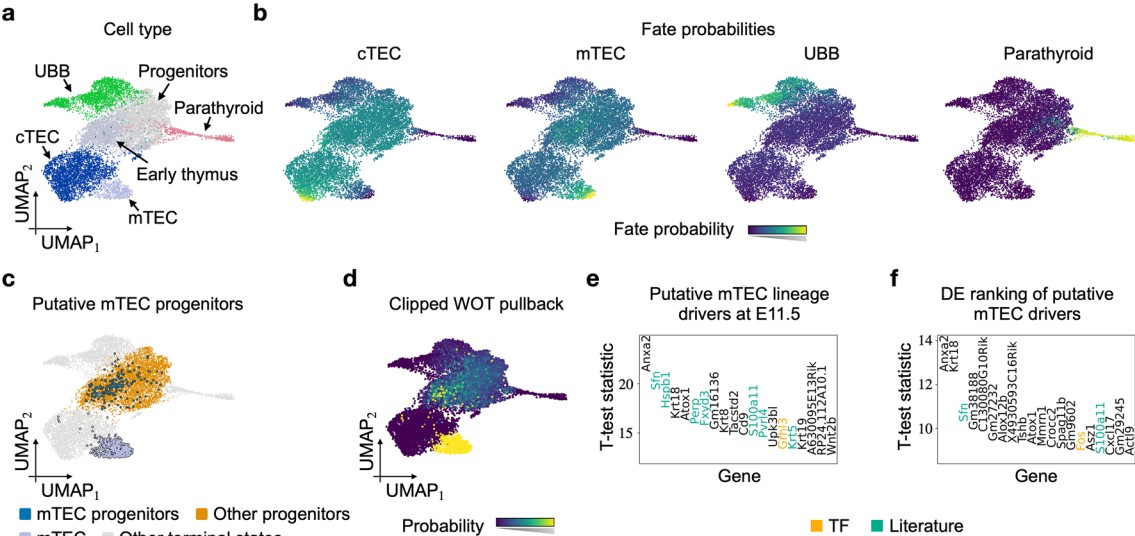

**Extended Data Fig. 9 | Putative mTEC progenitors. a**. UMAP embedding of subsetted pharyngeal endoderm dataset; cells are colored according to the original study. **b**. The fate probabilities towards each terminal state are estimated using the RealTimeKernel. **c**. A cluster of putative mTEC progenitors is identified by cells with fate probabilities towards mTEC larger than 0.5. **d**. UMAP embedding colored by WOT's pullback at E10.5 clipped to the 99 percentile. **e**. Gene ranking of potential mTEC drivers based on WOT's pullback distribution at E11.5, compared to the gene ranking based on the pullback at E10.5, as shown in Fig. 4f. **f**. Gene ranking when performing classical differential expression analysis on clusters 'mTEC progenitors' and 'Other progenitors' as shown in **c**.

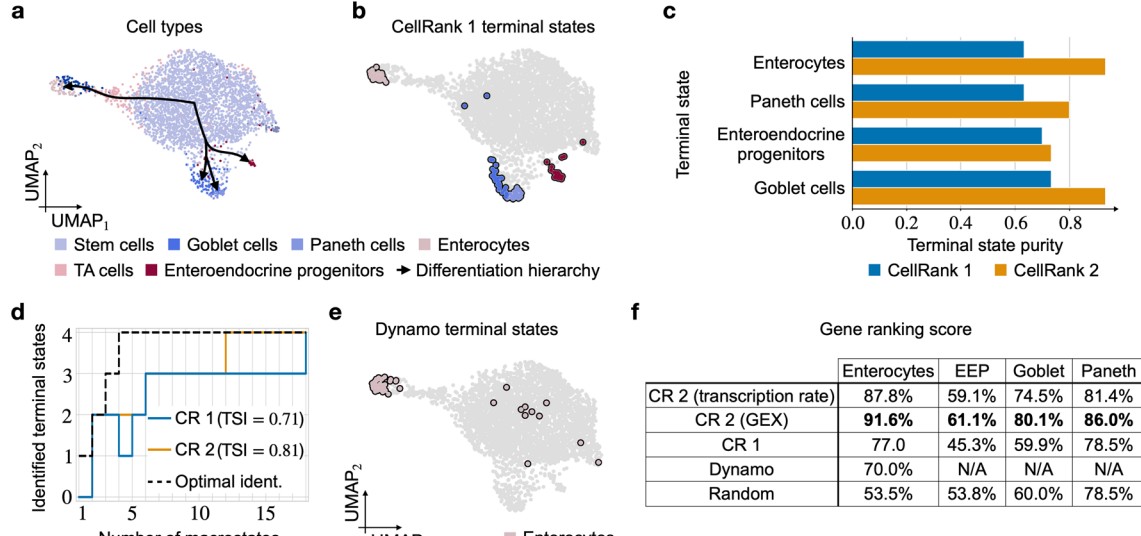

**Extended Data Fig. 10 | Terminal state analysis for metabolically labeled intestinal organoids. a**. UMAP embedding of intestinal cells colored according to their cell type (TA cells: transit-amplifying cells) according to the original publication[9]; black arrows indicate the known differentiation hierarchy. **b**. UMAP embedding highlighting terminal states identified with CellRank 1. **c**. Terminal state purity resulting from our metabolic-labeling-based vector field (CellRank 2) or classical RNA velocity (Cellrank 1). **d**. Terminal state identification using either CellRank 1 (CR 1) or CellRank 2 (CR 2) compared to an optimal identification (Optimal ident.) strategy. **e**. UMAP embedding highlighting terminal states identified by dynamo[16]. **f**. The ranking score metric for each method and terminal state. The metric quantifies the degree of optimality of a gene ranking compared to an optimal ranking. Dynamo only identified the trajectory leading up to enterocytes.

# Reporting Summary

## Statistics

For all statistical analyses, confirm that the following items are present in the figure legend, table legend, main text, or Methods section.

| n/a | Confirmed | |
|---|---|---|
| ☐ | ☒ | The exact sample size (*n*) for each experimental group/condition, given as a discrete number and unit of measurement |
| ☐ | ☒ | A statement on whether measurements were taken from distinct samples or whether the same sample was measured repeatedly |
| ☐ | ☒ | The statistical test(s) used AND whether they are one- or two-sided<br>*Only common tests should be described solely by name; describe more complex techniques in the Methods section.* |
| ☐ | ☒ | A description of all covariates tested |
| ☐ | ☒ | A description of any assumptions or corrections, such as tests of normality and adjustment for multiple comparisons |
| ☐ | ☒ | A full description of the statistical parameters including central tendency (e.g. means) or other basic estimates (e.g. regression coefficient) AND variation (e.g. standard deviation) or associated estimates of uncertainty (e.g. confidence intervals) |
| ☐ | ☒ | For null hypothesis testing, the test statistic (e.g. *F*, *t*, *r*) with confidence intervals, effect sizes, degrees of freedom and *P* value noted<br>*Give P values as exact values whenever suitable.* |
| ☐ | ☒ | For Bayesian analysis, information on the choice of priors and Markov chain Monte Carlo settings |
| ☒ | ☐ | For hierarchical and complex designs, identification of the appropriate level for tests and full reporting of outcomes |
| ☐ | ☒ | Estimates of effect sizes (e.g. Cohen's *d*, Pearson's *r*), indicating how they were calculated |

*Our web collection on statistics for biologists contains articles on many of the points above.*

## Software and code

Policy information about availability of computer code

| Data collection | No software was used |
|---|---|

| Data analysis | Python 3.10 |
|---|---|

anndata            0.8.0
scanpy             1.9.3
h5py               3.8.0
igraph             0.10.14
joblib             1.2.0
leidenalg          0.9.1
llvmlite           0.38.1
loompy             3.0.7
louvain            0.8.0
matplotlib         3.5.3
numba              0.56.4
numpy              1.23.5
pandas             2.0.3
scipy              1.10.1
scvelo             0.3.1
cellrank           2.0.2
cellrank2_reproducibility https://github.com/theislab/cellrank2_reproducibility

For manuscripts utilizing custom algorithms or software that are central to the research but not yet described in published literature, software must be made available to editors and reviewers. We strongly encourage code deposition in a community repository (e.g. GitHub). See the Nature Portfolio guidelines for submitting code & software for further information.

# Data

Policy information about availability of data

All manuscripts must include a data availability statement. This statement should provide the following information, where applicable:
- Accession codes, unique identifiers, or web links for publicly available datasets
- A description of any restrictions on data availability
- For clinical datasets or third party data, please ensure that the statement adheres to our policy

All data presented in this study is publicly available via the original publications; we provide additional access to each dataset, i.e., the PBMC, embryoid body development, mouse embryonic fibroblast, pharyngeal endoderm development, and intestinal organoid data, via a FigShare collection: https://doi.org/10.6084/m9.figshare.c.6843633.v1.

# Research involving human participants, their data, or biological material

Policy information about studies with human participants or human data. See also policy information about sex, gender (identity/presentation), and sexual orientation and race, ethnicity and racism.

| Reporting on sex and gender | N/A |
|---|---|
| Reporting on race, ethnicity, or other socially relevant groupings | N/A |
| Population characteristics | N/A |
| Recruitment | N/A |
| Ethics oversight | N/A |

Note that full information on the approval of the study protocol must also be provided in the manuscript.

# Field-specific reporting

Please select the one below that is the best fit for your research. If you are not sure, read the appropriate sections before making your selection.

☒ Life sciences          ☐ Behavioural & social sciences          ☐ Ecological, evolutionary & environmental sciences

For a reference copy of the document with all sections, see nature.com/documents/nr-reporting-summary-flat.pdf

# Life sciences study design

All studies must disclose on these points even when the disclosure is negative.

| Sample size | This study relied on previously published datasets, we did not generate new data. Thus, we used the sample size provided via the published datasets, which allowed the original authors to reach their conclusions. |
|---|---|
| Data exclusions | No data were excluded from the analysis. |

| Replication | All of the findings reported in this study are reproducible based on code in the CellRank 2 reproducibility GitHub repository (code availability). |
| Randomization | Not applicable, as this study relied on previously published datasets. |
| Blinding | Blinding was not relevant to this study as there were no case-control comparisons made. |

# Reporting for specific materials, systems and methods

We require information from authors about some types of materials, experimental systems and methods used in many studies. Here, indicate whether each material, system or method listed is relevant to your study. If you are not sure if a list item applies to your research, read the appropriate section before selecting a response.

## Materials & experimental systems

| n/a | Involved in the study |
|---|---|
| ☒ | Antibodies |
| ☒ | Eukaryotic cell lines |
| ☒ | Palaeontology and archaeology |
| ☒ | Animals and other organisms |
| ☒ | Clinical data |
| ☒ | Dual use research of concern |
| ☒ | Plants |

## Methods

| n/a | Involved in the study |
|---|---|
| ☒ | ChIP-seq |
| ☒ | Flow cytometry |
| ☒ | MRI-based neuroimaging |

## Plants

| Seed stocks | N/A |
| Novel plant genotypes | N/A |
| Authentication | N/A |

