## [Peer Review File · Nature Methods]

Peer Review Information

Manuscript Title: CellRank 2: Unified fate mapping in multiview single-cell data

Corresponding author name(s): Fabian Theis

Editorial Notes:

Reviewer Comments & Decisions:

Decision Letter, initial version:
--

13th Sep 2023

Dear Professor Theis,

Your Brief Communication, "Unified fate mapping in multiview single-cell data", has now been seen by 3 reviewers. As you will see from their comments below, although the reviewers find your work of considerable potential interest, they have raised a number of concerns. We are interested in the possibility of publishing your paper in Nature Methods, but would like to consider your response to these concerns before we reach a final decision on publication.

We therefore invite you to revise your manuscript to fully address all these concerns.

* include a point-by-point response to the reviewers and to any editorial suggestions

* please underline/highlight any additions to the text or areas with other significant changes to facilitate review of the revised manuscript

- * address the points listed described below to conform to our open science requirements
- * ensure it complies with our general format requirements as set out in our guide to authors at www.nature.com/naturemethods
- * resubmit all the necessary files electronically by using the link below to access your home page

[REDACTED]

We hope to receive your revised paper within 2 months. If you cannot send it within this time, please let us know. In this event, we will still be happy to reconsider your paper at a later date so long as nothing similar has been accepted for publication at Nature Methods or published elsewhere.

OPEN SCIENCE REQUIREMENTS

REPORTING SUMMARY AND EDITORIAL POLICY CHECKLISTS

DATA AVAILABILITY

We strongly encourage you to deposit all new data associated with the paper in a persistent repository where they can be freely and enduringly accessed. We recommend submitting the data to discipline-specific and community-recognized repositories; a list of repositories is provided here:

<http://www.nature.com/sdata/policies/repositories>

All novel DNA and RNA sequencing data, protein sequences, genetic polymorphisms, linked genotype and phenotype data, gene expression data, macromolecular structures, and proteomics data must be deposited in a publicly accessible database, and accession codes and associated hyperlinks must be provided in the “Data Availability” section.

Please include a “Data availability” subsection in the Online Methods. This section should inform readers about the availability of the data used to support the conclusions of your study, including accession codes to public repositories, references to source data that may be published alongside the paper, unique identifiers such as URLs to data repository entries, or data set DOIs, and any other statement about data availability. At a minimum, you should include the following statement: “The data that support the findings of this study are available from the corresponding author upon request”, describing which data is available upon request and mentioning any restrictions on availability. If DOIs are provided, please include these in the Reference list (authors, title, publisher (repository name), identifier, year). For more guidance on how to write this section please see: <http://www.nature.com/authors/policies/data/data-availability-statements-data-citations.pdf>

CODE AVAILABILITY

Please include a “Code Availability” subsection in the Online Methods which details how your custom code is made available. Only in rare cases (where code is not central to the main conclusions of the paper) is the statement “available upon request” allowed (and reasons should be specified).

For more information on our code sharing policy and requirements, please see: <https://www.nature.com/nature-research/editorial-policies/reporting-standards#availability-of-computer-code>

MATERIALS AVAILABILITY

ORCID

Sincerely,

Lin Tang, PhD
Senior Editor
Nature Methods

Reviewers' Comments:

Reviewer #1:

Remarks to the Author:

The authors presented CellRank2, which is an extended package of the published work CellRank. CellRank2 has a few innovations compared to CellRank, including the modular structure (which allows future addition of more data modalities), the usage of information from different data types, and the reduced running time. The authors demonstrated that by exploiting information in the dataset with the transition kernels developed in CellRank2, more accurate results like identification of terminal cell types can be achieved compared to individual existing metrics. Overall, this is a promising tool to study temporal dynamics and cell fates in single cell data. I have the following comments to further improve the manuscript:

1. The innovations of CellRank2 compared to CellRank is summarized in Methods but not specifically referred to in the main text, say, the Introduction. Since it is important for readers to understand the differences between these two tools, I suggest that in Introduction a sentence that points to the

summary in Methods is included.

2. Fig. 1c shows the gene expression levels of a given gene along the pseudotime in different lineages. First, these lines are hard to read so the authors may want to increase the width of the lines (also because the colors are light). Second, in order to plot these curves, I assume that one needs to know the cells that belong to each lineage. Since CellRank2 does not seem to project cells to lineages it would be worth clarifying how this information is obtained.

3. A major way to evaluate the performance of CellRank2 is to show that it finds the terminal cell states which are known in the dataset. Since CellRank2 also finds initial states, how does CellRank2 perform in terms of correctly determining the initial states?

4. Figs. 3e-f show the transcription rates and degradation rates of some genes which fall into two regulation strategy categories: destructive and cooperative regulations. These strategies are discussed in Battich et al (ref [15] in the manuscript). Does the same reference or any other references provide support for the predicted transcription rates and degradation rates of each gene shown in Figs. 3e-f?

5. Although many kernels are introduced, most of the results presented focus on using one single kernel. It is natural to ask the question: can we benefit from using more than one kernels when applicable? It is briefly mentioned in the Discussion that different kernels can be combined with user-defined weighting, but it would be helpful to see some examples on improvement of performance by using more than one kernels, for example, PseudotimeKernel plus VelocityKernel. In Supp. Fig. 4, these two kernels are used separately. From Supp. Fig. 4c, each of these two kernels can have some advantages, and it may help to better identify the cell states by combining these two kernels together. Including results like this can provide users an idea whether using a combination of more than one kernels is encouraged. The authors are also suggested to expand discussion on the potential benefits of using combinations of kernels.

6. On a similar note as the point above, Supp Fig. 1 should be extended to include possibilities of using combinations of kernels.

7. There are multiple minor problems in the figure and description of Supp. Fig. 4: in panel (a), the color codes of some cell types are not shown; Description of (a) says "The data subset of interest is highlighted in black" but there is not anything highlighted in black in the figure. Description of (b): should be "left" and "right" instead of "top" and "bottom".

8. Minor points in Fig. 3: (1) Fig. 3d, curves for CR2(GEX) and Dynamo use very similar colors and are hard to distinguish. (2) bottom line of that page in figure description: "exemplary gene A" -> "exemplary cell A".

Reviewer #2:

Remarks to the Author:

The authors report a new computational method for inferring trajectories and calculating the probability matrix for cell-to-cell transitions from single-cell RNA sequencing data. This method extends CellRank from the authors' previous work to calculate the probability matrix by incorporating additional pieces of information. With these new functionalities, CellRank2 can calculate transition

matrices using various input data types, including precalculated pseudo-time information, real-time point data, and metabolic labeling data. The issues addressed in the manuscript, which CellRank2 aims to resolve, are important challenges in trajectory estimation using single-cell RNA data. While some of the kernels implemented in CellRank2 are built upon existing algorithms, the new integrated implementations in CellRank2 are customized to achieve specific new functionalities, and they hold significance. Additionally, CellRank2 has improved calculation efficiency. Overall, the method holds substantial significance, and I anticipate that it will contribute to the broader field of single-cell analysis studies. The paper is mostly well-written, but there are a few aspects I found unclear and misleading. I would appreciate clarification on these points. Furthermore, while the primary advantage of CellRank2 compared to CellRankv1 is its ability to accommodate different types of data, rather than solely enhancing accuracy, the comparison of results between the two versions is insufficient. It is crucial to clarify whether the new results presented in the manuscript have been derived from new functionality, or the results could have been obtained using CellRankv1.

Here are specific comments

Major Comment 1:

- In Figure 1a, a total of six input data modalities are shown: RNA velocity, Similarity, Pseudotime, Cyto TRACE, Time-series, and Metabolic labeling. This illustration can give the impression that CellRank 2 requires all 6 modalities, but that is not the case. CellRank 2 requires one of the datasets shown here, and the interpretation can vary depending on the input data. I would recommend clarifying this point.
- In relation to the point above, it was unclear whether CellRank incorporates two (or more) modalities simultaneously. Do we need to select one of the transition matrices for downstream analysis? Or is it possible to use all six modalities together if multi-modal data is available? In other words, can we calculate a Markov Chain based on an integrated transition matrix if we input two pieces of information together, such as Pseudotime and Metabolic labeling? An algorithm that can analyze multiple data simultaneously is very different from an algorithm that can analyze one independently. In summary, please clarify in the Figure and main text whether Cellrank 2 can analyze multiple data simultaneously, or if the interpretation of each module of CellRank's multi-modal calculation is performed independently.
- Additionally, please add some explanation in the main text on how you interpret it if the transition matrices calculated from two different sources show different patterns. Showing some examples are helpful.

Major comment 2:

- I found that it is clear that CellRank 2 has new functions that can incorporate new data types, but most of the results and interpretations are not compared with those of CellRank v1. For example, in Line 60, the authors claimed, "Our approach for incorporation experimental time recovers terminal states more faithfully" However, the comparison target in the context of "more faithfully" was unclear. I think it would be necessary to perform the same analysis as in Figure 2 using CellRank v1 and demonstrate that CellRank 2 can extract better results than CellRank v1 if claiming to be "more faithful."
- In relation to the point above, the authors compared the results with different methods in Figure 3c and d, but the comparison did not include CellRank v1. Please display the terminal state inference

results by using CellRank v1 in Figure 3c, and the ranking analysis results with CellRank v1 in Figure 3d.

- Similar to the above, it is recommended to incorporate CellRank v1 analysis in all applicable analyses. This will clarify the advantages of incorporating new data modalities with CellRank 2.
- Including comparisons similar to Figure 1b is also recommended.

Major comment 3

- It is claimed by author that the lineage commitment driver gene can be identified by correlation with Pseudotime, but this is an overstatement. As the author stated in the last discussion section, correlation does not always imply causation. The analysis used here only examines the correlation between pseudotime and gene expression. Regardless of the accuracy of pseudotime and trajectory information, the correlation analysis between pseudotime and gene expression does not have any rationale to distinguish causation from correlation. For example, the correlation analysis identifies not only lineage commitment driver genes but also many downstream genes induced as a result of differentiation; these are inevitably picked up. The author referred to this point in the last discussion section, but some statements in the text and figures are still unclear or misleading on this point. I think it is very likely that many readers will misunderstand this point, and they may think that CellRank 2 can distinguish causation.
- I think the analysis performed by CellRank 2 is not lineage commitment "driver" gene identification, but rather lineage "correlated" gene identification. We need to be very careful about the choice of words and the explanation about them, as they can distort interpretation. I would recommend using "lineage-correlated gene identification" rather than "Lineage commitment driver gene identification". If the author still wants to use the word "driver", I think it is necessary to give up using the word "identified". Overall, I think it is much more accurate to use either of the following descriptions:
Suggestion 1: CellRank 2 identified lineage-correlated genes.
Suggestion 2: CellRank 2 enriched lineage commitment driver genes through correlation analysis.
- Specifically, I would like to recommend editing the following text: Line 70, 134, 174, 210, 224, (249), 253, 290

Major comment 4:

- In reference to the analysis of the initial state in Line 149 paragraph, the initial state inferred from CellRank v1 should be compared with the results obtained from CytoTRACEKernel."

Reviewer #3:

Remarks to the Author:

Summary

The main contribution of this work is software approaching a "one-stop-shop" for trajectory inference that is semi-supervised using a variety of time-like labels: pseudotime, sample collection times, RNA velocities, and differentiation potential (CytoTRACE). The software can handle millions of cells. Specific claims about the method include that it can recover terminal states and fate probabilities; recover genes promoting certain cell fates; and pinpoint regulatory strategies. The main demonstrations are in mouse endoderm, mouse thymus development, human haematopoiesis, and metabolically labeled murine intestinal organoids. A key point of the haematopoiesis example is the ability to overcome

limitations of RNA velocity by using pseudotime instead; this improves estimation of fate probabilities for the erythrocyte lineage. A key point of the endoderm example is ability to nominate known endoderm driver genes. A key point of the thymus example is detection of thymic epithelial progenitors biased toward medullary fates, which is not obvious from e.g. UMAP. A key point of the intestinal organoid example is superior detection of terminal states compared to a competing method, Dynamo.

Software test

From the `cellrank2_reproducibility` repo, I ran the installation steps on a Dell XPS running Ubuntu 20.04. The installation mostly worked, though it also produced an error:

```
"ERROR: pip's dependency resolver does not currently take into account all the packages that are installed. This behaviour is the source of the following dependency conflicts.
bucketcache 0.12.1 requires decorator>=4.0.2, which is not installed.
bleach 5.0.0 requires webencodings, which is not installed.
genomepy 0.12.0 requires appdirs, which is not installed.
genomepy 0.12.0 requires colorama, which is not installed.
genomepy 0.12.0 requires requests, which is not installed.
goatools 1.2.3 requires requests, which is not installed.
ipykernel 6.13.0 requires traitlets>=5.1.0, which is not installed.
ipywidgets 7.7.0 requires ipython-genutils~=0.2.0, which is not installed.
ipywidgets 7.7.0 requires traitlets>=4.3.1, which is not installed.
jupyter-console 6.4.3 requires pygments, which is not installed.
nbconvert 6.5.0 requires beautifulsoup4, which is not installed.
nbconvert 6.5.0 requires entrypoints>=0.2.2, which is not installed.
nbconvert 6.5.0 requires jupyter-core>=4.7, which is not installed.
nbconvert 6.5.0 requires pygments>=2.4.1, which is not installed.
nbconvert 6.5.0 requires traitlets>=5.0, which is not installed.
notebook 6.4.11 requires ipython-genutils, which is not installed.
notebook 6.4.11 requires jupyter-core>=4.6.1, which is not installed.
notebook 6.4.11 requires traitlets>=4.2.1, which is not installed.
qtconsole 5.3.0 requires ipython-genutils, which is not installed.
qtconsole 5.3.0 requires jupyter-core, which is not installed.
qtconsole 5.3.0 requires pygments, which is not installed.
qtconsole 5.3.0 requires traitlets, which is not installed."
```

Running ``import scanpy as sc`` in the `cr2-py38` environment, I get ``ImportError: cannot import name 'is_categorical' from 'pandas.api.types'``. See: <https://stackoverflow.com/questions/76234312/importerror-cannot-import-name-is-categorical-from-pandas-api-types>

CR2 devs can either update scanpy (if this has been fixed in scanpy) or require ``pandas<2.0.0``. Downgrading pandas solves that issue.

Next I encountered this issue:

<https://stackoverflow.com/questions/74947992/how-to-remove-the-error-systemerror-initialization-of-internal-failed-without>

I was unable to solve it after 10-15 minutes of trying different numpy or numba versions. I fell back on the official install instructions from the cellrank2 docs, which direct the user to do

```
conda install -c conda-forge cellrank
```

But this yielded the same problem, even in a fresh environment.

The Python scientific computing ecosystem is infamous for these types of problems, so this is not a distinctive problem with cellrank2. But, it seems reasonable to want to install cellrank2 on any unix system, and if that is not the case, the authors should discuss hardware and OS requirements or perhaps provide a working installation in a docker or singularity image. As it stands, these installation issues will greatly impede adoption of cellrank2 by new users, as well as the ability of reviewers to comment on the analyses in the paper and the experience of using cellrank2.

I was able to install CR2 on MacOS 12.3 with

```
conda install -c conda-forge cellrank
```

However, when I encountered several errors when applying CR to data from outside the tutorials. Specially, CR was unable to predict initial and terminal states based on GPCCA estimators generated with PseudotimeKernel.

Strengths

- The scalability of the software is impressive.
- The modularity is also highly appealing, and users will be able to try diverse methods more easily since they are collected here under a uniform API.
- The methods are impressively clear, especially given such a large and complex software contribution.
- The gut organoid metabolic labeling example has a convincing ground truth and shows much better performance than Dynamo.
- Figure 2 and Supplementary figure 7 nicely justify the choice of the RealTimeKernel over WOT.

Weaknesses

The manuscript misses opportunities to be transparent and systematic about CellRank2's limitations and its optimal use. In specific:

- The study would benefit from more systematic benchmarking to justify or amend Supplementary Figure 1. Incorporation of diverse kernels is a key advantage of CellRank2 over alternatives, but the guidelines for choosing a kernel seem arbitrary. Clearly RNA velocity can be helpful for trajectory inference in some cases (see e.g. VeloViz, DOI: 10.1093/bioinformatics/btab653), but the haematopoiesis results show that RNA velocity can also harm accuracy, and the revealed preference of

the authors in the endoderm example in Suppl. figure 6 is to avoid splicing-based velocity. Both examples contradict Supplementary Figure 1, which should prioritize splicing-based velocity in these examples; perhaps splicing-based velocity should be de-emphasized in the decision tree. Similarly, the gut organoid example shows superiority over Dynamo, but does not test other kernels, so it could be better used to inform supp fig 1 if more kernels were tested. Figure 2 and Supplementary figure 7 justify the choice of the RealTimeKernel over WOT, but could also be tested with kernels based on velocity, pseudotime, or CytoTRACE.

- Tests of accuracy mostly rest on correct identification of terminal cell types. It is also critically important to correctly identify progenitors and their fate probabilities. In ref 52, lineage tracing data is paired with scRNA-seq to identify progenitors biased toward the thymic cortex or medulla. Are the mTEC progenitors nominated by CellRank2 more similar to the latter? This would be one way to argue that CellRank2's new hypothesis about mTEC-biased progenitors is correct. Alternatively, could CellRank2 be directly tested on data with paired scRNA and lineage tracing, similar to the examples used by PRESCIENT (doi: 10.1038/s41467-021-23518-w)?

- The text overstates CellRank2's ability to pinpoint genes affecting cell fate, and many relevant references seem to be misunderstood. In the endoderm example, the ref for the "driver gene" IFITM1 (ref 37) is about germ cells, not endoderm. It mentions migration into the endoderm but PGC's themselves are not endoderm. Also, in a genome-wide CRISPR screen (DOI: 10.1038/s41588-019-0408-9), IFITM1 is targeted by 3 guides but is dispensable for endoderm differentiation (see supplement file 4 in that study). In the thymus example, the transcripts Hspb1, Perp, Sfn, Fxyd3, Pvr14, S100a11 are variously called "genes relevant for the thymus lineage", "genes promoting medullary thymic epithelial cell formation", or "driver gene candidates with known functions in mTEC development" (abstract). In support of these claims, the authors cite refs 50-54, none of which supports even the weakest version of this claim. In brief: reference 50 is a descriptive study based on scRNA-seq with no genetic perturbations. It also covers a later time-frame than the example used by CellRank2, so although SFN is mentioned, this is in the context of a post-AIRE mTEC subtype that does not exist in the example data used by CellRank2. Ref 51 develops and characterizes a flow cytometry panel for quantitation of thymic epithelial subsets in pediatric patients. It does not show Hspb1, Perp, Sfn, Fxyd3, Pvr14, or S100a11 as mTEC markers (figure 2c). Ref 52 combines scRNA-seq with lineage tracing to separate out thymic epithelial precursors and determine fate bias. Its extended data figure 2b covers gene sets marking the mTEC-biased precursors, but does not include Hspb1, Perp, Sfn, Fxyd3, Pvr14, or S100a11. Ref 53 is a review article claiming that cTECs arise from *ectoderm* and mTECs from endoderm; this would seem to undermine the credibility of Ref 53 regarding the cTEC-mTEC fate decision, and it would be better to rely on primary literature. Ref 54 is about detection of exosomes from cultured thymic epithelium, and it mentions S100a11 as a tissue-restricted antigen. Its criteria for tissue-restricted antigens specify genes *without* a known role in the thymus.

Minor points

- Legend in supp. figure 4a is missing some elements
- Please include full names for each abbreviation in figure 6a

Author Rebuttal to Initial comments

Point-by-point response to the reviewers' comments

Unified fate mapping in multiview single-cell data

Philipp Weiler^{1,2*}, Marius Lange^{1,2,3*}, Michal Klein^{4†}, Dana Pe'er^{5,6}, Fabian J. Theis^{1,2,7+}

¹ Institute of Computational Biology, Helmholtz Munich, Germany.

² Department of Mathematics, Technical University of Munich, Germany.

³ Department of Biosystems Science and Engineering, ETH Zürich, Basel, Switzerland.

⁴ Apple.

⁵ Program for Computational and Systems Biology, Sloan Kettering Institute, Memorial Sloan Kettering Cancer Center, New York, NY, USA.

⁶ Howard Hughes Medical Institute, 4000 Jones Bridge Road, Chevy Chase, Maryland 20815, United States

⁷ TUM School of Life Sciences Weihenstephan, Technical University of Munich, Germany.

* Equal contribution

† Work partially done while at Helmholtz Munich, Germany

+ Correspondence to fabian.theis@helmholtz-munich.de

In the following, we present our response to the reviewer's comments. We restate their original comments (black), give point-by-point answers (green) to the questions, and copy parts of the text or specific panels (*blue italic*), which directly correspond to comments or refer to them.

Editor comments

Your Brief Communication, "Unified fate mapping in multiview single-cell data", has now been seen by 3 reviewers. As you will see from their comments below, although the reviewers find your work of considerable potential interest, they have raised a number of concerns. We are interested in the possibility of publishing your paper in Nature Methods, but would like to consider your response to these concerns before we reach a final decision on publication.

We therefore invite you to revise your manuscript to fully address all these concerns.

Thank you for the opportunity to revise our submitted manuscript; we also thank the reviewers for their insightful and thorough comments and are very happy with their overall positive critique. Based on this feedback, we performed and included additional analyses that highlight the strength of our proposed framework, how it outperforms other approaches, and how the results of our analyses compare to and align with similar studies. These updates resulted in modifications to existing figures and two new supplementary figures. The main revisions can be summarized as follows:

1. **Benchmarking of CellRank 2 with the previously published CellRank 1 to demonstrate improvement and impact.** For each analysis, we consistently find that CellRank 2 identifies terminal states substantially better than CellRank 1; thus, this comparison highlights the advantage of incorporating new data modalities since CellRank 1 was based on RNA velocity alone. Additionally, in the case of hematopoiesis, we quantitatively highlight how the recovered fate probabilities of the *PseudotimeKernel* recapitulate ground-truth-transitions more faithfully. Similarly, our analysis of murine intestine development already showed how CellRank 2 ranks known drivers better than CellRank 1.
2. **Novel metrics to compare the performance of different kernels.** To compare different kernels, we devised two metrics: The first metric - the terminal state identification (*TSI*)

score - quantifies terminal state identification compared to an optimal strategy. Our second metric - the cross-boundary correctness (*CBC*) - assesses the correctness of inferred transitions between known ancestor and progenitor cell states; this metric is applicable whenever some ground-truth state transitions are known a priori. We use the TSI score to systematically compare the novel kernels against an RNA-velocity-based analysis, finding that our new proposals consistently outperform RNA velocity. Further, the CBC reveals that the *PseudotimeKernel* outperforms the *VelocityKernel* in our analysis of the hematopoiesis dataset.

3. **Comparing analysis results to existing studies.** We compared our predicted putative medullary thymic epithelial cell (mTEC) progenitor population to reportings from an independent study via an enriched gene set comparison. Although this study focuses on a different stage of mTEC development, we found significant agreement between our findings and theirs. Similarly, we expanded our findings on the metabolically labeled data of murine intestine development by highlighting the agreement with previous studies.

We discuss these findings in greater detail in the point-by-point answers to the reviewers below.

Reviewer #1

Summary

The authors presented CellRank2, which is an extended package of the published work CellRank. CellRank2 has a few innovations compared to CellRank, including the modular structure (which allows future addition of more data modalities), the usage of information from different data types, and the reduced running time. The authors demonstrated that by exploiting information in the dataset with the transition kernels developed in CellRank2, more accurate results like identification of terminal cell types can be achieved compared to individual existing metrics. Overall, this is a promising tool to study temporal dynamics and cell fates in single cell data. I have the following comments to further improve the manuscript:

We thank the reviewer for the positive evaluation of our tool and the comments to improve our manuscript and study. Our revisions include clarifying sentences and updated figures for improved readability and understanding. We also devised two metrics to compare the performance of different kernels: The first assesses cell transitions at cell state boundaries and is shown here; the second focuses on terminal states and is discussed below for reviewer 2.

Major comments

1. The innovations of CellRank2 compared to CellRank is summarized in Methods but not specifically referred to in the main text, say, the Introduction. Since it is important for readers to understand the differences between these two tools, I suggest that in Introduction a sentence that points to the summary in Methods is included.

We thank the reviewer for the suggestion to emphasize the difference between the proposed framework and its predecessor and see it as an important point. We updated and added an additional sentence to the introduction to highlight the three main differences between the two tools previously only mentioned in the Methods section:

CellRank 2 generalizes CellRank to exploit the full potential of new data modalities such as pseudotime, developmental potential, and experimental time points to study complex cellular state changes and identify initial and terminal states, fate probabilities, and lineage-correlated genes. Compared to our earlier work, the new framework is modular, applicable to many more data modalities, and substantially faster (Methods)

In our analyses, we now also compare each introduced kernel to CellRank 1, finding improvements everywhere due to the use of additional modalities.

2. Fig. 1c shows the gene expression levels of a given gene along the pseudotime in different lineages. First, these lines are hard to read so the authors may want to increase the width of the lines (also because the colors are light). Second, in order to plot these curves, I assume that one needs to know the cells that belong to each lineage. Since CellRank2 does not seem to project cells to lineages it would be worth clarifying how this information is obtained.

We thank the reviewer for pointing out the missing explanation on how gene expression trends are fitted. Trends are fitted in the same manner as previously proposed in the CellRank manuscript (Lange et al. 2022). To refer to our earlier work and give a high-level explanation of the fitting procedure, we updated the description of Figure 1c:

We show lineage-specific trends as proposed in our earlier work²² by fitting generalized additive models to gene expression (y-axis) in pseudotime (x-axis); the contribution of each cell to each lineage is weighted according to CellRank-recovered fate probabilities. Colors correspond to lineages as in b.

Importantly, CellRank's GAM-fitting procedure employs a soft assignment of cells to lineages, thereby accounting for the gradual nature of cellular fate commitment. To improve the readability of the results, we also increased the line width of gene expression trends.

3. A major way to evaluate the performance of CellRank2 is to show that it finds the terminal cell states which are known in the dataset. Since CellRank2 also finds initial states, how does CellRank2 perform in terms of correctly determining the initial states?

We thank the reviewer for this insightful question that we did not address in our manuscript. Given that the initial states in the datasets used in this study are known, we extended each analysis to also show how CellRank 2 successfully recovers these states. We identify the population at the beginning of the differentiation processes by transposing the inferred cell-cell transition matrix to invert the induced Markov chain. Following, we apply our estimator-framework as usual. A corresponding explanation has been added to the Methods section:

Initial state identification

Kernel-derived transition matrices quantify probabilities of cell transitions to putative progenitor states. To estimate initial states, instead, we work with the transposed transition matrix, thereby quantifying transition probabilities from progenitor cells to their putative ancestors. Each kernel automatically row-normalizes the transposed transition matrix.

We updated Supplementary Figures 4, 7, and 8 and added an additional Supplementary Figure 10 to include visualizations of the correctly identified initial states.

4. Figs. 3e-f show the transcription rates and degradation rates of some genes which fall into two regulation strategy categories: destructive and cooperative regulations. These strategies are discussed in Battich et al (ref [15] in the manuscript). Does the same reference or any other references provide support for the predicted transcription rates and degradation rates of each gene shown in Figs. 3e-f?

The genes reported in Figure 3e-f are known drivers and markers of the Goblet lineage (Haber et al. 2017). Consequently, their overall gene expression is expected to increase which is only possible by an increase in transcription or constant transcription but increasing degradation. Similarly, we expect the opposing behavior in the enterocyte lineage. To emphasize this expected behavior more, we added the following sentence to the manuscript:

Although it is so far not possible to directly measure transcription and degradation rates in single-cell sequencing protocols, the increase in transcription rate aligns with the role of these genes as regulators and markers of the Goblet lineage⁶¹.

As mentioned by the reviewer and in our main text, our identified regulatory mechanisms align with the strategies reported in the study introducing the intestinal organoid dataset (Battich et al. 2020). The study also explicitly addresses subsets of genes sharing common rate behavior. Within these groups, we find high agreement between their reported mechanisms and ours for the secretory lineage (83% and 62% for cooperative and destabilizing genes, respectively). In the enterocyte lineage, we observed good agreement among genes identified as exhibiting cooperative behavior (83%) but a larger discrepancy in the destructive case (19%). Considering all genes, over half of the regulatory strategies agree between the original study and ours. It is important to note, however, that the original study did not validate the regulatory behavior experimentally; thus, their findings cannot serve as definitive ground truth.

5. Although many kernels are introduced, most of the results presented focus on using one single kernel. It is natural to ask the question: can we benefit from using more than one kernels when applicable? It is briefly mentioned in the Discussion that different kernels can be combined with user-defined weighting, but it would be helpful to see some examples on improvement of performance by using more than one kernels, for example, PseudotimeKernel plus VelocityKernel. In Supp. Fig. 4, these two kernels are used

separately. From Supp. Fig. 4c, each of these two kernels can have some advantages, and it may help to better identify the cell states by combining these two kernels together. Including results like this can provide users an idea whether using a combination of more than one kernels is encouraged. The authors are also suggested to expand discussion on the potential benefits of using combinations of kernels.

We agree with the reviewer that kernel combination is an integral part of our proposed framework. In fact, the *RealTimeKernel* already poses a concrete example of a kernel combination; similarity-based kernels capturing within-time-point dynamics complement OT-based couplings capturing across-time-point dynamics of cell-cell transitions. Supplementary Figure 8b highlights the advantage of this unified view through kernel combination; our analysis of the pharyngeal endoderm development reaches the same conclusion.

Similar to our analyses of datasets containing experimental time points, we combined the *VelocityKernel* with the *ConnectivityKernel* in the analysis of the hematopoiesis sample. In practice, we also observed that this specific kernel combination improves the condition number of the constructed transition matrix, leading to more robust downstream analyses; the condition number is a measure related to numerical stability.

It is important to note that kernel combinations are case-specific; the applicability of each kernel based on its requirements and assumptions is outlined in Supplementary Figure 1. To evaluate such combinations, we extended the manuscript by including a comparative metric for cases where some ground-truth cell state transitions are known. In such cases, we propose the calculation of a cross-boundary correctness (CBC) score (Gao, Qiao, and Huang 2022). A log ratio of kernel-specific scores then quantifies if one kernel recapitulates ground truth more faithfully than the other. We outline our computation of the CBC in a new paragraph added to the Methods section:

Kernel comparison via the cross-boundary correctness score

While the CellRank 2 framework aims at quantifying cell trajectories, correct transitions between coarse cell states (e.g., cell types) are sometimes known a priori. In such cases, the cross-boundary correctness (CBC) score³⁴ can be used to compare two kernels: Consider two cell states \mathcal{C}_1 and \mathcal{C}_2 , where \mathcal{C}_2 is a progenitor state of \mathcal{C}_1 , a pre-computed nearest-neighbor graph with weights w_{jk} between observations j and k , and denote the neighborhood of observation j by $\mathcal{N}(j)$. The representation of observation j is denoted by x_j ; all cell representations are collected in the matrix X . We define the boundary of \mathcal{C}_1 to \mathcal{C}_2 as all cells with at least one neighbor in \mathcal{C}_2 and denote it by $\partial_{1 \rightarrow 2}\mathcal{C}_1$, i.e.,

$$\partial_{1 \rightarrow 2} \mathcal{C}_1 = \{j \in \mathcal{C}_1 \mid \exists k \in \mathcal{N}(j) : k \in \mathcal{C}_2\}$$

For every boundary cell, we empirically define the velocity $v(j)$ of observation $j \in \mathcal{C}_1$ as

$$v(j) = \sum_{k \in \mathcal{N}(j) \cap \mathcal{C}_2} w_{jk}(x_k - x_j).$$

Similarly, for a given kernel κ , we estimate the velocity of observation j via

$$v^{(\kappa)}(j) = T_{j,:}^{(\kappa)} X - x_j,$$

where $T_{j,:}^{(\kappa)}$ denotes the j -th row of the transition matrix computed with kernel κ . The cross boundary correctness score $\beta^{(\kappa)}(j)$ of cell j under kernel κ is then given by the Pearson correlation between $v(j)$ and $v^{(\kappa)}(j)$.

To compare two kernels κ_1 and κ_2 , for each observation, we compute the log ratio of the corresponding CBC scores $\beta^{(\kappa_1)}(j)$ and $\beta^{(\kappa_2)}(j)$. If the velocity estimate based on kernel κ_1 aligns more with the empirical estimate, the log ratio is positive and negative otherwise. A Welch's t-test can be used to test if kernel κ_1 significantly outperforms kernel κ_2 .

As most cell state transitions in hematopoiesis are well understood, we extended the manuscript to quantify that the *PseudotimeKernel* indeed outperforms the *VelocityKernel* in most cell state transitions. These results are presented in a new Supplementary Figure 5 complemented by the following addition to the main text:

For an additional, quantitative metric, we computed the log ratio of the kernels' cross-boundary correctness scores³⁴ (Methods). This metric provides a quantitative measure of two kernel-derived cell-cell

transition matrices for known transitions between cell states. As indicated by the visualization of fate probabilities, the *PseudotimeKernel* significantly outperforms the competing approach for most cell state transitions (6 out of 8; Supplementary Fig. 5b; Methods). As an alternative comparison, we introduce the terminal state identification (TSI) score to quantify the identification of known terminal states compared to an optimal identification strategy ($TSI=1$; Methods). Our pseudotime-based approach again outperformed the RNA-velocity-based alternative ($TSI=0.9$ vs. $TSI=0.81$; Supplementary Fig. 5c).

Similarly, when comparing combinations of the *VelocityKernel* and *PseudotimeKernel*, choosing only the latter results in the highest cross-boundary correctness score (Response Fig. 1).

Response Fig. 1 | Evaluation of combinations of *PseudotimeKernel* and *VelocityKernel*.

The cross-boundary correctness score evaluated for different combinations of the *PseudotimeKernel* and *VelocityKernel* on the dataset of peripheral blood mononuclear cells (Lance et al. 2022). Box plots indicate the median (center line), interquartile range (hinges), and whiskers at 1.5x interquartile range (N=3747 cells each).

Finally, since our computational framework has been publicly available on GitHub prior to our manuscript, several publications have already utilized it in their analyses. These use cases include examples of kernel combinations to generate biological insights, such as

- The *PseudotimeKernel* with the *VelocityKernel* to study the developmental process in epicardioids (Meier et al. 2023).

- The VelocityKernel, CytoTRACEKernel, and a custom kernel based on a method estimating cell age to reveal the developmental history during human cortical gyrification (X. Wang et al. 2022).

We refer to these use cases in our Discussion section.

6. On a similar note as the point above, Supp Fig. 1 should be extended to include possibilities of using combinations of kernels.

We thank the reviewer for the suggested change to Supplementary Figure 1 that better conveys the capabilities of the kernel architecture of our proposed framework. We updated Supplementary Figure 1 to include a visual representation of the optional combination of kernels. Additionally, we explicitly split the requirements and assumptions of the underlying method for each kernel.

Minor comments

1. There are multiple minor problems in the figure and description of Supp. Fig. 4: in panel (a), the color codes of some cell types are not shown; Description of (a) says “The data subset of interest is highlighted in black” but there is not anything highlighted in black in the figure. Description of (b): should be “left” and “right” instead of “top” and “bottom”.

We thank the reviewer for spotting and pointing out these inconsistencies. We updated the corresponding panel and figure description.

2. Minor points in Fig. 3: (1) Fig. 3d, curves for CR2(GEX) and Dynamo use very similar colors and are hard to distinguish. (2) bottom line of that page in figure description: “exemplary gene A” → “exemplary cell A”.

As for the previous point, we updated the corresponding panel and figure description to remove errors and improve readability.

Reviewer #2

Summary

The authors report a new computational method for inferring trajectories and calculating the probability matrix for cell-to-cell transitions from single-cell RNA sequencing data. This method extends CellRank from the authors' previous work to calculate the probability matrix by incorporating additional pieces of information. With these new functionalities, CellRank2 can calculate transition matrices using various input data types, including precalculated pseudo-time information, real-time point data, and metabolic labeling data. The issues addressed in the manuscript, which CellRank2 aims to resolve, are important challenges in trajectory estimation using single-cell RNA data. While some of the kernels implemented in CellRank2 are built upon existing algorithms, the new integrated implementations in CellRank2 are customized to achieve specific new functionalities, and they hold significance. Additionally, CellRank2 has improved calculation efficiency.

Overall, the method holds substantial significance, and I anticipate that it will contribute to the broader field of single-cell analysis studies. The paper is mostly well-written, but there are a few aspects I found unclear and misleading. I would appreciate clarification on these points.

Furthermore, while the primary advantage of CellRank2 compared to CellRankv1 is its ability to accommodate different types of data, rather than solely enhancing accuracy, the comparison of results between the two versions is insufficient. It is crucial to clarify whether the new results presented in the manuscript have been derived from new functionality, or the results could have been obtained using CellRankv1.

We thank the reviewer for the positive comments about our proposed unified framework for fate mapping and its significance for the single-cell field. We also thank the reviewer for pointing out formulations that were either insufficiently explained or ambiguous; we updated the corresponding sections in our manuscript. To highlight how the information from data modalities not included in CellRank 1 result in improved performance, we added specific benchmarks of the two methods for the different use cases.

Here are specific comments

Major comments

Major Comment 1

- A. In Figure 1a, a total of six input data modalities are shown: RNA velocity, Similarity, Pseudotime, Cyto TRACE, Time-series, and Metabolic labeling. This illustration can give the impression that CellRank 2 requires all 6 modalities, but that is not the case. CellRank 2 requires one of the datasets shown here, and the interpretation can vary depending on the input data. I would recommend clarifying this point.

We thank the reviewer for making us aware of the potential misinterpretation of our overview of CellRank 2. We, therefore, updated Figure 1a and its description to explicitly state that at least one kernel is needed but not all.

- B. In relation to the point above, it was unclear whether CellRank incorporates two (or more) modalities simultaneously. Do we need to select one of the transition matrices for downstream analysis? Or is it possible to use all six modalities together if multi-modal data is available? In other words, can we calculate a Markov Chain based on an integrated transition matrix if we input two pieces of information together, such as Pseudotime and Metabolic labeling? An algorithm that can analyze multiple data simultaneously is very different from an algorithm that can analyze one independently. In summary, please clarify in the Figure and main text whether Cellrank 2 can analyze multiple data simultaneously, or if the interpretation of each module of CellRank's multi-modal calculation is performed independently.

CellRank 2 can infer a transition matrix based on any individual kernel or any combination of kernels, including two or more kernels. We implement kernel combinations as weighted means of the corresponding transition matrices and describe this feature in details in the Methods section "Kernel combination".

Related to the preceding point and comment 5 from reviewer 1, we updated Figure 1a and its description to make the option to combine different kernels more explicit. Our revised Figure 1 now describes the kernel combination more explicitly and stresses that, while any individual kernel is enough

to run CellRank 2, several kernels may be combined. We also added a clarifying sentence to the main text that both a single kernel as well as a combination of kernels can be used:

Depending on the dataset and the biological question, we use a single kernel or combine several kernels into multiview Markov chains.

To guide users in their choice of a kernel, we also updated Supplementary Figure 1 to highlight the requirements and assumptions of each kernel. Finally, we devised two metrics for kernel comparison - the *TSI score* and *CBC score*: The *TSI score* compares terminal state identification to an optimal strategy (see Major comment 2C); the *CBC score* assesses the cross-boundary correctness of inferred transitions between known ancestor and progenitor cell states (see Major comment 5 of reviewer 1).

- C. Additionally, please add some explanation in the main text on how you interpret it if the transition matrices calculated from two different sources show different patterns. Showing some examples are helpful.

We updated the main text related to the analysis of the hematopoietic system to explicitly reiterate that the different results of the *VelocityKernel* and *PseudotimeKernel* are caused by violated RNA velocity model assumption:

This inconsistency to known ground-truth transitions stems from violated assumptions of the RNA velocity model (Supplementary Fig. 3 and Supplementary Note 1).

Additionally, we extended the discussion to emphasize the results derived from different kernels may differ if the underlying assumptions of a kernel are not met:

*Importantly, the proposed kernels lead to different results if the underlying assumptions are violated or not sufficiently satisfied (Supplementary Note 1). For example, the *VelocityKernel* failed to faithfully recapitulate the known differentiation hierarchy of hematopoiesis due to unsatisfied assumptions of the RNA velocity model.*

We quantitatively addressed the discrepancy between the two kernels with the *CBC* metric as discussed in Major comment 5 of reviewer 1; our findings justify using the *PseudotimeKernel* instead of its RNA-velocity-based counterpart. Finally, to provide each kernel's requirements and assumption in a more systematic manner, we updated Supplementary Figure 1.

Major comment 2

- A. I found that it is clear that CellRank 2 has new functions that can incorporate new data types, but most of the results and interpretations are not compared with those of CellRank v1. For example, in Line 60, the authors claimed, "Our approach for incorporation experimental time recovers terminal states more faithfully" However, the comparison target in the context of "more faithfully" was unclear. I think it would be necessary to perform the same analysis as in Figure 2 using CellRank v1 and demonstrate that CellRank 2 can extract better results than CellRank v1 if claiming to be "more faithful."

We thank the reviewer for pointing out the ambiguity of our formulation. Here, we were referring to traditional approaches, such as Waddington OT, to map cellular differentiation across time points. We updated the corresponding sentence to clarify what we compare CellRank 2 to:

Our approach for incorporating experimental time recovers terminal states more faithfully compared to traditional approaches mapping between time points²³, allows studying [...]

Additionally, we analyzed this dataset using CellRank 1's *VelocityKernel* and compared performance in terms of the TSI score, finding that our new *RealTimeKernel* outperforms CellRank 2 (see Major comment 2C below for more details).

- B. In relation to the point above, the authors compared the results with different methods in Figure 3c and d, but the comparison did not include CellRank v1. Please display the terminal state inference results by using CellRank v1 in Figure 3c, and the ranking analysis results with CellRank v1 in Figure 3d.

Related to the previous point, we updated Figure 3d to clarify that we already compared the performance to CellRank 1. Previously, our legend read scVelo as we computed RNA velocity using scVelo, and passed the resulting vector field to CellRank 1 for further analysis. In addition, we updated Supplementary Figure 13 to also show the terminal states identified by CellRank 1.

- C. Similar to the above, it is recommended to incorporate CellRank v1 analysis in all applicable analyses. This will clarify the advantages of incorporating new data modalities with CellRank 2.

To continue the previous point, some of our analyses already included an explicit comparison to CellRank 1; specifically, both the analysis using the *PseudotimeKernel* and our proposed approach for computing kinetic rates using metabolically labeled transcripts included benchmarks against CellRank 1.

To compare the performance of CellRank 2 with CellRank 1, and more generally, any set of kernels, we devised a new metric: the terminal state identification (TSI) score. In an ideal scenario, with every additional macrostate, a new terminal state is identified, and the number of identified terminal states increases strictly monotonically until all terminal states have been identified. For a given kernel, the TSI score is defined as the terminal state identification relative to the optimal identification. We outline the computation of the TSI score in a new paragraph in the Methods section:

The terminal state identification score for kernel comparison

If terminal states of the studied system are known a priori, kernels can be compared by considering how well the kernels identify terminal states with increasing number of macrostates. An optimal strategy identifies a new terminal state with every added macrostate, until all terminal states have been identified. We summarize the performance of an arbitrary kernel relative to such an optimal identification with the terminal state identification (TSI) score: Consider a system containing m terminal states and the function f that assigns each number of macrostates n the corresponding number of identified terminal states. In the case of a strategy that identifies terminal states optimally, f_{opt} describes the step function

$$f_{\text{opt}}(n) = n,$$

for $n < m$ and $f_{\text{opt}}(n) = m$ otherwise. We define the TSI score for an arbitrary kernel κ as the area under the curve f_{κ} relative to the area under the curve f_{opt} , i.e.,

$$TSI(\kappa) = \frac{\sum_{n=1}^{N_{\text{max}}} f_{\kappa}(n)}{\sum_{n=1}^{N_{\text{max}}} f_{\text{opt}}} = \frac{2}{m(1 + N_{\text{max}} - m)} \sum_{n=1}^{N_{\text{max}}} f_{\kappa}(n),$$

with maximum number of macrostates assessed N_{max} .

The TSI score showed that the new data modalities incorporated through CellRank 2 indeed improved the identification of terminal states and, thus, demonstrated an improvement over CellRank 1. We updated Supplementary Figures 8 and 13 and added new Supplementary Figures 5 and 10 to visualize the TSI of the kernels used for each analysis and include comparisons with CellRank 1. Additionally, the TSI scores are reported in the main text for each analysis. As we were unable to obtain the data required to run RNA velocity, i.e., either raw FASTQ files or processed spliced/unspliced counts, from the original authors of the embryoid body development example, we could not conduct the comparison between CellRank 1 and CellRank 2 in this case.

D. Including comparisons similar to Figure 1b is also recommended.

We agree that comparisons to CellRank 1 are an important aspect to highlight the usefulness of alternative data modalities. However, we would like to stress that we do not recommend kernel comparisons based on low-dimensional streamline projections. These projections of high-dimensional velocity fields have been shown to be error-prone and sensitive to input data such as the considered set of genes (Soneson et al. 2021; Zheng et al. 2023). Additionally, stream plots are only applicable to neighbor-graph-based kernels, i.e., not the *RealTimeKernel*, for example.

Instead of visualizing streamlines, kernels can be compared by (1) visualizing the high-dimensional random walks in lower dimensions as shown in Supplementary Figure 8b, for example, (2) the TSI score, or (3) the CBC score. We discuss TSI scores for every applicable

analysis as outlined in our response to the previous point; the CBC score is used to justify the use of the *PseudotimeKernel* instead of *VelocityKernel* on the hematopoiesis data

Major comment 3

- A. It is claimed by author that the lineage commitment driver gene can be identified by correlation with Pseudotime, but this is an overstatement. As the author stated in the last discussion section, correlation does not always imply causation. The analysis used here only examines the correlation between pseudotime and gene expression. Regardless of the accuracy of pseudotime and trajectory information, the correlation analysis between pseudotime and gene expression does not have any rationale to distinguish causation from correlation. For example, the correlation analysis identifies not only lineage commitment driver genes but also many downstream genes induced as a result of differentiation; these are inevitably picked up. The author referred to this point in the last discussion section, but some statements in the text and figures are still unclear or misleading on this point. I think it is very likely that many readers will misunderstand this point, and they may think that CellRank 2 can distinguish causation.

We agree with the reviewer that our current analysis lacks a basis for distinguishing between correlation and causation. Consequently, we updated our manuscript to only refer to identified genes as drivers if their relevance for lineage commitment is known; in the other cases, we now refer to these genes as lineage-correlated genes or putative driver genes.

Further, we would like to stress that we correlate gene expression with fate probabilities to identify putative driver genes, rather than with pseudotime. Correlating with fate probabilities ensures that our candidate genes are lineage-specific.

- B. I think the analysis performed by CellRank 2 is not lineage commitment "driver" gene identification, but rather lineage "correlated" gene identification. We need to be very careful about the choice of words and the explanation about them, as they can distort interpretation. I would recommend using "lineage-correlated gene identification" rather than "Lineage commitment driver gene identification". If the author still wants to use the word "driver", I think it is necessary to give up using the word "identified". Overall, I think it is much more accurate to use either of the following descriptions:

- a. Suggestion 1: CellRank 2 identified lineage-correlated genes.
- b. Suggestion 2: CellRank 2 enriched lineage commitment driver genes through correlation analysis.

Related to the previous point, we agree with the reviewer that the proposed formulation describes our approach more accurately. We updated our manuscript accordingly.

- C. Specifically, I would like to recommend editing the following text: Line 70, 134, 174, 210, 224, (249), 253, 290

Continuing the preceding point, we thank the reviewer for the detailed list of recommended edits and applied most suggestions. We kept the formulations in lines 224, 249, 253, and 290 unchanged as they specifically refer to known lineage drivers, *i.e.*, a subset of our lineage-correlated genes.

Major comment 4

- In reference to the analysis of the initial state in Line 149 paragraph, the initial state inferred from CellRank v1 should be compared with the results obtained from CytoTRACEKernel."

We thank the reviewer for this suggestion. We agree that such an analysis would be insightful and highlight the capabilities of our proposed framework beyond terminal state identification. Consequently, we used the proposed kernels to infer initial states for every dataset analyzed in this study. We updated Supplementary Figures 4, 7, and 8 and added an additional Supplementary Figure 10 to include visualizations of the correctly identified initial states. However, we were unable to perform the same analysis with CellRank 1 on the embryoid body development data as we were unable to obtain the data required to run RNA velocity from the authors of the study.

Reviewer #3

Summary

The main contribution of this work is software approaching a "one-stop-shop" for trajectory inference that is semi-supervised using a variety of time-like labels: pseudotime, sample collection times, RNA velocities, and differentiation potential (CytoTRACE). The software can handle millions of cells. Specific claims about the method include that it can recover terminal states and fate probabilities; recover genes promoting certain cell fates; and pinpoint regulatory strategies. The main demonstrations are in mouse endoderm, mouse thymus development, human haematopoiesis, and metabolically labeled murine intestinal organoids. A key point of the haematopoiesis example is the ability to overcome limitations of RNA velocity by using pseudotime instead; this improves estimation of fate probabilities for the erythrocyte lineage. A key point of the endoderm example is ability to nominate known endoderm driver genes. A key point of the thymus example is detection of thymic epithelial progenitors biased toward medullary fates, which is not obvious from e.g. UMAP. A key point of the intestinal organoid example is superior detection of terminal states compared to a competing method, Dynamo.

We thank the reviewer for the thorough assessment and evaluation of our method, summary of our aims and results, and valuable feedback and follow-up questions to improve our software and manuscript. Our framework integrates seamlessly with common Python packages; as such, conflicting versions are an unfortunate, yet rare consequence that we regularly address and encourage the community to report. We implemented changes to our and others' software to ensure the reproducibility of the results shown in our manuscript. Additionally, we compared our findings to a similar study on thymic epithelial development, and we revised our assessment of results on the pharyngeal endoderm development analysis.

Software test

From the cellrank2_reproducibility repo, I ran the installation steps on a Dell XPS running Ubuntu 20.04. The installation mostly worked, though it also produced an error:

We thank the reviewer for testing our reproducibility repository on their computational setup. We are dedicated to ensuring users of our framework can install and use it effortlessly, and that we spot software updates from third-party packages which may cause dependency conflicts.

Consequently, we have set up a continuous integration pipeline for CellRank 2 that runs weekly and whenever we implement any changes; additionally, we encourage users to open issues on our GitHub page whenever they encounter problems, have general questions, or would like to request new features. In the past, we have continuously reacted to installation problems reported via GitHub issues and have worked towards resolving these (*e.g.*, GitHub issues 847, 867, 992, and many more at <https://github.com/theislab/cellrank/issues>). We address the different problems encountered individually in the following.

"ERROR: pip's dependency resolver does not currently take into account all the packages that are installed. This behaviour is the source of the following dependency conflicts.

bucketcache 0.12.1 requires decorator>=4.0.2, which is not installed.

bleach 5.0.0 requires webencodings, which is not installed.

genomepy 0.12.0 requires appdirs, which is not installed.

genomepy 0.12.0 requires colorama, which is not installed.

genomepy 0.12.0 requires requests, which is not installed.

goatools 1.2.3 requires requests, which is not installed.

ipykernel 6.13.0 requires traitlets>=5.1.0, which is not installed.

ipywidgets 7.7.0 requires ipython-genutils~0.2.0, which is not installed.

ipywidgets 7.7.0 requires traitlets>=4.3.1, which is not installed.

jupyter-console 6.4.3 requires pygments, which is not installed.

nbconvert 6.5.0 requires beautifulsoup4, which is not installed.

nbconvert 6.5.0 requires entrypoints>=0.2.2, which is not installed.

nbconvert 6.5.0 requires jupyter-core>=4.7, which is not installed.

nbconvert 6.5.0 requires pygments>=2.4.1, which is not installed.

nbconvert 6.5.0 requires traitlets>=5.0, which is not installed.

notebook 6.4.11 requires ipython-genutils, which is not installed.

notebook 6.4.11 requires jupyter-core>=4.6.1, which is not installed.

notebook 6.4.11 requires traitlets>=4.2.1, which is not installed.

qtconsole 5.3.0 requires ipython-genutils, which is not installed.

qtconsole 5.3.0 requires jupyter-core, which is not installed.

qtconsole 5.3.0 requires pygments, which is not installed.

qtconsole 5.3.0 requires traitlets, which is not installed."

We note that the reported traceback is not caused by our software but by a third-party tool; CellRank 2 does not depend on `bucketcache`, the Python package leading to the reported error message. To investigate the reported issue, we installed the reproducibility repository in a clean conda environment on a Linux and MacOS-based system but did not observe the same problem. Additionally, the provided error message should not interfere with running our reproducibility code and can most likely be resolved by installing the package `decorator`.

Running ``import scanpy as sc`` in the `cr2-py38` environment, I get ``ImportError: cannot import name 'is_categorical' from 'pandas.api.types'``. See: <https://stackoverflow.com/questions/76234312/importerror-cannot-import-name-is-categorical-from-pandas-api-types>

CR2 devs can either update scanpy (if this has been fixed in scanpy) or require ``pandas<2.0.0``. Downgrading pandas solves that issue.

Related to our previous point, one of our highest priorities is making our software useable which entails resolving version conflicts caused by third-party software. CellRank 2 and its dependencies have by now been updated to work with `pandas>=2.0.0` as well.

Next I encountered this issue:

<https://stackoverflow.com/questions/74947992/how-to-remove-the-error-systemerror-initialization-of-internal-failed-without>

I was unable to solve it after 10-15 minutes of trying different numpy or numba versions. I fell back on the official install instructions from the cellrank2 docs, which direct the user to do

```
conda install -c conda-forge cellrank
```

As for the previously encountered issue with `pandas>=2.0.0` CellRank 2 and the reproducibility repository now automatically install compatible versions of `numba` and `numpy`.

But this yielded the same problem, even in a fresh environment.

The Python scientific computing ecosystem is infamous for these types of problems, so this is not a distinctive problem with cellrank2. But, it seems reasonable to want to install cellrank2 on any unix system, and if that is not the case, the authors should discuss hardware and OS requirements or perhaps provide a working installation in a docker or singularity image. As it stands, these installation issues will greatly impede adoption of cellrank2 by new users, as well as the ability of reviewers to comment on the analyses in the paper and the experience of using cellrank2.

I was able to install CR2 on MacOS 12.3 with

```
conda install -c conda-forge cellrank
```

Summarizing our response to the individual issues, we appreciate the reviewer's efforts to install our software and reproduce the reported results, and we regret the encountered problems. To prevent similar experiences by others, we made new CellRank 2 and scVelo releases available through conda; additionally, we updated the installation instructions to reproduce our findings easily.

However, when I encountered several errors when applying CR to data from outside the tutorials. Specially, CR was unable to predict initial and terminal states based on GPCCA estimators generated with PseudotimeKernel.

Continuing our response, we welcome the reviewer applying our tool to their own data. Since CellRank 2 was publicly available prior to our manuscript, other researchers already applied the proposed kernels to different datasets as well; for example, these applications included studies of

- the tumor immune microenvironment with the *CytoTRACEKernel* (Xue et al. 2022)
- the developing human immune system via the *PseudotimeKernel* (Suo et al. 2022)
- healthy and diseased developmental dynamics of the human heart in organoid models with the *VelocityKernel* and *PseudotimeKernel* (Meier et al. 2023).

If users encounter issues running our software, we encourage them to open an issue on our GitHub repository or contact us directly to discuss their specific use case and problem. As a result of this policy, we currently have over 640 closed issues on our GitHub page, many from external users that we supported in applying our framework (see <https://github.com/theislab/cellrank/issues>).

Several reasons can cause a specific kernel to fail to recover the expected initial and terminal states. For the *PseudotimeKernel*, for example, a pseudotime capturing the underlying biological process faithfully is essential. We outline this use case in our Pseudotime tutorial (<https://tinyurl.com/cr2-ptk-tutorial>) and give users practical guidance on choosing a good pseudotime measure.

Strengths

- The scalability of the software is impressive.
- The modularity is also highly appealing, and users will be able to try diverse methods more easily since they are collected here under a uniform API.
- The methods are impressively clear, especially given such a large and complex software contribution.
- The gut organoid metabolic labeling example has a convincing ground truth and shows much better performance than Dynamo.
- Figure 2 and Supplementary figure 7 nicely justify the choice of the *RealTimeKernel* over WOT.

Weaknesses

The manuscript misses opportunities to be transparent and systematic about CellRank2's limitations and its optimal use. In specific:

Major comments

1. The study would benefit from more systematic benchmarking to justify or amend Supplementary Figure 1. Incorporation of diverse kernels is a key advantage of CellRank2 over alternatives, but the guidelines for choosing a kernel seem arbitrary. Clearly RNA velocity can be helpful for trajectory inference in some cases (see e.g. VeloViz, DOI: 10.1093/bioinformatics/btab653), but the haematopoiesis results show that RNA velocity can also harm accuracy, and the revealed preference of the authors in the endoderm example in Suppl. figure 6 is to avoid splicing-based velocity. Both examples contradict Supplementary Figure 1, which should prioritize splicing-based velocity in these examples; perhaps splicing-based velocity should be de-emphasized in the decision tree. Similarly, the gut organoid example shows superiority over Dynamo, but does not test other kernels, so it could be better used to inform supp fig 1 if more kernels were tested. Figure 2 and Supplementary figure 7 justify the choice of the RealTimeKernel over WOT, but could also be tested with kernels based on velocity, pseudotime, or CytoTRACE.

We thank the reviewer for their critical assessment of our proposed guidelines for choosing a kernel. We agree that a pre-defined hierarchy is not ideal and revised Supplementary Figure 1 accordingly. During our adaptation, we removed the hierarchy, explicitly split requirements from underlying assumptions of the methods used to infer the directed differentiation dynamics, and added the option to combine different kernels as we already showed in Figure 1a.

We also agree that RNA-velocity-based analyses can indeed lead to relevant insight in selected cases. However, the corresponding estimates should only be used with the *VelocityKernel* if the assumptions of the RNA velocity model are met. Finally, we would like to emphasize that our analysis of the intestinal organoid dataset already included a comparison with the *VelocityKernel* based on classical RNA velocity as used in CellRank 1. To make this comparison more apparent, we update the legend in Figure 3 to read CellRank 1 instead of scVelo.

We now also include additional analysis to compare different kernels on various data examples, using two new metrics that capture terminal state identification (our TSI score; see our response to Reviewer 2, Major comment 2C) and predicted differentiation direction for transitions with known ground truth (our CBC score; see our response to Reviewer 1, Major comment 5).

2. Tests of accuracy mostly rest on correct identification of terminal cell types. It is also critically important to correctly identify progenitors and their fate probabilities. In ref 52, lineage tracing data is paired with scRNA-seq to identify progenitors biased toward the thymic cortex or medulla. Are the mTEC progenitors nominated by CellRank2 more similar to the latter? This would be one way to argue that CellRank2's new hypothesis about mTEC-biased progenitors is correct. Alternatively, could CellRank2 be directly tested on data with paired scRNA and lineage tracing, similar to the examples used by PRESCIENT (doi: 10.1038/s41467-021-23518-w)?

We agree with the reviewer that using the data of (Nusser et al. 2022) (our old reference 52) represents an opportunity to further validate the mTEC progenitors nominated by CellRank 2. Although both publications study the development of thymic epithelial cells (TECs) in mice, they do so at very different developmental stages: while the data we use in our manuscript (Magaletta et al. 2022) represents initial thymus formation from the pharyngeal endoderm at embryonic days (E) 9.5-12.5, the data from (Nusser et al. 2022) represents much later pre- and postnatal developmental dynamics among TECs at E18.5 and postnatal days (P) 0, 28, and 365 (Response Fig. 2a,b). Thus, as we believe the two datasets are not directly comparable, we decided to cover this additional analysis here, but we did not include it in our revised manuscript.

Among other findings, (Nusser et al. 2022) characterize a population of "Late progenitors" and confirm using lineage tracing that these cells are predominantly mTEC biased. Using their original gene sets, we computed scores quantifying mTEC, cTEC, and late progenitor identities on our data. While the cTEC and mTEC scores marked our cTEC and mTEC populations, respectively, we did not find an enrichment of the Late progenitor score in our progenitor populations ("Progenitors" and "Early thymus"; Response Fig. 2c). Given that the mTEC biased (late) bipotent progenitor population identified in (Nusser et al. 2022) is a postnatal population and the latest day in our data is E12.5, this results was not unexpected. Nevertheless, when we compared the Late progenitor score with CellRank 2-predicted mTEC fate probabilities (Response Fig. 2d), we found a significant positive Pearson correlation ($r = 0.10$, $P = 4.95 \times 10^{-13}$). Similarly, we found a significant negative Pearson correlation of the Late Progenitor Score with CellRank 2 predicted cTEC fate probabilities ($r = -0.05$, $P = 4.83 \times 10^{-4}$; Response Fig. 2e). We computed two-sided p-values using the test implemented in `scipy.stats.pearsonr` over 5,355 cells, which amounts to using an exact distribution of r . Although these correlations are small, they provide some evidence that CellRank 2's fate probabilities recapitulate the cTEC and mTEC-biased populations of (Nusser et al. 2022).

As for lineage tracing data, although not part of this study, we would like to make the reviewer aware of the moslin method (Lange et al. 2023), which uses Optimal Transport (OT) to infer lineage-informed coupling matrices from early to late time points in time-series single-cell studies. CellRank 2 can directly operate on these lineage-informed couplings via the *RealTimeKernel*, as we demonstrate in the moslin preprint for *C. elegans* developmental data. On this dataset, following the approach we outlined in this study, we combine lineage- and state-informed transitions across time points with similarity-informed transitions within time points to uncover terminal states, fate probabilities, and putative driver genes. Similarly, in the moscot preprint (Klein et al. 2023), we use OT to uncover spatiotemporal trajectories of mouse embryogenesis and pass these to CellRank 2's *RealTimeKernel* for further analysis.

Response Fig. 2 | mTEC fate probabilities correlate with late progenitor score.

a,b. UMAPs of the pharyngeal endoderm subset from (Magaletta et al. 2022) that we show in our manuscript (**a**) and of the thymic epithelium data from (Nusser et al. 2022) (**b**), colored by time points (left) and cell types (right). Inlets show pie charts over time points (left) and cell types (right). **c.** cTEC, mTEC, and Late progenitors scores, computed using the gene sets defined by (Nusser et al. 2022), visualized with UMAPs (top) and violin plots (bottom) over our pharyngeal endoderm subset. **d.** UMAPs visualizing fate probabilities computed using CellRank 2's *RealTimeKernel*. **e.** Heatmap showing Pearson correlation of gene set scores (panel **c**; x-axis) versus fate probabilities (panel **d**; y-axis), computed over the "Progenitors" and "Early thymus" clusters of (**a**). The yellow box highlights correlations with the "Late progenitors" score.

3. The text overstates CellRank2's ability to pinpoint genes affecting cell fate, and many relevant references seem to be misunderstood. In the endoderm example, the ref for the "driver gene" IFITM1 (ref 37) is about germ cells, not endoderm. It mentions migration into the endoderm but PGC's themselves are not endoderm. Also, in a genome-wide CRISPR screen (DOI: 10.1038/s41588-019-0408-9), IFITM1 is targeted by 3 guides but is dispensable for endoderm differentiation (see supplement file 4 in that study).

In the thymus example, the transcripts *Hspb1*, *Perp*, *Sfn*, *Fxyd3*, *Pvrl4*, *S100a11* are variously called "genes relevant for the thymus lineage", "genes promoting medullary thymic epithelial cell formation", or "driver gene candidates with known functions in mTEC development" (abstract). In support of these claims, the authors cite refs 50-54, none of which supports even the weakest version of this claim.

In brief: reference 50 is a descriptive study based on scRNA-seq with no genetic perturbations. It also covers a later time-frame than the example used by CellRank2, so although SFN is mentioned, this is in the context of a post-AIRE mTEC subtype that does not exist in the example data used by CellRank2.

Ref 51 develops and characterizes a flow cytometry panel for quantitation of thymic epithelial subsets in pediatric patients. It does not show *Hspb1*, *Perp*, *Sfn*, *Fxyd3*, *Pvrl4*, or *S100a11* as mTEC markers (figure 2c).

Ref 52 combines scRNA-seq with lineage tracing to separate out thymic epithelial precursors and determine fate bias. Its extended data figure 2b covers gene sets marking the mTEC-biased precursors, but does not include *Hspb1*, *Perp*, *Sfn*, *Fxyd3*, *Pvrl4*, or *S100a11*.

Ref 53 is a review article claiming that cTECs arise from *ectoderm* and mTECs from endoderm; this would seem to undermine the credibility of Ref 53 regarding the cTEC-mTEC fate decision, and it would be better to rely on primary literature.

Ref 54 is about detection of exosomes from cultured thymic epithelium, and it mentions *S100a11* as a tissue-restricted antigen. Its criteria for tissue-restricted antigens specify genes *without* a known role in the thymus.

We appreciate and thank the reviewer for the thorough revision of our references. In response to the reviewer's comments, we have made changes to the references we cite, and we are more explicit about the biological system that each study employed; in particular, we removed our old reference 53 (H.-X. Wang et al. 2019). Further, we made the following changes for the genes addressed by the reviewer:

Embryoid bodies (*CytoTRACEKernel* application)

- IFITM1
 - We agree with the reviewer that our old reference 37 (Tanaka et al. 2005) was focused on primordial germ cells migrating into endoderm, rather than endoderm development itself and thank the reviewer for bringing (Li et al. 2019) to our attention; accordingly, we removed IFITM1 from our manuscript.

Pharyngeal endoderm (*RealTimeKernel* application)

- *Hspb1*
 - We respectfully disagree with the reviewer, who claimed that our old reference 52 (Nusser et al. 2022) does not mention *Hspb1*. In fact, the authors of that study write: "However, when assayed by RNA in situ hybridization in the intact thymic lobe of 4-week-old mice, *Hspb1* marks a subset of medullary cells, indicating that

its expression is an intrinsic characteristic of TECs [...]” (see Extended Data Figure 1, panel d legend). Thus, we continue to use this reference as evidence that *Hspb1* is a marker gene of mTECs but now explicitly acknowledge in the text that (Nusser et al. 2022) describes a later window of murine TEC development.

- *Perp* and *Sfn*
 - We agree with the reviewer and acknowledge that our old reference 50 (Bautista et al. 2021) is a descriptive study that does not include genetic perturbations. Thus, in our revised manuscript, we only use this study to ascertain that *Perp*, *Sfn*, and *Fxyd3* are marker genes for corneocyte-like mTECs, rather than lineage drivers. We now acknowledge in the text that (Bautista et al. 2021) describes human rather than mouse TEC development.
 - Both *Perp* and *Sfn* are part of the p53 signaling pathway that controls mTEC differentiation in mice (Rodrigues et al. 2017; Bautista et al. 2021).
 - *Perp* is a target of the TF p63 (Dooley, Erickson, and Farr 2008), which in turn is known to be involved in murine TEC development (Dooley, Erickson, and Farr 2008; Stefanski et al. 2022)
- *Fxyd3*
 - *Fxyd3* marks corneocyte-like mTECs in humans (Bautista et al. 2021).
 - We respectfully disagree with the reviewer regarding our old reference 51 (Haunerding et al. 2021). While we acknowledge that this study is focused on pediatric patients, it confirms *Fxyd3*'s role as a marker gene for the same subset of corneocyte-like mTECs in that setting: “We therefore assessed our dataset from bulk-sorted cTEC- and mTEC subpopulations for signatures associated with proposed subsets. The distribution of genes associated with tuft-like cells (POU2F3, IL25), [...], corneocyte-like cells (FXD3, LYPD2, IL1RN), [...] were found to be upregulated in the respective TEC population (S3C Figure).” (see Results section). Thus, in our revised manuscript, we cite (Haunerding et al. 2021) to confirm the gene's role as a marker gene for corneocyte-like mTECs in pediatric patients.
 - (Carter et al. 2022) further confirm that *Fxyd3* is a marker for a subset of mTECs in pediatric patients.
- *Pvrl4*:
 - (Magaletta et al. 2022), the original study from our dataset, confirms *Pvrl4*'s role as an mTEC marker gene (Fig. 1).
- *S100a11*:
 - We respectfully disagree with the reviewer regarding our old reference 54 (Skogberg et al. 2015). We acknowledge that the authors of that study write that

“Proteins with a known expression in the thymus were not considered as TRAs.” However, in the same paragraph, the authors also write that “Examples of TRAs with a previously described mTEC expression found in the cultured cells were Adenylate kinase 2, (...), and S100 proteins (S100A2, S100A8, S100A9 and S100A11). Also in the exosomes, previously reported mTEC-enriched TRAs such as Glutathione [...] and S100 proteins (S100A8, S100A9 and S100A11) were identified.” We acknowledge in our revised manuscript that (Skogberg et al. 2015) studied cultured human cells.

- (Gotter et al. 2004) confirm that S100A11 is expressed in mTECs (see Fig. 4d) in humans.

Accordingly, we changed the corresponding text in the Pharyngeal endoderm results section to:

Next, we used our correlation-based analysis to identify possible drivers of this fate decision and found TFs (Fos, Grhl3, Elf5) and genes relevant for the thymus lineage among the 20 genes with highest correlation (Fig. 2d): Sfn and Perp are part of the p53 signaling pathway controlling murine mTEC differentiation^{50,51}; additionally, the TF p63 targets Perp and is involved in murine mTEC differentiation^{52,53}. Similarly, we recovered previously reported markers of murine mTECs, including Grhl3, Pvrl4, and Cd9^{40,54}). In addition to these known markers of mTECs in mouse, our top-ranked genes also included S100a11 and Fxyd3, markers of mTEC subpopulations in different human settings^{51,55–58} and Hspb1, a marker of later-stage murine mTECs⁵⁹. Notably, the original study of our dataset identified the TF Grhl3 as a putative early mTEC marker with higher specificity compared to markers traditionally used.

Minor comments

1. Legend in supp. figure 4a is missing some elements

We thank the reviewer for spotting and pointing out these inconsistencies. We updated the corresponding panel and figure description.

2. Please include full names for each abbreviation in figure 6a

Related to the previous point, we added the missing information to the figure legend.

References

- Battich, Nico, Joep Beumer, Buys de Barbanson, Lenno Krenning, Chloé S. Baron, Marvin E. Tanenbaum, Hans Clevers, and Alexander van Oudenaarden. 2020. "Sequencing Metabolically Labeled Transcripts in Single Cells Reveals mRNA Turnover Strategies." *Science* 367 (6482): 1151–56.
- Bautista, Jhoanne L., Nathan T. Cramer, Corey N. Miller, Jessica Chavez, David I. Berrios, Lauren E. Byrnes, Joe Germino, et al. 2021. "Single-Cell Transcriptional Profiling of Human Thymic Stroma Uncovers Novel Cellular Heterogeneity in the Thymic Medulla." *Nature Communications* 12 (1): 1–15.
- Carter, Jason A., Léonie Strömich, Matthew Peacey, Sarah R. Chapin, Lars Velten, Lars M. Steinmetz, Benedikt Brors, Sheena Pinto, and Hannah V. Meyer. 2022. "Transcriptomic Diversity in Human Medullary Thymic Epithelial Cells." *Nature Communications* 13 (1): 1–15.
- Dooley, James, Matthew Erickson, and Andrew G. Farr. 2008. "Alterations of the Medullary Epithelial Compartment in the Aire-Deficient Thymus: Implications for Programs of Thymic Epithelial Differentiation." *Journal of Immunology* 181 (8): 5225–32.
- Gao, Mingze, Chen Qiao, and Yuanhua Huang. 2022. "UniTVelo: Temporally Unified RNA Velocity Reinforces Single-Cell Trajectory Inference." *Nature Communications* 13 (1): 1–11.
- Gotter, Jörn, Benedikt Brors, Manfred Hergenhausen, and Bruno Kyewski. 2004. "Medullary Epithelial Cells of the Human Thymus Express a Highly Diverse Selection of Tissue-Specific Genes Colocalized in Chromosomal Clusters." *The Journal of Experimental Medicine* 199 (2): 155–66.
- Haber, Adam L., Moshe Biton, Noga Rogel, Rebecca H. Herbst, Karthik Shekhar, Christopher Smillie, Grace Burgin, et al. 2017. "A Single-Cell Survey of the Small Intestinal Epithelium." *Nature* 551 (7680): 333–39.
- Hauerdinger, Veronika, Maria Domenica Moccia, Lennart Opitz, Stefano Vavassori, Hitendu Dave, and Mathias M. Hauri-Hohl. 2021. "Novel Combination of Surface Markers for the Reliable and Comprehensive Identification of Human Thymic Epithelial Cells by Flow Cytometry: Quantitation and Transcriptional Characterization of Thymic Stroma in a Pediatric Cohort." *Frontiers in Immunology* 12 (September): 740047.
- Klein, Dominik, Giovanni Palla, Marius Lange, Michal Klein, Zoe Piran, Manuel Gander, Laetitia Meng-Papaxanthos, et al. 2023. "Mapping Cells through Time and Space with Moscot." *bioRxiv*. <https://doi.org/10.1101/2023.05.11.540374>.
- Lance, Christopher, Malte D. Luecken, Daniel B. Burkhardt, Robrecht Cannoodt, Pia Rautenstrauch, Anna Laddach, Aidyn Ubungazhibov, et al. 2022. "Multimodal Single Cell Data Integration Challenge: Results and Lessons Learned." In *NeurIPS 2021 Competitions and Demonstrations Track*, 162–76. PMLR.

- Lange, Marius, Volker Bergen, Michal Klein, Manu Setty, Bernhard Reuter, Mostafa Bakhti, Heiko Lickert, et al. 2022. "CellRank for Directed Single-Cell Fate Mapping." *Nature Methods*, January. <https://doi.org/10.1038/s41592-021-01346-6>.
- Lange, Marius, Zoe Piran, Michal Klein, Bastiaan Spanjaard, Dominik Klein, Jan Philipp Junker, Fabian J. Theis, and Mor Nitzan. 2023. "Mapping Lineage-Traced Cells across Time Points with Moslin." *bioRxiv*. <https://doi.org/10.1101/2023.04.14.536867>.
- Li, Qing V., Gary Dixon, Nipun Verma, Bess P. Rosen, Miriam Gordillo, Renhe Luo, Chunlong Xu, et al. 2019. "Genome-Scale Screens Identify JNK-JUN Signaling as a Barrier for Pluripotency Exit and Endoderm Differentiation." *Nature Genetics* 51 (6): 999–1010.
- Lucas, Beth, Andrea J. White, Fabian Klein, Clara Veiga-Villauriz, Adam Handel, Andrea Bacon, Emilie J. Cosway, et al. 2023. "Embryonic keratin19+ Progenitors Generate Multiple Functionally Distinct Progeny to Maintain Epithelial Diversity in the Adult Thymus Medulla." *Nature Communications* 14 (1): 1–14.
- Magaletta, Margaret E., Macrina Lobo, Eric M. Kernfeld, Hananeh Aliee, Jack D. Huey, Teagan J. Parsons, Fabian J. Theis, and René Maehr. 2022. "Integration of Single-Cell Transcriptomes and Chromatin Landscapes Reveals Regulatory Programs Driving Pharyngeal Organ Development." *Nature Communications* 13 (1): 457.
- Meier, Anna B., Dorota Zawada, Maria Teresa De Angelis, Laura D. Martens, Gianluca Santamaria, Sophie Zengerle, Monika Nowak-Imialek, et al. 2023. "Epicardioid Single-Cell Genomics Uncovers Principles of Human Epicardium Biology in Heart Development and Disease." *Nature Biotechnology*, April, 1–14.
- Nusser, Anja, Sagar, Jeremy B. Swann, Brigitte Krauth, Dagmar Diekhoff, Lesly Calderon, Christiane Happe, Dominic Grün, and Thomas Boehm. 2022. "Developmental Dynamics of Two Bipotent Thymic Epithelial Progenitor Types." *Nature* 606 (7912): 165–71.
- Rodrigues, Pedro M., Ana R. Ribeiro, Chiara Perrod, Jonathan J. M. Landry, Leonor Araújo, Isabel Pereira-Castro, Vladimir Benes, et al. 2017. "Thymic Epithelial Cells Require p53 to Support Their Long-Term Function in Thymopoiesis in Mice." *Blood* 130 (4): 478–88.
- Skogberg, Gabriel, Vanja Lundberg, Martin Berglund, Judith Gudmundsdottir, Esbjörn Telemo, Susanne Lindgren, and Olov Ekwall. 2015. "Human Thymic Epithelial Primary Cells Produce Exosomes Carrying Tissue-Restricted Antigens." *Immunology and Cell Biology* 93 (8): 727–34.
- Sonesson, Charlotte, Avi Srivastava, Rob Patro, and Michael B. Stadler. 2021. "Preprocessing Choices Affect RNA Velocity Results for Droplet scRNA-Seq Data." *PLoS Computational Biology* 17 (1): e1008585.
- Stefanski, Heather E., Yan Xing, Jemma Nicholls, Leslie Jonart, Emily Goren, Patricia A. Taylor, Alea A. Mills, et al. 2022. "P63 Targeted Deletion under the FOXP1 Promoter Disrupts Pre- and Post-Natal Thymus Development, Function and Maintenance as Well as Induces Severe Hair Loss." *PloS One* 17

(1): e0261770.

- Suo, Chenqu, Emma Dann, Issac Goh, Laura Jardine, Vitalii Kleshchevnikov, Jong-Eun Park, Rachel A. Botting, et al. 2022. "Mapping the Developing Human Immune System across Organs." *Science* 376 (6597): eabo0510.
- Tanaka, Satomi S., Yasuka L. Yamaguchi, Bonny Tsoi, Heiko Lickert, and Patrick P. L. Tam. 2005. "IFITM/Mil/fragilis Family Proteins IFITM1 and IFITM3 Play Distinct Roles in Mouse Primordial Germ Cell Homing and Repulsion." *Developmental Cell* 9 (6): 745–56.
- Wang, Hong-Xia, Wenrong Pan, Lei Zheng, Xiao-Ping Zhong, Liang Tan, Zhanfeng Liang, Jing He, Pingfeng Feng, Yong Zhao, and Yu-Rong Qiu. 2019. "Thymic Epithelial Cells Contribute to Thymopoiesis and T Cell Development." *Frontiers in Immunology* 10: 3099.
- Wang, Xinghuan, Wan Jin, Gang Wang, Lingao Ju, Fangjin Chen, Kaiyu Qian, Yu Xiao, and Yi Zhang. 2022. "Tracking Single Cell Evolution via Clock-like Chromatin Accessibility." *bioRxiv*.
<https://doi.org/10.1101/2022.05.12.491736>.
- Xue, Ruidong, Qiming Zhang, Qi Cao, Ruirui Kong, Xiao Xiang, Hengkang Liu, Mei Feng, et al. 2022. "Liver Tumour Immune Microenvironment Subtypes and Neutrophil Heterogeneity." *Nature* 612 (7938): 141–47.
- Zheng, Shijie C., Genevieve Stein-O'Brien, Leandros Boukas, Loyal A. Goff, and Kasper D. Hansen. 2023. "Pumping the Brakes on RNA Velocity by Understanding and Interpreting RNA Velocity Estimates." *Genome Biology* 24 (1): 246.

Decision Letter, first revision:

Our ref: NMETH-BC53215A

30th Jan 2024

Dear Dr. Theis,

Thank you for submitting your revised manuscript "Unified fate mapping in multiview single-cell data" (NMETH-BC53215A). It has now been seen by the original referees and their comments are below. The reviewers find that the paper has improved in revision, and therefore we'll be happy in principle to publish it in Nature Methods, pending minor revisions to satisfy the referees' final requests and to comply with our editorial and formatting guidelines.

We are now performing detailed checks on your paper and will send you a checklist detailing our

editorial and formatting requirements within two weeks or so. Please do not upload the final materials and make any revisions until you receive this additional information from us.

TRANSPARENT PEER REVIEW

Please note: we allow redactions to authors' rebuttal and reviewer comments in the interest of confidentiality. If you are concerned about the release of confidential data, please let us know specifically what information you would like to have removed. Please note that we cannot incorporate redactions for any other reasons. Reviewer names will be published in the peer review files if the reviewer signed the comments to authors, or if reviewers explicitly agree to release their name. For more information, please refer to our FAQ page.

ORCID

Sincerely,

Lin Tang, PhD
Senior Editor
Nature Methods

Reviewer #1 (Remarks to the Author):

The authors have addressed my concerns.

Reviewer #1 (Remarks on code availability):

I did not run the code but went through the GitHub repository. The tool appear to be well documented. There are users actively using the tool and posting issues. The authors have been responsive to the issues.

Reviewer #2 (Remarks to the Author):

The authors answered most of the comments, clarifying the method details and applications of CellRankv2. Regarding the new results added by the author, I have the following request. CellRank2 is compatible with different types of input data. It is very important to compare the results for interpretation and safe usage. The authors have added several metrics for comparison and benchmarking. I would like to request tutorials and documentation for CBC and TSI. I could not find them in the current documentation.

Reviewer #2 (Remarks on code availability):

I did not have enough time to review the codes in this review cycle.

Reviewer #3 (Remarks to the Author):

The authors have addressed my questions and concerns

Reviewer #3 (Remarks on code availability):

we examined the code associated with the initial submission

Author Rebuttal, first revision:

Point-by-point response to the reviewers' comments

CellRank 2: Unified fate mapping in multiview single-cell data

Philipp Weiler^{1,2*}, Marius Lange^{1,2,3*}, Michal Klein^{4†}, Dana Pe'er^{5,6}, Fabian J. Theis^{1,2,7+}

¹ Institute of Computational Biology, Helmholtz Munich, Germany.

² Department of Mathematics, Technical University of Munich, Germany.

³ Department of Biosystems Science and Engineering, ETH Zürich, Basel, Switzerland.

⁴ Apple.

⁵ Program for Computational and Systems Biology, Sloan Kettering Institute, Memorial Sloan Kettering Cancer Center, New York, NY, USA.

⁶ Howard Hughes Medical Institute, 4000 Jones Bridge Road, Chevy Chase, Maryland 20815,
United States

⁷ TUM School of Life Sciences Weihenstephan, Technical University of Munich, Germany.

* Equal contribution

† Work partially done while at Helmholtz Munich, Germany

† Correspondence to fabian.theis@helmholtz-munich.de

In the following, we present our response to the reviewer's comments. We restate their original comments (black), give point-by-point answers (green) to the questions, and copy parts of the text or specific panels (*blue italic*), which directly correspond to comments or refer to them.

Editor comments

Thank you for submitting your revised manuscript "Unified fate mapping in multiview single-cell data" (N METH-BC53215A). It has now been seen by the original referees and their comments are below. The reviewers find that the paper has improved in revision, and therefore we'll be happy in principle to publish it in Nature Methods, pending minor revisions to satisfy the referees' final requests and to comply with our editorial and formatting guidelines.

TRANSPARENT PEER REVIEW

Nature Methods offers a transparent peer review option for new original research manuscripts submitted from 17th February 2021. We encourage increased transparency in peer review by publishing the reviewer comments, author rebuttal letters and editorial decision letters if the authors agree. Such peer review material is made available as a supplementary peer review file. Please state in the cover letter 'I wish to participate in transparent peer review' if you want to opt in, or 'I do not wish to participate in transparent peer review' if you don't. Failure to state your preference will result in delays in accepting your manuscript for publication.

Please note: we allow redactions to authors' rebuttal and reviewer comments in the interest of confidentiality. If you are concerned about the release of confidential data, please let us know specifically what information you would like to have removed. Please note that we cannot incorporate redactions for any other reasons. Reviewer names will be published in the peer review files if the reviewer signed the comments to authors, or if reviewers explicitly agree to release their name. For more information, please refer to our FAQ page.

We thank the reviewers for their comments, further recommendations, and feedback. Based on the reviewer's feedback and the shared checklist, we have made the following main revisions:

- 1. Reformatting the manuscript.** We restructured the figures to accommodate the manuscript being published as an Article. The original Figure 1 was split into two; previous panels a (now Figure 1), and panels b-c (now Figure 2). Parts of formerly Supplementary Figure 7a-e have been reorganized into Figure 3. Supplementary Figures 3 and 11 have been kept as Supplementary Figures, all other formerly Supplementary Figures have been updated if necessary and are now Extended Data Figures.
- 2. Accessibility of TSI and CBC scores.** We included the TSI and CBC scores in the CellRank 2 source code; kernels and estimators can now compute the corresponding metrics. We uploaded example use cases to our reproducibility repository.

We wish to participate in transparent peer review, and as requested, we provide Twitter handles for making announcements (@PhilippWeiler7, @MariusLange8, @dana_peer, @fabian_theis).

Reviewer #1

Remarks to the Author

The authors have addressed my concerns.

Remarks on code availability

I did not run the code but went through the GitHub repository. The tool appear to be well documented. There are users actively using the tool and posting issues. The authors have been responsive to the issues.

We thank the reviewer for their time and positive assessment of our work.

Reviewer #2

Remarks to the Author

The authors answered most of the comments, clarifying the method details and applications of CellRankv2. Regarding the new results added by the author, I have the following request.

CellRank2 is compatible with different types of input data. It is very important to compare the results for interpretation and safe usage. The authors have added several metrics for comparison and benchmarking. I would like to request tutorials and documentation for CBC and TSI. I could not find them in the current documentation.

We thank the reviewer for their positive evaluation of our revision. We agree with the reviewer that the introduced metrics are valuable beyond this study. We thus incorporated them into our software and uploaded example applications to our reproducibility repository.

Remarks on code availability

I did not have enough time to review the codes in this review cycle.

Reviewer #3

Remarks to the Author

The authors have addressed my questions and concerns

Remarks on code availability

We examined the code associated with the initial submission.

We thank the reviewer for their time and positive assessment of our work.

Final Decision Letter:

9th May 2024

Dear Professor Theis,

Thank you very much for your patience when we evaluate the revised version of your Article of "CellRank 2: Unified fate mapping in multiview single-cell data". I am pleased to inform you that it has now been accepted for publication in Nature Methods. The received and accepted dates will be 18th Jul 2023 and 9th May 2024. This note is intended to let you know what to expect from us over the next month or so, and to let you know where to address any further questions.

Over the next few weeks, your paper will be copyedited to ensure that it conforms to Nature Methods style. Once your paper is typeset, you will receive an email with a link to choose the appropriate publishing options for your paper and our Author Services team will be in touch regarding any additional information that may be required. It is extremely important that you let us know now whether you will be difficult to contact over the next month. If this is the case, we ask that you send us the contact information (email, phone and fax) of someone who will be able to check the proofs and deal with any last-minute problems.

Please note that Nature Methods is a Transformative Journal (TJ). Authors may publish their research with us through the traditional subscription access route or make their paper immediately open access through payment of an article-processing charge (APC). Authors will not be required to make a final decision about access to their article until it has been accepted. Find out more about Transformative Journals

Authors may need to take specific actions to achieve compliance with funder and institutional open access mandates. If your research is supported by a funder that requires immediate open access (e.g. according to Plan S principles) then you should select the gold OA route, and we will direct you to the compliant route where possible. For authors selecting the subscription publication route, the journal's standard licensing terms will need to be accepted, including self-archiving policies. Those licensing terms will supersede any other terms that the author or any third party may assert apply to any version of the manuscript.

You may wish to make your media relations office aware of your accepted publication, in case they consider it appropriate to organize some internal or external publicity. Once your paper has been scheduled you will receive an email confirming the publication details. This is normally 3-4 working days in advance of publication. If you need additional notice of the date and time of publication, please let the production team know when you receive the proof of your article to ensure there is sufficient time

to coordinate. Further information on our embargo policies can be found here:
<https://www.nature.com/authors/policies/embargo.html>

Please feel free to contact me if you have questions about any of these points. Thank you very much for publishing your paper at Nature Methods!

Best regards,

Lin Tang, PhD
Senior Editor
Nature Methods